# Interleukin-11-expressing fibroblasts have a unique gene signature correlated with poor prognosis of colorectal cancer

Takashi Nishina [1]✉, Yutaka Deguchi[1], Daisuke Ohshima[2], Wakami Takeda[1,3], Masato Ohtsuka[4,5], Shigeyuki Shichino[6], Satoshi Ueha[6], Soh Yamazaki[1], Mika Kawauchi[1], Eri Nakamura[7], Chiharu Nishiyama [3], Yuko Kojima[8], Satomi Adachi-Akahane [2], Mizuho Hasegawa[9], Mizuho Nakayama[10], Masanobu Oshima [10], Hideo Yagita[11], Kazutoshi Shibuya[12], Tetuo Mikami[13], Naohiro Inohara[9], Kouji Matsushima[6], Norihiro Tada[7] & Hiroyasu Nakano [1,14]✉

Interleukin (IL)-11 is a member of the IL-6 family of cytokines and is involved in multiple cellular responses, including tumor development. However, the origin and functions of IL-11-producing (IL-11⁺) cells are not fully understood. To characterize IL-11⁺ cells in vivo, we generate *Il11* reporter mice. IL-11⁺ cells appear in the colon in murine tumor and acute colitis models. *Il11ra1* or *Il11* deletion attenuates the development of colitis-associated colorectal cancer. IL-11⁺ cells express fibroblast markers and genes associated with cell proliferation and tissue repair. IL-11 induces the activation of colonic fibroblasts and epithelial cells through phosphorylation of STAT3. Human cancer database analysis reveals that the expression of genes enriched in IL-11⁺ fibroblasts is elevated in human colorectal cancer and correlated with reduced recurrence-free survival. IL-11⁺ fibroblasts activate both tumor cells and fibroblasts via secretion of IL-11, thereby constituting a feed-forward loop between tumor cells and fibroblasts in the tumor microenvironment.

[1] Department of Biochemistry, Toho University School of Medicine, Tokyo, Japan. [2] Department of Physiology, Toho University School of Medicine, Tokyo, Japan. [3] Laboratory of Molecular Biology and Immunology, Department of Biological Science and Technology, Faculty of Industrial Science and Technology, Tokyo University of Science, Tokyo, Japan. [4] Department of Molecular Life Science, Division of Basic Medical Science and Molecular Medicine, School of Medicine, Tokai University, Isehara, Kanagawa, Japan. [5] The Institute of Medical Sciences, Tokai University, Isehara, Kanagawa, Japan. [6] Division of Molecular Regulation of Inflammatory and Immune Diseases, Research Institute for Biomedical Sciences, Tokyo University of Science, Chiba, Japan. [7] Research Institute for Diseases of Old Age, Juntendo University School of Medicine, Tokyo, Japan. [8] Laboratory of Morphology and Image Analysis, Research Support Center, Juntendo University Graduate School of Medicine, Tokyo, Japan. [9] Department of Pathology, University of Michigan Medical School, Ann Arbor, MI, USA. [10] WPI Nano Life Science Institute (WPI-Nano LSI), Division of Genetics, Cancer Research Institute, Kanazawa University, Kanazawa, Ishikawa, Japan. [11] Department of Immunology, Juntendo University Graduate School of Medicine, Tokyo, Japan. [12] Department of Surgical Pathology, Toho University School of Medicine, Tokyo, Japan. [13] Department of Pathology, Toho University School of Medicine, Tokyo, Japan. [14] Host Defense Research Center, Toho University School of Medicine, Tokyo, Japan. ✉email: takashi.nishina@med.toho-u.ac.jp; hiroyasu.nakano@med.toho-u.ac.jp

Maintenance of intestinal homeostasis involves a variety of cell types, including epithelial, immune, and stromal cells[1–3]. Within the intestinal lamina propria, stromal cells include fibroblasts, α-smooth muscle actin (αSMA)-positive myofibroblasts, endothelial cells, and pericytes[4,5]. These stromal cells organize the tissue architecture and have recently been revealed to play crucial roles in regulating immune responses, tissue repair, and tumor development[3–5]. Recent studies have focused on fibroblasts that can support tumor growth, termed cancer-associated fibroblasts (CAFs)[6–8]. In a recent study, single-cell RNA sequencing (scRNA-seq) was performed to analyze colon biopsies from healthy individuals and ulcerative colitis (UC) patients. The results revealed that UC patients' colon samples include a unique subset of fibroblasts, termed inflammation-associated fibroblasts (IAFs), with high expression of *Interleukin 11* (*IL11*), *IL24*, *IL13RA2*, and *TNFSFR11B*[9]. Another study reported that constitutive STAT3 activation in collagen VI (ColVI)-expressing fibroblasts enhances azoxymethane (AOM) and dextran sulfate sodium (DSS)-induced colitis-associated colorectal cancer (CAC) in mice[10]. Conversely, deletion of *Stat3* in ColVI+ fibroblasts attenuates the development of CAC. Together, these findings suggest that stromal fibroblasts may constitute different cell populations and play a crucial role in the development of cancer and colitis, but the full picture of fibroblast heterogeneity and function remains unclear.

IL-11 is a multifunctional member of the IL-6 family, contributing to hematopoiesis, bone development, tissue repair, and tumor development[11,12]. The IL-11 receptor comprises IL-11Rα1, which binds IL-11, and gp130, which transmits signals to the nucleus via Janus kinase (JAK) activation[13]. JAKs phosphorylate STAT3, and phosphorylated STAT3 enters the nucleus where it activates the transcription of various target genes associated with cell proliferation and cell survival[14–17]. IL-11 production is regulated by several cytokines, including TGFβ, IL-1β, IL-17A, and IL-22[18–21]. We have previously demonstrated that IL-11 production is induced by reactive oxygen species (ROS) and the electrophile 1,2-naphthoquinone. This, in turn, promotes liver and intestine tissue repair[22,23]. While IL-11 is reportedly produced in various cell types (including stromal, hematopoietic, and epithelial cells) in response to different stimuli[16,17,24,25], the cellular sources of IL-11 in vivo are not fully understood.

TGFβ expression is frequently elevated in human CRC and is correlated with a high risk of tumor recurrence[24]. Tumor-derived TGFβ induces *Il11* expression in stromal fibroblasts in a paracrine manner, whereas enforced expression of TGFβ in tumor cells promotes metastasis in tumor xenograft model[24]. Moreover, TGFβ induces expression of a long noncoding RNA activated by TGFβ (LncRNA-ATB), which subsequently elevates IL-11 production by stabilizing *Il11* mRNA in tumor cells[26]. Consistently, IL-11 expression is elevated in CRCs in humans and mice[10,17,24,27]. Repeated injection of IL-11 results in an increase in the number and size of colorectal tumors in AOM/DSS-treated mice[10]. Conversely, deletion of the *Il11ra1* gene attenuates CAC development in mice treated with AOM/DSS or in adenomatous polyps of mice harboring a mutation in the *Adenomatous polyposis coli* (*Apc*) gene (*Apc*^Min/+)[17]. Together, these results suggest there is intimate crosstalk between TGFβ and IL-11 and that TGFβ-induced tumor promotion is mediated, at least in part, by IL-11. Moreover, recent studies have reported that IL-11 plays a crucial role in fibrosis development in various organs, including the lung, heart, liver, and kidney[28,29].

In this study, we aim to characterize IL-11-producing cells in vivo. We generate *Il11-Enhanced green fluorescence protein (Egfp)* reporter mice and detect IL-11+ cells in the colonic tumor tissues of a CAC mouse model and *Apc*^Min/+ mice. Deletion of *Il11ra* or *Il11* attenuates CAC development in mice. When tumor organoids are transplanted into the colon, IL-11+ cells appear in the tumor tissues. Moreover, IL-11+ cells rapidly appear in mouse colon after DSS treatment. IL-11+ cells express stromal cell markers and genes associated with cell proliferation and tissue repair, suggestive of colonic fibroblasts. IL-11 induces robust STAT3 phosphorylation in both colonic fibroblasts and colon epithelial organoids. Using the human cancer database, we find that the expression of some genes with enriched expression in IL-11+ fibroblasts is also elevated in human CRC and that high expression of a set of these genes is correlated with reduced recurrence-free survival in CRC patients. Together, our present results demonstrate that tumor cells induce IL-11+ fibroblasts and that a feed-forward loop between colon tumor epithelial cells and IL-11+ fibroblasts via secretion of IL-11 may contribute to tumor development.

## Results

**Characterization of IL-11+ cells in CAC using *Il11-Egfp* reporter mice.** To characterize IL-11-producing (IL-11+) cells in vivo, we generated transgenic mice in which *Egfp* expression was under the control of the *Il11* gene promoter, using a bacterial artificial chromosome (BAC) vector (Supplementary Fig. 1a). An *Egfp* cDNA and a polyA signal were inserted in-frame in the second exon of the *Il11* gene (Supplementary Fig. 1a). As expected, *Il11* mRNA expression was correlated with *Egfp* mRNA expression in various tissues (Supplementary Fig. 1b, c). Notably, *Il11* mRNA expression was highest in the testis and was very low in other mouse tissues under normal conditions (Supplementary Fig. 1b).

Although *Il11* mRNA expression is elevated in CRC in mice and humans[16,17], cellular sources of IL-11 remain controversial. We used *Il11-Egfp* reporter mice to monitor the appearance of IL-11+ cells in a CAC model. AOM administration followed by repeated exposure to DSS causes CAC development in mice[30,31]. On day 77 after AOM/DSS administration, the *Il11-Egfp* reporter mice developed large tumors in the colon (Fig. 1a, b). We isolated tumor and nontumor tissues and examined *Il11* mRNA expression using qPCR. *Il11* and *Egfp* mRNA expression was elevated in tumor tissues compared with nontumor tissues from mice with AOM/DSS-induced CAC (Fig. 1c).

We next isolated and characterized IL-11+ cells from tumors of *Il11-Egfp* reporter mice. We observed that the percentage of cells that were EGFP (IL-11)+ was increased in tumor tissues compared with nontumor colon tissues from AOM/DSS-treated *Il11-Egfp* reporter mice (Fig. 1d, Supplementary Fig. 1d). The majority of IL-11+ cells expressed mesenchymal stromal cell markers, such as Thy1.2, podoplanin, CD29, and Sca-1, but not CD31 or Lyve-1, whereas only very small percentages of IL-11+ cells expressed EpCAM (Fig. 1e). Conversely, only 5% of podoplanin+ cells (mostly fibroblasts) expressed IL-11 (Fig. 1f). Immunohistochemistry (IHC) revealed that IL-11+ cells appeared in the stroma surrounding tumor cells (Fig. 1g). Of note, we could not detect EGFP+ cells in the colon of *Il11-Egfp* reporter mice before AOM/DSS treatment, or wild-type mice even after AOM/DSS treatment (Supplementary Fig. 1e), suggesting that GFP signals did not reflect the autofluorescence of the tissues. We found that most IL-11+ cells also expressed GFP (Fig. 1h, i), further substantiating *Il11-Egfp* reporter mice indeed monitored endogenous IL-11+ cells. IL-11+ cells also expressed vimentin, collagen I, and collagen IV, but not αSMA (Supplementary Fig. 1f), suggesting that these cells were fibroblasts but not myofibroblasts. Consistent with the flow cytometry results (Fig. 1f), a few E-cadherin-positive tumor cells were positive for EGFP expression (Fig. 1j). Overall, these results suggest that most IL-11+ cells were stromal fibroblasts, but few IL-11+ cells were colonic epithelial cells.

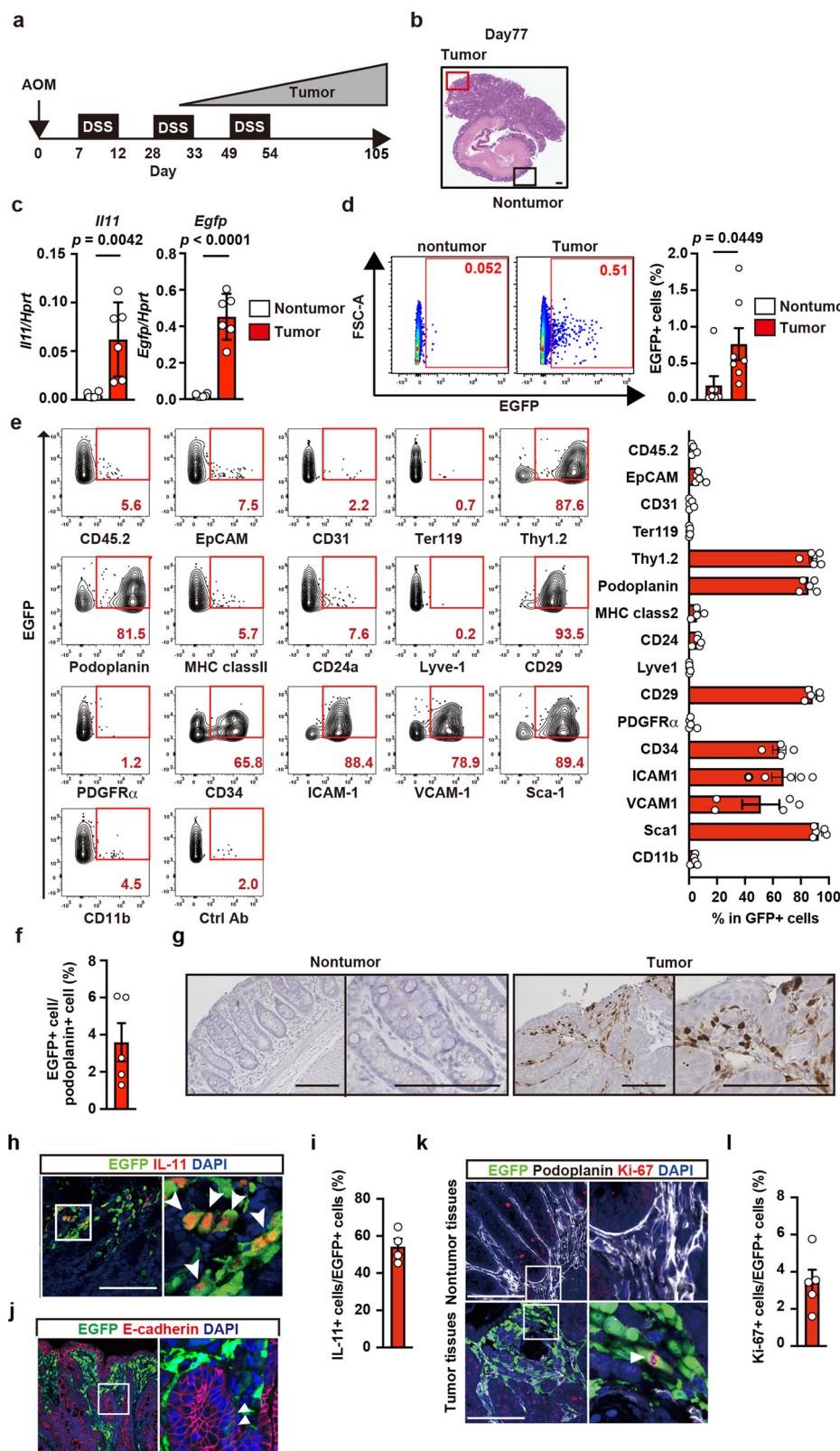

Expression of *Il11* in colons from untreated mice was lower than that in colon tumor tissues from AOM/DSS-treated mice (0.002 in normal colon vs. 0.05 in tumor tissues) (Supplementary Fig. 1b, Fig. 1c). These results suggest that small numbers of resident IL-11[+] cells might proliferate and expand in situ; alternatively, IL-11[−] cells could be converted into IL-11[+] cells after AOM/DSS treatment. Another possibility is that IL-11[+] cells

are recruited to the colon from other tissues. To discriminate among these three possibilities, we investigated whether IL-11[+] cells expressed the cell-proliferating antigen Ki67. We quantified percentages of Ki67[+] and podoplanin[+] cells among EGFP[+] cell populations in nontumor and tumor tissues. We could not detect EGFP[+] cells in nontumor tissues (Fig. 1k, upper panels). Consistent with the flow cytometry results, most EGFP[+] cells

**Fig. 1 Characterization of IL-11$^+$ cells in CAC using *Il11-Egfp* reporter mice. a** Protocol for induction of AOM/DSS-induced CAC in mice. *Il11-Egfp* reporter mice were intraperitoneally injected with AOM on day 0, followed by repeated DSS administration. Colorectal cancer gradually develops ~30 days after AOM injection. Unless otherwise indicated, the following experiments used tumor and nontumor tissues collected on day 98–105 after AOM/DSS treatment. **b** Representative image of tumor and adjacent nontumor tissues in the mouse colon on day 77 after AOM injection. Colon sections were stained with hematoxylin & eosin (H&E) ($n = 20$ mice). Scale bar, 200 μm. **c** mRNA samples from tumor and nontumor tissues were analyzed using qPCR to determine the expression levels of the indicated genes. Results are mean ± SEM ($n = 6$ mice). Results are representative of two independent experiments. **d**–**f** From colonic tumor tissues of *Il11-Egfp* reporter mice, single-cell suspensions were prepared, and percentages of IL-11$^+$ (EGFP$^+$) cells were determined. Representative flow cytometry images are shown (**d**). Results are mean ± SEM ($n = 7$ mice). Cells were stained with the indicated antibodies, and marker expression on EGFP$^+$ cells was analyzed by flow cytometry. A representative dot plot from three independent experiments shows the percentages of EGFP$^+$ cells expressing each marker ($n = 5$ mice except for CD34 where $n = 4$ mice) (**e**). Percentages of EGFP$^+$ cells among podoplanin$^+$ cells (**f**). Results are mean ± SEM ($n = 5$ mice). **g** Nontumor and tumor tissue sections were stained with anti-GFP antibody ($n = 4$ mice). Scale bars, 100 μm. **h**–**l** Tumor sections were stained with anti-GFP along with anti-IL-11 (**h**) ($n = 4$ mice), anti-E-cadherin (**j**) ($n = 4$ mice), or anti-Ki67 (**k**) and anti-podoplanin (**k**) antibodies ($n = 5$ mice). The right-hand panels show enlarged images of white boxes from the left-hand panels. White arrowheads indicate EGFP$^+$ cells expressing the respective markers shown in red. IL-11$^+$ and EGFP$^+$ cell numbers were calculated, and IL-11$^+$/EGFP$^+$ areas as percentages of the total area are shown (**i**). Ki67$^+$ and EGFP$^+$ cell numbers were calculated, and are shown as percentages of the total area (**l**). Results are mean ± SEM ($n = 5$ mice). Scale bars, 100 μm. Statistical significance was determined using the two-tailed unpaired Student's $t$-test (**c**, **d**). Source data are provided as a Source Data file.

expressed podoplanin, while only 4% of EGFP$^+$ cells expressed Ki67 (Fig. 1k lower panels, and Fig. 1l).

We also examined the possibility that IL-11$^+$ cells were derived from bone marrow (BM). We transferred BM cells from *Il11-Egfp* reporter mice to lethally irradiated wild-type mice. One month following reconstitution, *Il11-Egfp* BM chimeric mice were treated with AOM/DSS to induce colorectal tumors (Supplementary Fig. 1g). Although the expression of *Il11* was elevated in tumor tissues compared with non-tumor tissues, the *Egfp* expression that can represent *Il11* expression in BM-derived cells was not elevated in either tumor tissues or non-tumor tissues from *Il11-Egfp* BM chimeric mice (Supplementary Fig. 1h). Notably, cells expressing both EGFP and IL-11 were easily detected in colon tumor tissues of AOM/DSS-treated *Il11-Egfp* reporter mice (Fig. 1h, i). In contrast, only IL-11$^+$ and not EGFP$^+$ cells were detected in the colon tumor tissues of AOM/DSS-treated *Il11-Egfp* BM chimeric mice (Supplementary Fig. 1i). Together, these results suggest that most IL-11$^+$ cells are not likely derived from BM cells, but that IL-11$^-$ cells became IL-11$^+$ cells during tumor development. We refer to IL-11$^+$ cells as IL-11$^+$ fibroblasts hereafter.

**Attenuated CAC development in *Il11ra1*$^{-/-}$ and *Il11*$^{-/-}$ mice.** A previous study reported attenuated CAC development in *Il11ra1*$^{-/-}$ mice[17]. Our present data consistently confirmed the attenuation of AOM/DSS-induced CAC in *Il11ra1*$^{-/-}$ mice compared with wild-type mice (Supplementary Fig. 2a). To further substantiate that the IL-11/IL-11R-dependent signaling pathway contributed to CAC development, we investigated *Il11*$^{-/-}$ mice. As we previously reported[32], *Il11*$^{-/-}$ mice showed no abnormalities. When wild-type and *Il11*$^{-/-}$ mice were treated with AOM/DSS, we observed attenuated CAC development in *Il11*$^{-/-}$ mice compared with wild-type mice (Supplementary Fig. 2b). Moreover, reciprocal BM transfer experiments revealed that *Il11* expression in non-hematopoietic cells might be primarily responsible for the increased tumor numbers and tumor load in the colon (Supplementary Fig. 2c, d). We focused our subsequent analyses on IL-11$^+$ fibroblasts.

**IL-11$^+$ fibroblasts appear in tumor tissues in the absence of colitis.** The above-described results suggested that IL-11$^-$ cells might cell-autonomously become IL-11$^+$ cells in the colonic tissues within the inflammatory milieu or in tumors triggered by AOM/DSS. We used two different colitis-independent murine tumor models to discriminate between these possibilities: adenomatous polyps in *Apc*$^{Min/+}$ mice[33] and tumor organoids transplanted into wild-type mice. To monitor IL-11$^+$ cells, we crossed *Apc*$^{Min/+}$ mice with *Il11-Egfp* reporter mice. In *Apc*$^{Min/+}$ mice, *Il11* and *Egfp* mRNA expression levels were elevated in colon tumors compared with nontumor tissues (Fig. 2a), and IL-11$^+$ cells appeared in the stroma surrounding tumor cells in the colon (Fig. 2b). Consistent with the IHC results, we found increased percentages of IL-11$^+$ cells in the tumor tissues compared with non-tumor colon tissues from *Apc*$^{Min/+}$;*Il11-Egfp* reporter mice by flow cytometry (Fig. 2c). Most IL-11$^+$ cells expressed podoplanin and Thy1.2, and small numbers of IL-11$^+$ cells expressed epithelial cell markers (Fig. 2d). Hence, the majority of IL-11$^+$ cells were fibroblasts. Of note, EGFP$^+$ cells only appeared in the colon tumors of *Apc*$^{Min/+}$;*Il11-Egfp* reporter mice, but not *Apc*$^{Min/+}$ mice (Supplementary Fig. 3a), indicating that EGFP signals in *Apc*$^{Min/+}$;*Il11-Egfp* reporter mice did not likely reflect the autofluorescence, but represented specific signals of IL-11. Again, we observed weak EGFP expression in some E-cadherin-positive tumor cells themselves (Fig. 2e). Only 4% of IL-11$^+$ fibroblasts were positive for Ki67$^+$ (Fig. 2f, g). IL-11$^+$ fibroblasts expressed vimentin, collagen I, and collagen IV, but not αSMA (Fig. 2h). We also found that the expression levels of *Il11* and *Egfp* were elevated in the tumor tissues compared with non-tumor tissues in small intestine from *Apc*$^{Min/+}$;*Il11-Egfp* reporter mice, and IHC revealed that IL-11$^+$ fibroblasts appeared in the tumor tissues (Supplementary Fig. 3b, c). Moreover, consistent with the results from *Apc*$^{min/+}$;*Il11ra1*$^{-/-}$ mice[17], deletion of *Il11* attenuated the development of tumors in the colon and small intestine of *Apc*$^{min/+}$ mice (Fig. 2i, j).

We previously generated tumor organoids from the intestines of AKTP mice; these exhibit mutations in *Apc*, *Kras*, *Tgfbr2*, and *Tp53* in intestinal epithelial cells[34]. These tumor organoids were transplanted into the colons of *Il11-Egfp* reporter mice, and we examined whether IL-11$^+$ fibroblasts appeared along with tumor development (Supplementary Fig. 4a). On day 30 post-transplantation, the tumor organoids developed large tumors and destroyed normal epithelial tissues in the colon (Supplementary Fig. 4b). Importantly, IL-11$^+$ fibroblasts appeared in the tumor tissues (Supplementary Fig. 4c). These IL-11$^+$ fibroblasts expressed podoplanin and vimentin, but not CD45 or E-cadherin (Supplementary Fig. 4d). Together, these results are strong evidence that

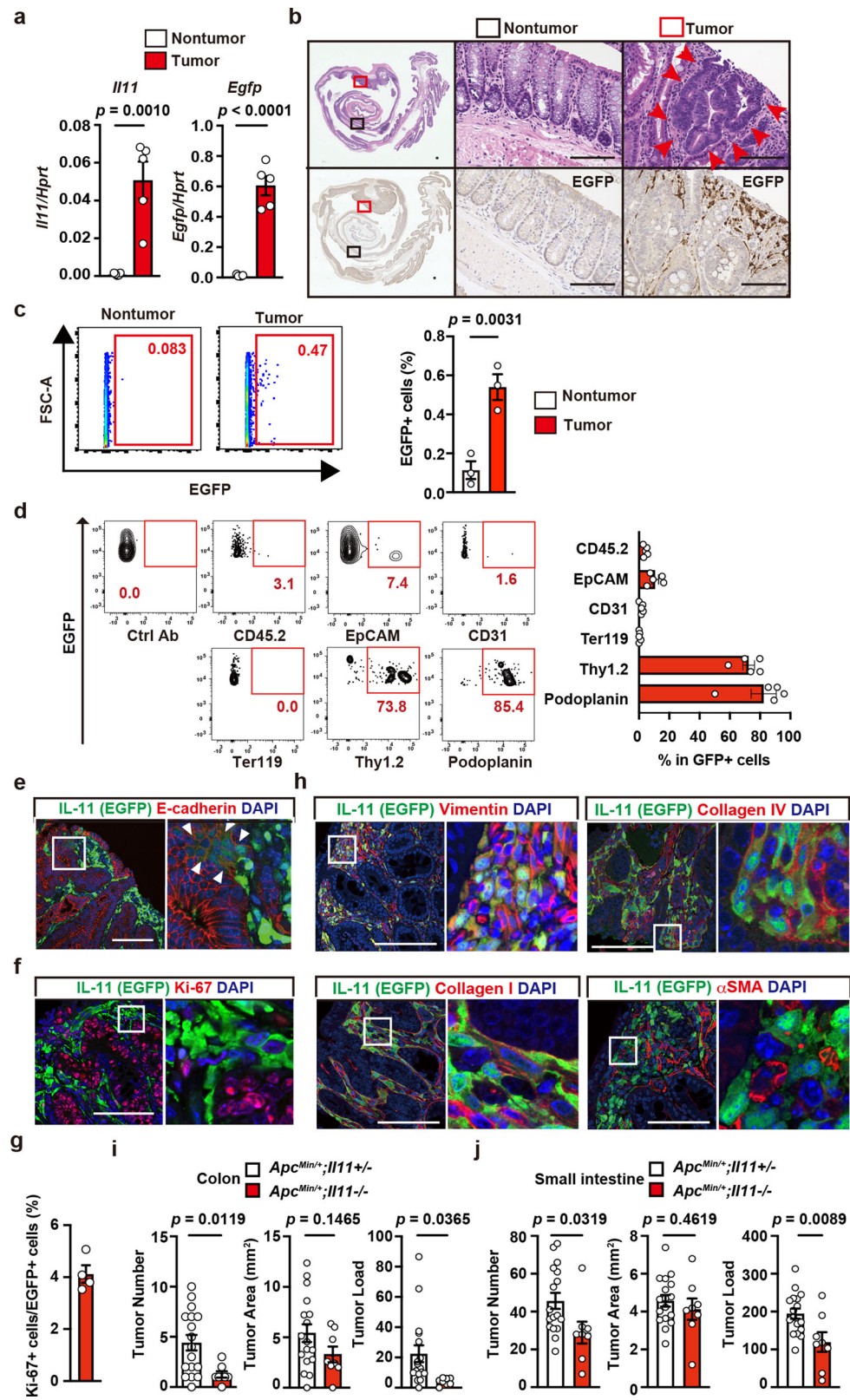

IL-11− cells became IL-11+ cells; alternatively, IL-11+ cells could have been recruited to tumor tissues in the absence of colitis.

To detect the interplay between IL-11+ fibroblasts and colon tumor cells, we tested whether IL-11 induced the activation of colon tumor cells. IL-11 stimulation-induced phosphorylation of STAT3 in *AKTP* tumor organoids and human colon cancer cell lines (Supplementary Fig. 4e). We also found that IL-11 induced

both STAT3 and ERK activation in colonic fibroblasts (Supplementary Fig. 4e). These results suggest that IL-11 induces activation signals in colon tumor cells and fibroblasts.

**IL-11+ fibroblasts appear in the colon in DSS-treated mice and express stromal cell markers.** To determine whether colitis alone

**Fig. 2 IL-11$^+$ Fibroblasts appear in tumor tissues in the absence of colitis. a** Elevated *Il11* and *Egfp* expression in colon tumors from *Apc$^{Min/+}$;Il11-Egfp* reporter mice. Experiments were performed using tumor and nontumor tissues from the colons of *Apc$^{Min/+}$;Il11-Egfp* reporter mice. *Il11* and *Egfp* mRNA expression was determined by qPCR. Results are mean ± SEM ($n = 5$ mice). Results are representative of two independent experiments. **b** Colon tissue sections from *Apc$^{Min/+}$;Il11-Egfp* reporter mice were stained with H&E (upper panels) or anti-GFP antibody (lower panels) ($n = 2$ mice). Middle and right panels, respectively, show enlargements of the black and red boxes from the left panels. Red arrowheads indicate tumor cells. Scale bar, 100 μm. **c, d** The majority of IL-11$^+$ cells express stromal cell markers, and a few IL-11$^+$ cells express epithelial cell markers. Colonic cells were prepared from tumors of *Apc$^{Min/+}$;Il11-Egfp* reporter mice, stained with the indicated antibodies, and analyzed by flow cytometry. Percentages of IL-11$^+$ cells were determined (**c**). Results are mean ± SEM ($n = 3$ mice). A representative dot plot from four independent experiments shows the percentages of EGFP$^+$ cells expressing each marker (**d**). Results are mean ± SEM ($n = 5$ mice). **e–h** Characterization of IL-11$^+$ cells by IHC. Colon tumor sections from *Apc$^{Min/+}$;Il11-Egfp* reporter mice were stained with the indicated antibodies and anti-GFP antibody (**e–h**) ($n = 4$ mice). White arrowheads indicate EGFP$^+$ cells expressing E-cadherin (**e**). The right-hand panels show enlarged images of white boxes from the left-hand panels. Scale bar, 100 μm. Numbers of Ki67$^+$ EGFP$^+$ cells and EGFP$^+$ cells were calculated, and percentages of Ki67$^+$ cells among EGFP$^+$ cells are shown (**g**) ($n = 4$ mice). **i, j** Deletion of the *Il11* gene attenuated tumor development in the colon and small intestine of *Apc$^{Min/+}$* mice. Numbers and areas of tumors in colon and small intestine of 20- to 24-week-old mice of the indicated genotypes were calculated and are plotted. $n = 17$ (*Apc$^{Min/+}$;Il11$^{+/-}$*) or 8 (*Apc$^{Min/+}$;Il11$^{-/-}$*) mice; pooled data from seven independent experiments. Statistical significance was determined by using the two-tailed unpaired Student's *t*-test (**a, c, i, j**). Source data are provided as a Source Data file.

induces IL-11 expression, we treated *Il11-Egfp* reporter mice with only DSS. *Il11* expression in the colon was very low in untreated wild-type mice, but gradually increased and peaked on day 7 after DSS treatment (Fig. 3a). IHC revealed that on day 8 after DSS treatment, many IL-11$^+$ cells appeared in the subepithelial tissues, where intestinal epithelial cells were detached due to severe inflammation (Fig. 3b). We detected the rapid appearance of IL-11$^+$ cells just 1 day after DSS treatment (Fig. 3c). Consistently, small numbers of IL-11$^+$ cells rapidly appeared in the colon on day 1 and they had increased as a percentage of all cells on day 7 after DSS treatment (Fig. 3d, e). For a more detailed comparison of the phenotypes of IL-11$^+$ fibroblasts in tumor tissues and the IL-11$^+$ cells that appeared in colitis, we also analyzed the expression of various cell surface markers in the latter. These IL-11$^+$ cells expressed stromal cell markers, including Thy1.2, podoplanin, CD29, ICAM-1, VCAM-1, and Sca-1, but not CD45.2, CD31, EpCAM, Lyve1, or Ter119 (Fig. 3f). IL-11$^+$ cells associated with colitis were also positive for vimentin, collagen I, and collagen IV, but not αSMA (Fig. 3g), suggesting that these cells were fibroblasts, but not myofibroblasts. Moreover, very few of these IL-11$^+$ fibroblasts incorporated 5-bromo-2′-deoxyuridine (BrdU), a hallmark of cell proliferation, suggesting that most of them did not proliferate in situ (Fig. 3h, i). Together, these findings suggest that colitis alone was sufficient to induce *Il11* expression in fibroblasts.

**IL-11$^+$ fibroblasts express genes associated with cell proliferation and tissue repair.** To further characterize the IL-11$^+$ fibroblasts that appeared in colitis, we used a cell sorter to isolate EGFP$^+$ and EGFP$^-$ fibroblasts from the colons of DSS-treated *Il11-Egfp* reporter mice and compared their gene expression profiles using RNA-seq (Supplementary Fig. 5a). We confirmed that sorted IL-11$^+$ and IL-11$^-$ cells exhibited high and low expression, respectively, of *Il11* and *Egfp* (Fig. 4a). A volcano plot showed significantly elevated expression of some genes in IL-11$^+$ fibroblasts compared with IL-11$^-$ fibroblasts (Fig. 4b). Gene ontology (GO) enrichment analysis revealed that EGFP$^+$ cell RNA was enriched in transcripts of genes associated with cell proliferation, angiogenesis, and wound healing (Fig. 4c). The expression levels of a cytokine, cytokine receptor, and chemokine (*Il11, Il1rl1,* and *Cxcl5*), and of genes associated with organ development (*Hgf* and *Tnfsf11*), were elevated in IL-11$^+$ fibroblasts (Fig. 4d). Moreover, transcript levels of genes associated with CRC susceptibility loci (e.g., *Grem1* and *Bmp4*) and tumor development and invasion (e.g., *Wnt5a, Ereg, Mmp3, Mmp13, Timp1, Saa3, Ptgs2,* and *Acsl4*) were elevated in IL-11$^+$ fibroblasts (Fig. 4d). Of note, expression of *Il11ra1* (encoding IL-11 receptor) was not different between IL-11$^+$ and IL-11$^-$ fibroblasts (Fig. 4e),

suggesting that IL-11 can induce signals in both IL-11$^+$ and IL-11$^-$ fibroblasts.

A unique subset of fibroblasts termed IAFs has recently been reported in UC patients[9]. To investigate the relationship between IL-11$^+$ fibroblasts and IAFs, we examined their gene expression. We found elevated expression of *Il13ra2* and *Tnfsfr11b* (markers of IAFs) in IL-11$^+$ fibroblasts compared with IL-11$^-$ fibroblasts (Fig. 4d). However, gene set enrichment analysis revealed that the global gene expression profiles of IL-11$^+$ fibroblasts and IAFs were different (Fig. 4f), suggesting that IL-11$^+$ fibroblasts and IAFs are not identical, but rather different cell populations.

To test whether fibroblasts with gene signatures similar to that of IL-11$^+$ fibroblasts appear in tumor tissues in mouse and human colon cancers, we examined gene expression profiles of tumor tissues. We found that the expression levels of some genes enriched in IL-11$^+$ fibroblasts, including *Wnt5a, Mmp13, Timp1, Il1rl1, Cxcl5, Saa3, Inhiba, Ascl4,* and *Tnfrsf11b*, were elevated in mouse AOM/DSS-induced colon tumor tissues (Supplementary Fig. 5b). IHC also confirmed that IL-11$^+$ fibroblasts expressed Wnt5a in colons affected by DSS-induced colitis and colon tumor tissues from AOM/DSS-treated mice (Supplementary Fig. 5c). Collectively, the gene expression profiles of IL-11$^+$ fibroblasts present in DSS-induced colitis may at least partly overlap those of colon tumors in mice treated with AOM/DSS.

**The MEK/ERK pathway is involved in *Il11* upregulation in the colon in DSS-treated mice.** Since IL-11$^+$ fibroblasts might promote tumor development, it is crucial to investigate the mechanisms by which acute inflammation or tumor cells induce IL-11 expression. Several studies have reported that TGFβ induces IL-11 production in various cells[20,21,24]. Indeed, TGFβ strongly induced IL-11 production by colonic fibroblasts (Supplementary Fig. 6a). However, expression of *Tgfb1-3* was not elevated in colon from DSS-treated mice (Supplementary Fig. 6b), and neutralizing antibody against TGFβ did not block *Il11* expression in the colon (Supplementary Fig. 6c). These results suggest that TGFβ is not responsible for *Il11* expression in DSS-induced colitis.

We previously reported that oxidative stress induces *Il11* mRNA expression in an ERK/Fra-1-dependent manner[22,23]. Thus, we tested whether DSS treatment-induced oxidative stress in the colon. We observed enhanced oxidative stress in colonic cells after DSS treatment and found that administration of an antioxidant, N-acetyl cysteine (NAC), attenuated this DSS-induced oxidative stress (Fig. 5a). In addition, DSS-induced oxidative stress was ameliorated by administering antibiotics (Abx) (Fig. 5b), suggesting that bacterial infection induces oxidative stress. Consistent with these findings, both an antioxidant, NAC, and Abx blocked the DSS-

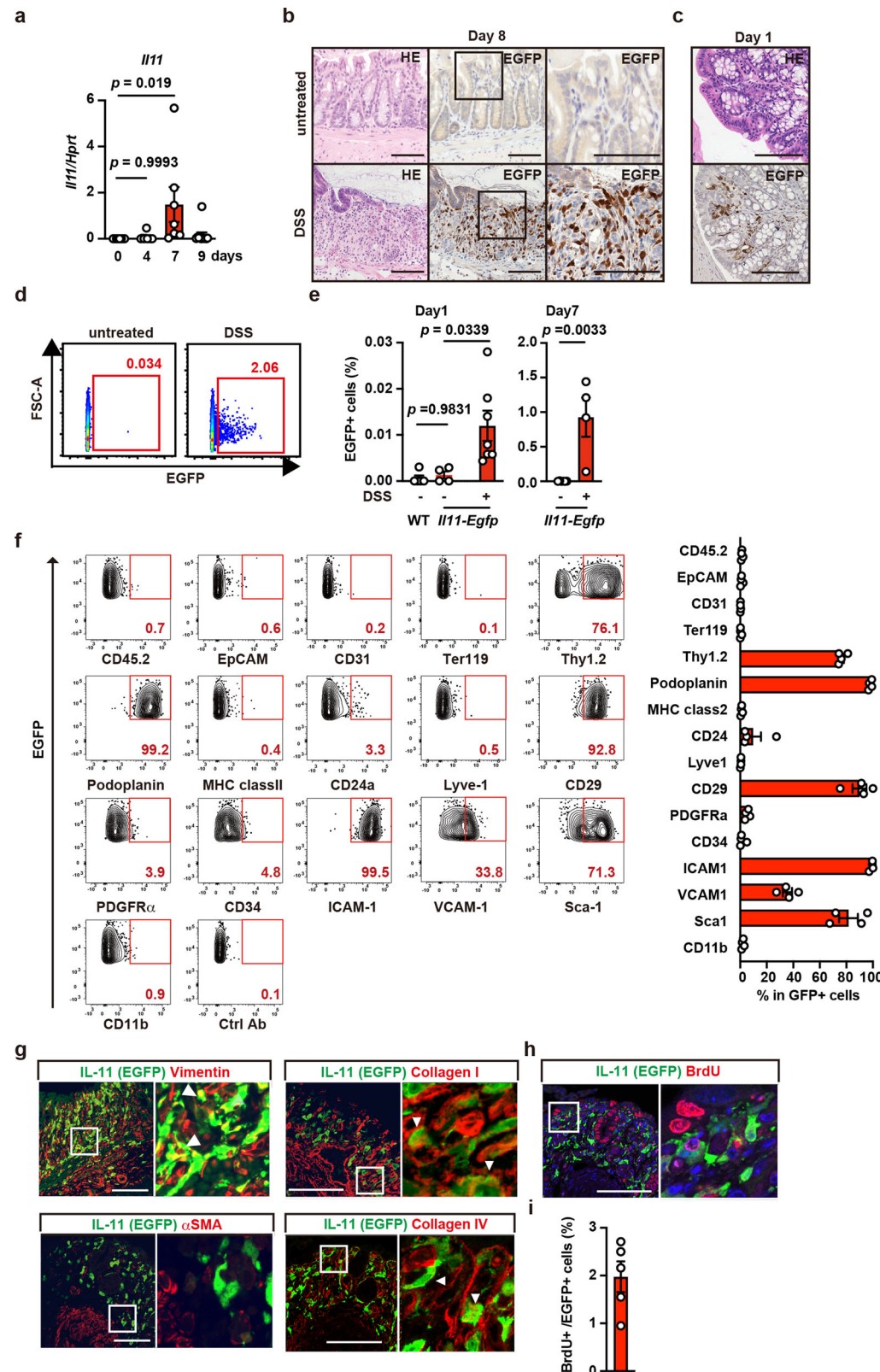

induced upregulation of *Il11* mRNA (Fig. 5c, d). Furthermore, DSS-induced ERK phosphorylation was blocked by the MEK inhibitor trametinib and accompanied by downregulation of *Il11* expression (Fig. 5e–g). We also found that NAC treatment reduced pERK[+] areas (Fig. 5h, i). These results indicate that oxidative stress-dependent ERK activation might contribute to *Il11* expression in the colon.

## Blockade of the MEK/ERK pathway reduces proliferation and induces apoptosis in tumor tissues of AOM/DSS-treated mice.

We next investigated whether *Il11* expression was induced in the tumor microenvironment in a manner similar to that in DSS-induced colitis. Consistent with the results of DSS-induced colitis, the expression of *Tgfb1-3* was not elevated in the colon in AOM/DSS-treated mice and *Apc*[Min/+] mice, and anti-TGFβ antibody

**Fig. 3 IL-11⁺ fibroblasts appear in the colon in DSS-treated mice and express stromal cell markers. a** Kinetics of *Il11* expression in the colon after DSS treatment. WT mice were treated with 1.5% DSS in drinking water for 5 days, followed by a change to regular water. Expression of *Il11* in the colon of mice on the indicated days after DSS treatment was determined using qPCR. Results are mean ± SEM. $n = 8$ (day 0), 9 (day 4), 7 (day 7), or 11 (day 9) mice; pooled data from two independent experiments. **b, c** Appearance of IL-11⁺ cells in submucosal tissues of the colon in *Il11-Egfp* reporter mice on post-DSS treatment day 8 (**b**) or day 1 (**c**). Colon tissue sections from untreated or DSS-treated *Il11-Egfp* reporter mice were H&E stained or immunostained with anti-GFP antibody. $n = 10$ (day 7-8) and 2 (day 1) mice. Right panels show enlargements of the boxes (**c**). Scale bar, 100 μm. **d–f** Characterization of cell surface markers on IL-11⁺ cells. WT and *Il11-Egfp* reporter mice were untreated or treated with DSS as in (**a**), and EGFP⁺ (IL-11⁺) cells were analyzed by flow cytometry. Representative dot plots of untreated or DSS-treated *Il11-Egfp* reporter mice on day 7 (**d**). Percentages of IL-11⁺ cells from the colon before and on day 1 or day 7 after DSS treatment (**e**). Results are mean ± SEM. $n = 5$ (untreated WT on day 1), 4 (untreated *Il11-Egfp* on day 1), 7 (DSS-treated *Il11-Egfp* on day 1), 6 (untreated *Il11-Egfp* on day 7), or 4 (DSS-treated *Il11-Egfp* on day 7) mice. Cells were stained with the indicated antibodies, and marker expression was analyzed on EGFP⁺ cells (**f**). Representative dot plot of four independent experiments ($n = 4$ mice). **g** Representative immunostaining of IL-11⁺ cells. Colon tissue sections were prepared from *Il11-Egfp* reporter mice as in (**b**) and immunostained with the indicated antibodies and anti-GFP antibody ($n = 4$ mice). Results are merged images. Right panels are enlarged images from the boxes. White arrowheads indicate merged cells. **h, i** IL-11⁺ cells do not proliferate in situ. *Il11-Egfp* reporter mice were treated with DSS as in (**a**) and intraperitoneally injected BrdU (40 mg/kg) on day 6. On day 7, colon sections were prepared and stained with anti-GFP and anti-BrdU antibodies ($n = 5$ mice). Scale bars, 100 μm. BrdU⁺ and EGFP⁺ cells were counted, and the percentages of BrdU⁺ cells among EGFP⁺ cells were plotted (**i**). Results are mean ± SEM ($n = 5$ mice). Statistical significance was determined using the one-way ANOVA with Tukey's multiple comparison test (**a**, left graph in **e**) or the unpaired two-tailed Student's *t*-test (right graph in **e**). Source data are provided as a Source Data file.

did not reduce the expression of *Il11* in colon tumor tissues in these mice (Supplementary Fig. 7a–f). Of note, administration of trametinib, but not Abx or NAC, reduced *Il11* expression in CAC after AOM/DSS treatment (Fig. 6a–d). Intriguingly, trametinib reduced the percentage of Ki67⁺ proliferating cells, and conversely, increased the percentage of cleaved caspase 3 (CC3)⁺ cells (a hallmark of apoptosis) (Fig. 6e, f).

**Blockade of the MEK/ERK pathway reduces proliferation and induces apoptosis in tumor tissues of *Apc^{Min/+}* mice.** We observed a similar inhibitory effect of trametinib, but not Abx or NAC, on colon tumors in *Apc^{Min/+}* mice (Fig. 7a–d). Consistent with colon tumors in AOM/DSS-treated mice, trametinib treatment decreased the percentage of Ki67⁺ cells and, conversely, increased the percentage of CC3⁺ cells (Fig. 7e, f). We also found similar effects of trametinib on tumors of the small intestine in *Apc^{Min/+}* mice, although the decrease in *Il11* expression in response to trametinib was not statistically significant (Fig. 7g–i). Together, these results suggest that *Il11* expression was induced in a MEK/ERK-dependent manner, although the upstream signals that induce activation of the MEK/ERK pathway could differ between colitis and tumors.

**IL-11 induces signals in colonic epithelial cells and fibroblasts.** Previous studies report that IL-11 induces signals in colonic epithelial cells and fibroblasts[17,29]. We first examined the *Il11ra1* expression on colonic fibroblasts and normal colon organoids. Since IL-22 induces signals in intestinal epithelial cells[35], we also examined the IL-22 receptor expression on both cell types. Colon organoids showed high expression of *Il22ra*, but not *Il11ra*, while colon fibroblasts exhibited high expression of *Il11ra* and *Il6st* (encoding gp130), but not *Il22ra* (Fig. 8a). Consistently, while IL-11 stimulation-induced STAT3 phosphorylation in colon fibroblasts, and to a lesser extent in colon organoids, IL-22 preferentially induced STAT3 phosphorylation in colon organoids (Fig. 8b). IL-11 stimulation-induced STAT3 phosphorylation in *AKTP* tumor organoids and human tumor cell lines (Supplementary Fig. 4e), prompting us to compare the signals induced by IL-11 in normal colon and tumor organoids. We prepared colon organoids from WT and *Apc^{delta716}* mice carrying a truncated *Apc* gene (*Apc^{delta716}*)[36]. Intriguingly, the expression of *Il11ra* and *Il6st* was higher in *Apc^{delta716}* colon organoids than in WT colon organoids (Fig. 8c). Consistent with the expression levels of *Il11ra*, IL-11 strongly induced STAT3 phosphorylation in

*Apc^{delta716}* colon organoids and, to a lesser extent, WT colon organoids (Fig. 8d). Moreover, IL-11 stimulation-induced ERK phosphorylation in WT and *Apc^{delta716}* colon organoids. Together, *Il11ra* expression in colonic epithelial cells increased along with tumor development, thereby enabling colon tumor cells to respond to IL-11 stimulation efficiently.

To test whether IL-11 stimulation contributes to the upregulation of enriched genes in IL-11⁺ fibroblasts in a paracrine or autocrine manner, we injected an IL-11R agonist manipulated to increase the stability and biological activity of human IL-11 into wild-type mice[22]. As expected, IL-11R agonist injection resulted in the upregulation of many genes in the whole colon from wild-type mice (Fig. 8e). Some of these genes were enriched in IL-11⁺ fibroblasts (Fig. 8f). We performed qPCR to verify the induction of these genes in the whole colon after injection of the IL-11R agonist (Supplementary Fig. 8). These results suggest that IL-11, at least in part, activates IL-11⁺ fibroblasts in an autocrine or paracrine manner, but other signals may induce the majority of enriched genes in IL-11⁺ fibroblasts.

A recent study reported that ColVI⁺ fibroblasts critically contribute to the development of CAC[10]. It also analyzed gene expression profiles in ColVI⁺ fibroblasts stimulated with IL-6 or IL-11. We tested whether IL-11 stimulation-induced gene expression patterns in ColVI⁺ fibroblasts similar to those in IL-11⁺ fibroblasts. However, most enriched genes in IL-11⁺ fibroblasts were different from those in IL-11-stimulated ColVI⁺ fibroblasts (Fig. 8g), suggesting that these two fibroblasts constitute different cell populations or represent different stages of activation.

**IL-11 expression is correlated with the progression of human cancers.** To characterize IL-11⁺ cells in human colon tumor samples, we collected human tumor samples (Supplementary Table 1), including adenoma and early and advanced CRC for IHC with anti-human IL-11 antibody. First, we verified that anti-human IL-11 antibody detected endogenous IL-11, using lysates of the human breast cancer cell line MDA-MB-231 that constitutively produce IL-11[37], in the presence of control or human *Il11* siRNAs (Supplementary Fig. 9a). IL-11⁺ cells were scarce in normal colonic tissues but were numerous in adenomas and in early and advanced CRC tissues (Fig. 9a). In contrast, advanced CRC tissues were not stained with control IgG (Supplementary Fig. 9b), suggesting that anti-IL-11 antibody staining did not represent background tissue signals. Intriguingly, the relative area

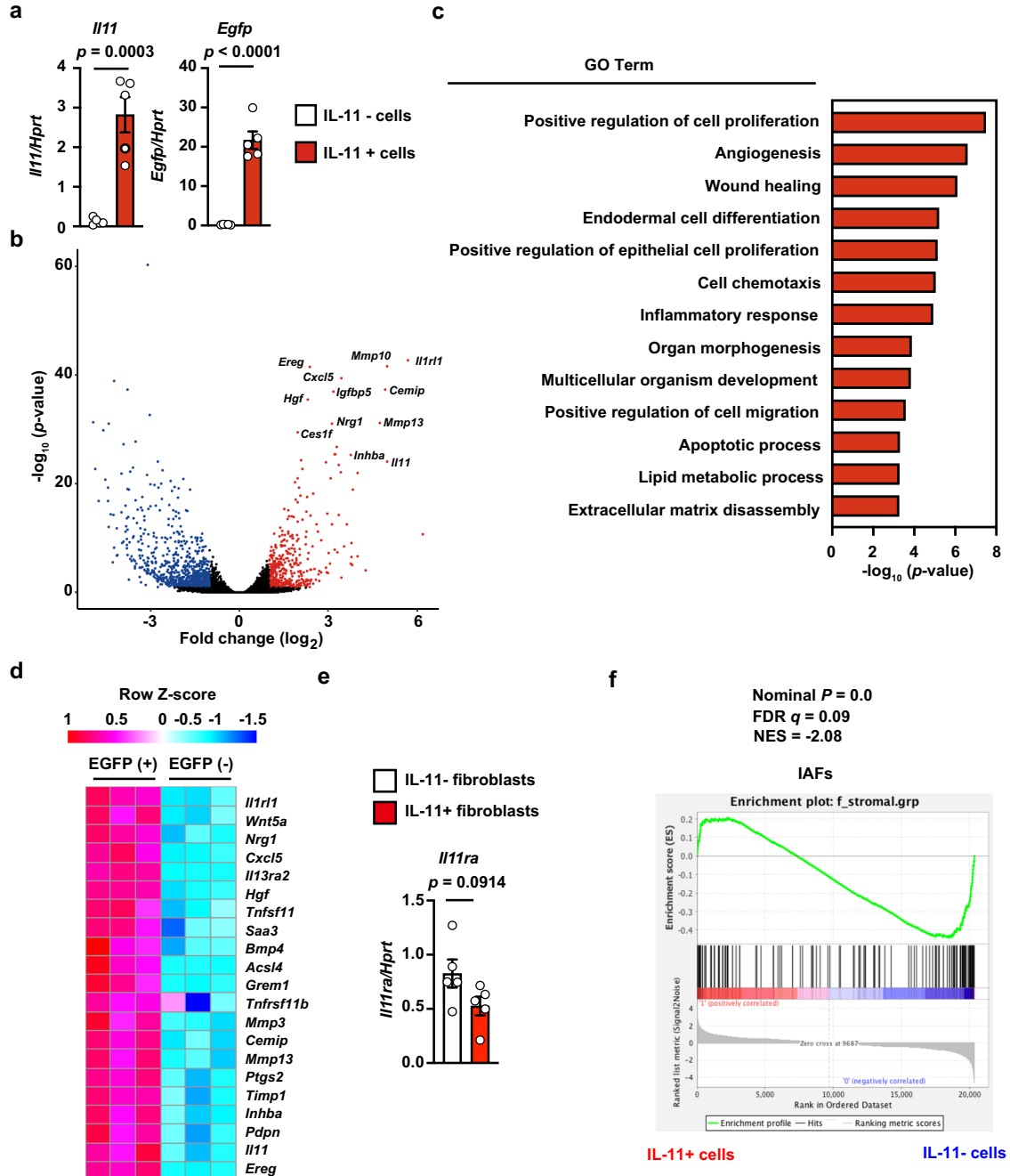

**Fig. 4 IL-11⁺ fibroblasts express genes associated with cell proliferation and tissue repair.** *Il11-Egfp* mice were treated with DSS. On day 7 after DSS treatment, cells were isolated from colons of *Il11-Egfp* reporter mice, and IL-11⁺ cells (EGFP⁺) and IL-11⁻ (EGFP⁻) cells among Ter119⁻ CD45⁻ CD31⁻ EpCAM⁻ cell populations were sorted by flow cytometry. We isolated mRNA from IL-11⁻ and from IL-11⁺ cells and analyzed both using RNA-seq ($n = 3$ mice). **a** Verification of enrichment of IL-11⁺ cells by flow cytometry. Expression of *Il11* and *Egfp* mRNAs was determined by qPCR. Results are mean ± SEM ($n = 5$ mice). **b** Volcano plot of whole genes. The horizontal line indicates genes differentially regulated in IL-11⁺ cells compared with IL-11⁻ cells, shown in log₂. The vertical line indicates *p*-values, shown in −log₁₀. Significantly upregulated and downregulated genes are indicated by red and blue dots, respectively. Several upregulated genes are plotted. **c** Gene ontology (GO) terms that were significantly enriched in IL-11⁺ cells compared with IL-11⁻ cells. GO enrichment analysis of differentially expressed genes were performed using the DAVID Bioinformatics Resources. **d** Heat map of enriched genes in IL-11⁺ fibroblasts compared with IL-11⁻ fibroblasts. Color code for heatmap indicates Z-score of gene expression. **e** Expression of *Il11ra* is not different between IL-11⁺ and IL-11⁻ fibroblasts. Expression of *Il11ra* mRNA was determined by qPCR. Results are mean ± SEM ($n = 5$ mice). **f** Gene set enrichment analysis of IAFs-related genes in IL-11⁺ and IL-11⁻ fibroblasts. FDR false discovery rate, NES normalized enrichment score. Statistical significance was determined by using the two-tailed unpaired Student's *t*-test (**a**, **e**) or two-tailed Kolmogorov-Smirnov test (**f**). The estimation of logFC and adjusted *p*-value were calculated using edgeR package in R (**b**, **c**).

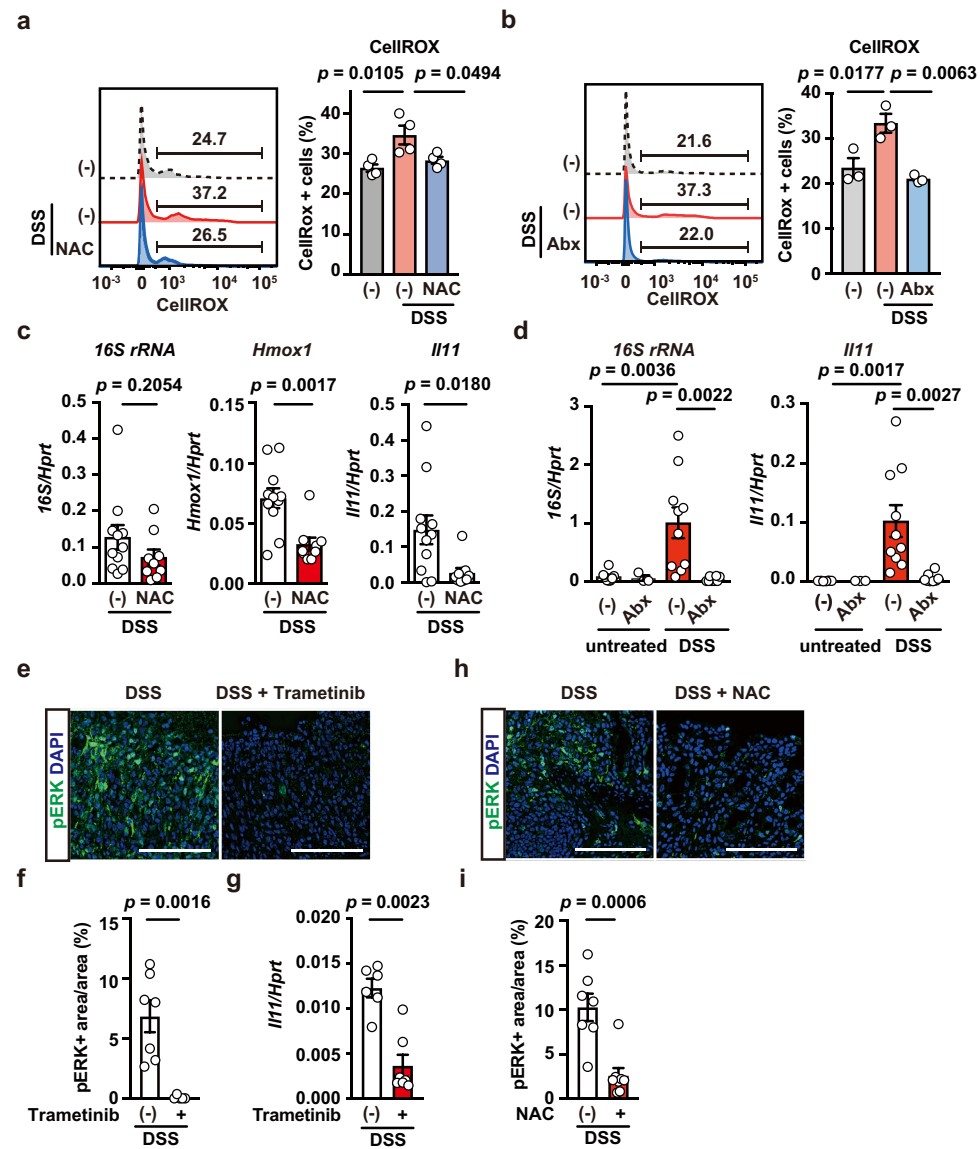

**Fig. 5 The MEK/ERK Pathway is involved in *Il11* upregulation in DSS-induced colitis. a**, **b** Wild-type mice were treated with DSS, without or with NAC (**a**) or Abx (**b**). Colon cells were prepared and stained with CellRox-green, and ROS accumulation was analyzed by flow cytometry. Left panels show representative histograms of ROS levels in colon cells. Right panels show percentages of CellRox-green[+] cells from an individual mouse. Results are mean ± SEM. $n = 4$ (untreated or NAC-treated) or 3 (untreated or Abx-treated) mice. **c** Wild-type mice were treated as in **a**. Colonic expression of *16S rRNA*, *Hmox1*, and *Il11* mRNA was determined using qPCR. Results are mean ± SEM. $n = 11$ (untreated) or 9 (NAC-treated) mice; pooled data from three independent experiments. **d** Abx blocks *Il11* mRNA upregulation in the colon in DSS-treated mice. Wild-type mice were untreated or treated with DSS in the absence or presence of Abx. On day 5 after DSS treatment, qPCR was performed to determine the expression of bacterial *16S rRNA* and *Il11* mRNA in the colon. Results are mean ± SEM. $n = 8$ (untreated), 3 (Abx-treated), 10 (DSS-treated), or 8 (DSS + Abx-treated) mice; pooled data from three independent experiments. **e–g** Trametinib inhibits ERK phosphorylation and *Il11* mRNA expression in the colon in DSS-treated mice. Wild-type mice were treated with DSS in the absence or presence of trametinib injection (at –6 and –30 h), and then colon sections were prepared and stained with anti-pERK antibody (**e**). Scale bars, 100 μm. pERK[+] and DAPI[+] areas were calculated, and the ratios of pERK[+]/DAPI[+] areas (%) were plotted (**f**). Results are mean ± SEM. $n = 7$ (untreated) or 5 (Trametinib-treated) mice; pooled data from two independent experiments. *Il11* mRNA expression was determined using qPCR (**g**). Results are mean ± SEM. $n = 6$ (untreated) or 7 (Trametinib-treated) mice; Pooled data from two independent experiments. **h**, **i** NAC inhibits ERK phosphorylation in the colon in DSS-treated mice. Wild-type mice were treated with DSS in the absence or presence of NAC in the drinking water for 5 days. Colon sections were stained with anti-pERK antibody (**h**). Scale bar, 100 μm. The ratios of pERK[+]/DAPI[+] areas (%) are plotted as in (**f**) (**i**). Results are mean ± SEM. $n = 7$ (untreated) or 8 (NAC-treated) mice; pooled data from two independent experiments. Statistical significance was determined using the unpaired two-tailed Student's *t*-test (**c**, **f**, **i**), two-tailed Mann–Whitney *U* test (**g**), two-way ANOVA with Tukey's multiple comparison test (**d**), or one-way ANOVA with Tukey's multiple comparison test (**a**, **b**). Source data are provided as a Source Data file.

of IL-11[+] cells and the IL-11 signaling intensity were increased in the stroma of advanced colon cancers compared with normal tissues (Fig. 9b, c), suggesting an intimate correlation between IL-11[+] fibroblasts and CRC progression.

**Some genes enriched in IL-11[+] fibroblasts is correlated with reduced recurrence disease-free durations in human CRC.** Assuming that the appearance of IL-11[+] fibroblasts was correlated with CAC development in mice and humans

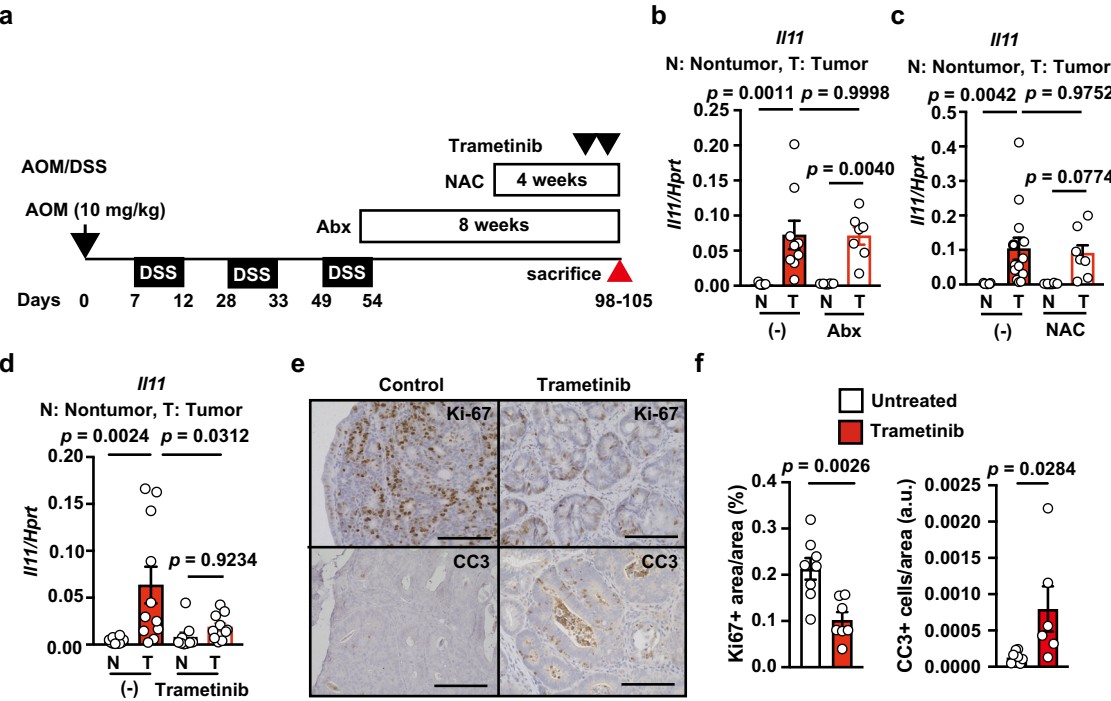

**Fig. 6 Blockade of the MEK/ERK Pathway reduces proliferation and induces apoptosis in Tumor Tissues of AOM/DSS-treated mice. a** Schema of administration of various inhibitors in AOM/DSS-treated mice. **b–f** After induction of colorectal tumors, wild-type mice were not treated or treated with Abx for 8 weeks (**b**), NAC for 4 weeks (**c**), or trametinib at −6 and −30 h (**d–f**) (before sacrifice). On day 98–105 after AOM injection, mRNA was extracted from tumor (T) and non-tumor tissues (N), and *Il11* expression was determined by qPCR (**b–d**). Results are mean ± SEM. $n = 9$ (untreated), 8 (nontumor, Abx-treated), or 7 (nontumor, Abx-treated) mice; pooled data from two independent experiments (**b**). $n = 13$ (untreated) or 8 (NAC-treated) mice; pooled data from two independent experiments (**c**). $n = 10$ (nontumor, untreated), 11 (tumor, untreated), 9 (nontumor, trametinib-treated), or 10 (nontumor, trametinib-treated) mice; pooled data from four independent experiments (**d**). Colon tumor sections were stained with anti-Ki67 or anti-CC3 antibodies (**e**). Ki67+ area and total area were calculated and the percentages of Ki67+ area per area are expressed (**f**). $n = 8$ (untreated) or 7 (trametinib-treated) mice. Numbers of CC3+ cells were counted and the total area was calculated, and CC3+ cells per area are expressed as arbitrary units (a.u.) (**f**). $n = 8$ (untreated) or 6 (trametinib-treated) mice. Statistical significance was determined using the unpaired two-tailed Student's *t*-test (**f**) and two-way ANOVA with Tukey's multiple comparison test (**b–d**). Source data are provided as a Source Data file.

(Figs. 1, 2 and 9), we hypothesized that the genes enriched in IL-11+ fibroblasts might also be elevated in human CRC. We focused on genes with over twofold greater expression in IL-11+ fibroblasts compared with IL-11− fibroblasts (Fig. 4), and extracted 22 genes matching our criteria from the human cancer databases (GSE33133 and GSE35602). Intriguingly, most but not all subsets of genes elevated in IL-11+ fibroblasts were significantly upregulated in colon cancer tissues relative to normal mucosa (Fig. 10a). Moreover, analysis using the human CRC database (GSE35602) revealed that the expression of some genes with high expression in IL-11+ fibroblasts was also elevated in tumor stromal compartments compared with tumor epithelial compartments (Supplementary Fig. 10).

Since many of the genes enriched in IL-11+ fibroblasts were also upregulated in tumor tissues (Fig. 10a and Supplementary Fig. 5a), we hypothesized that the genes with elevated expression in IL-11+ fibroblasts might critically affect the prognosis of cancer patients. Using human CRC datasets (GSE17536, GSE17537, and GSE14333), we first divided patients into two clusters based on the expression of *IL11* or signature genes of IAFs (*IL11/IL24/IL13RA2/TNFRSR11B*) by the hierarchical clustering method. However, neither clusters did not result in any differences in recurrence-free survival durations (Fig. 10b, c). We next divided patients based on the expression patterns of genes preferentially expressed in IL-11+ fibroblasts (twofold higher than IL-11− fibroblasts), which we referred to as the IL-11+ fibroblast signature (IL11FS). Intriguingly, recurrence-free survival durations were significantly decreased in cluster 6

compared with cluster 5 (Fig. 10d). Given that the expression of a set of genes including *HGF*, *IL13RA2*, *PTGS2*, and *TNFSF11* in cluster 6 was higher than in cluster 5 (Supplementary Data 1), these upregulated genes may be associated with poor CRC prognosis.

## Discussion

In this study, we generated *Il11-Egfp* reporter mice to characterize IL-11+ cells in different murine tumor and colitis models. We found that the IL-11+ cells in tumor tissues were mostly fibroblasts and a few epithelial cells and that deletion of *Il11ra* or *Il11* attenuated CAC development in mice. BM transfer experiments revealed that IL-11+ fibroblasts critically contributed to tumor progression and that IL-11+ fibroblasts were not derived from BM. IL-11 efficiently activated IL-11+ colonic fibroblasts and colon tumor organoids. Transcriptome analysis showed that IL-11+ fibroblasts expressed genes associated with tissue repair and cell proliferation. Thus, IL-11+ fibroblasts produce several growth factors that induce the proliferation of nearby tumor cells. Moreover, the expression levels of some of the genes enriched in IL-11+ fibroblasts were elevated in stromal tissues in human CRC, and this high expression was associated with reduced recurrence-free survival duration in human CRC. Thus, our results suggest that IL-11+ fibroblasts activate both colon tumor epithelial cells and colon fibroblasts through IL-11 secretion, thereby contributing to tumor progression.

The newly developed *Il11-Egfp* reporter mice enabled us to characterize the IL-11+ fibroblasts that appeared in mouse

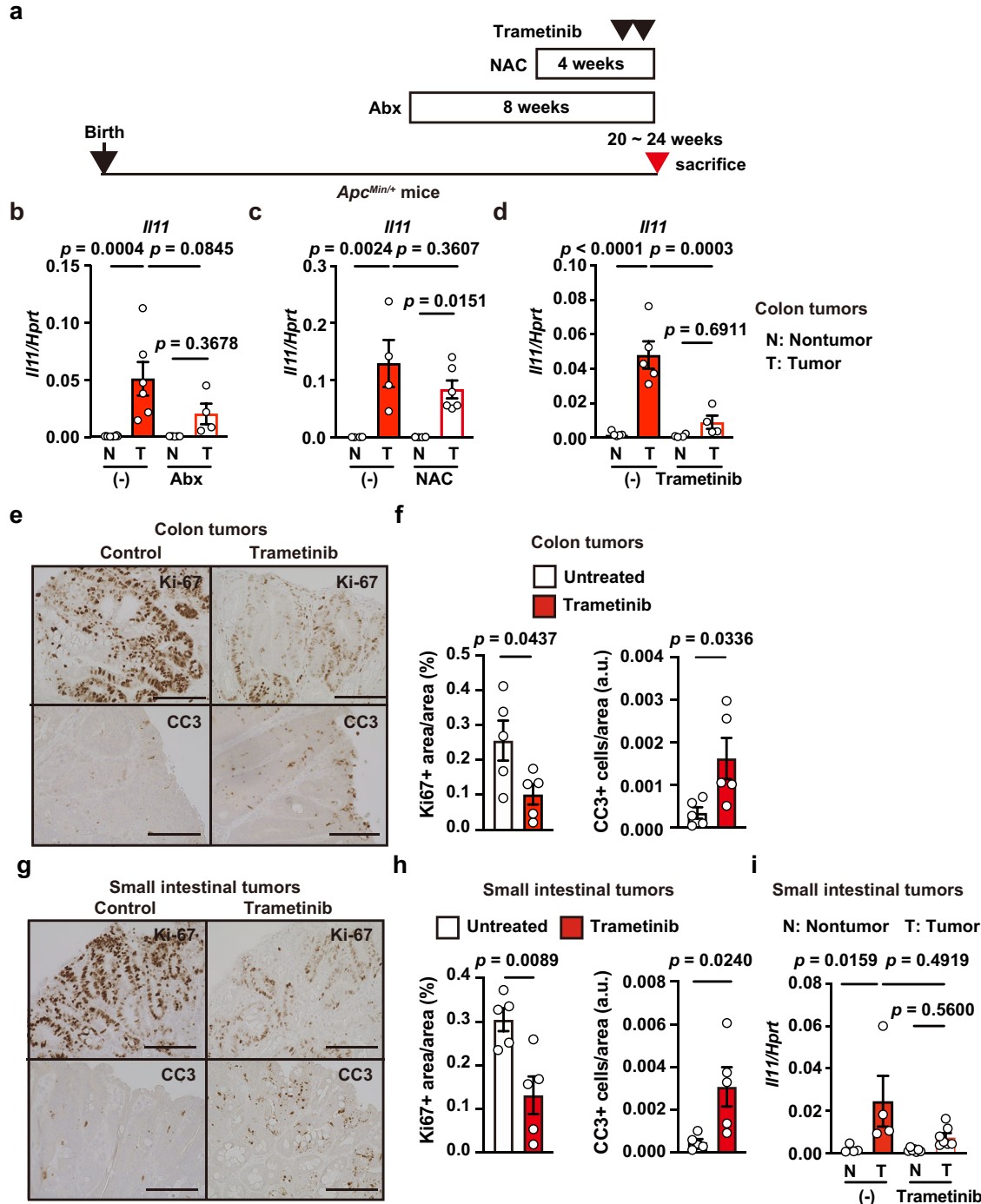

**Fig. 7 Blockade of the MEK/ERK Pathway reduces proliferation and induces apoptosis in Tumor Tissues of $Apc^{Min/+}$ mice. a** Schema of administration of various inhibitors to $Apc^{Min/+}$ mice. **b–i** $Apc^{Min/+}$ mice were not treated or treated with Abx for 8 weeks (**b**), NAC for 4 weeks (**c**), or trametinib at –6 and –30 h (**d–i**), then sacrificed 20–24 weeks after birth. **b–d** *Il11* expression in tumors and nontumor tissues in the colon was determined using qPCR. Results are mean ± SEM. $n = 8$ (nontumor, untreated), 6 (tumor, untreated), 7 (nontumor, Abx-treated) or 4 (tumors, Abx-treated) mice; pooled data from three independent experiments. $n = 4$ (untreated) or 6 (NAC-treated) mice; pooled data from two independent experiments. $n = 5$ (untreated) or 4 (trametinib-treated) mice; pooled data from three independent experiments. Colon (**e**) or small intestine (**g**) tumor tissue sections were stained with anti-Ki67 or anti-CC3 antibodies, and analyzed and are expressed as in Fig. 6f, g (**f**, **h**). Results are mean ± SEM ($n = 5$ mice). *Il11* expression in tumors and nontumor tissues in the small intestine was determined using qPCR (**i**). Results are mean ± SEM. $n = 4$ (untreated) or 7 (trametinib-treated) mice; pooled data from three independent experiments. Statistical significance was determined using the two-way ANOVA with Tukey's multiple comparison test (**b–d**, **i**), or two-tailed unpaired Student's *t*-test (**f**, **h**). Source data are provided as a Source Data file.

models of colitis and CAC. In the colitis model, IL-11+ fibroblasts exclusively expressed stromal cell markers, such as Thy1 and podoplanin. A recent single-cell RNA-seq analysis of colon biopsies from healthy individuals and UC patients revealed that colon samples from UC patients exhibited a unique subset of fibroblasts, termed IAFs[9]. IAFs express *IL11*, *IL24*, *IL13RA2*, and *TNFSFR11B*, which were also elevated in IL-11+ fibroblasts. However, large numbers of genes enriched in IL-11+ fibroblasts were not

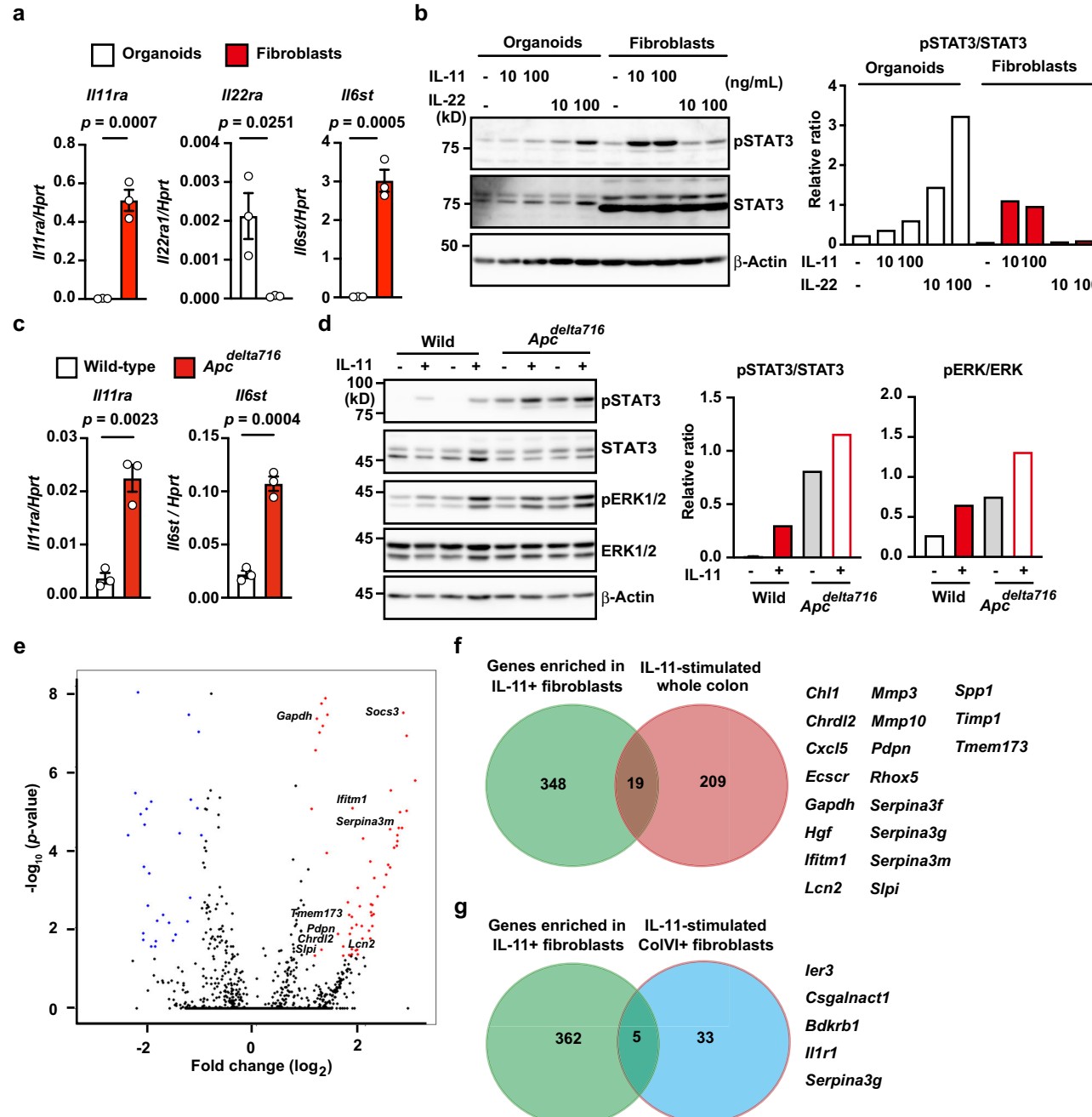

**Fig. 8 IL-11 induces signals to colonic epithelial cells and fibroblasts. a** Colonic epithelial organoids and fibroblasts were established from three independent wild-type mice as described in the methods. Expressions of the indicated genes were determined using qPCR. Results are mean ± SEM ($n = 3$ mice). Results are representative of two independent experiments. **b** Colonic epithelial organoids and fibroblasts were unstimulated or stimulated for 30 min with IL-11 (10 or 100 ng/mL) or IL-22 (10 or 100 ng/mL). Total STAT3 and phosphorylated STAT3 (pSTAT3) were analyzed by Western blotting. STAT3 and pSTAT3 signaling intensities were calculated by Fiji, and the relative ratio of pSTAT3/STAT3 is shown. Results are representative of two independent experiments (**a**, **b**). **c**, **d** Relative expression of *Il11ra1* and *Il6st* in colon organoids. Colon organoids were established from WT and *Apc^delta716^* mice carrying a truncated *Apc* gene. Expression of the indicated genes in WT and *Apc^delta716^* organoids was determined using qPCR (**c**). Results are mean ± SEM ($n = 3$ independently established organoids from different mice). Colon organoids were not stimulated or stimulated with IL-11 (100 ng/mL) for 30 min (**d**). Total STAT3, phosphorylated STAT3 (pSTAT3), total ERK, and phosphorylated ERK (pERK) were analyzed and described as in (**b**). Results are representative of two independent experiments. **e**, **f** Administration of IL-11R agonist induces expression of the genes expressed in IL-11$^+$ cells. We injected 8-week-old wild-type mice with 10 μg IL-11R agonist. At 3 h after injection, mRNA was isolated from the whole colon, and gene expression was analyzed using microarray analysis as described in "Methods" ($n = 2$ for untreated samples; $n = 3$ for injected samples). Volcano plot of whole genes (**e**). The horizontal line indicates differentially regulated genes in the whole colon after IL-11R agonist injection compared with the whole colon before injection, shown in log2. The vertical line indicates $P$ values, shown in −log10. Red dots indicate significantly upregulated genes. Venn diagram of genes enriched in FACS-sorted IL-11$^+$ fibroblasts and genes with elevated expression in the whole colon after IL-11R agonist treatment compared with an untreated colon (**f**). Nineteen overlapping genes are shown. **g** Venn diagram of genes enriched in FACS-sorted IL-11$^+$ fibroblasts and genes induced by IL-11 stimulation in ColVI $^+$ fibroblasts. Five overlapping genes are shown. Statistical significance was determined using the two-tailed unpaired Student's *t*-test (**a**, **c**). *p*-values were calculated by two-tailed local-pooled-error test (**e**). Source data are provided as a Source Data file.

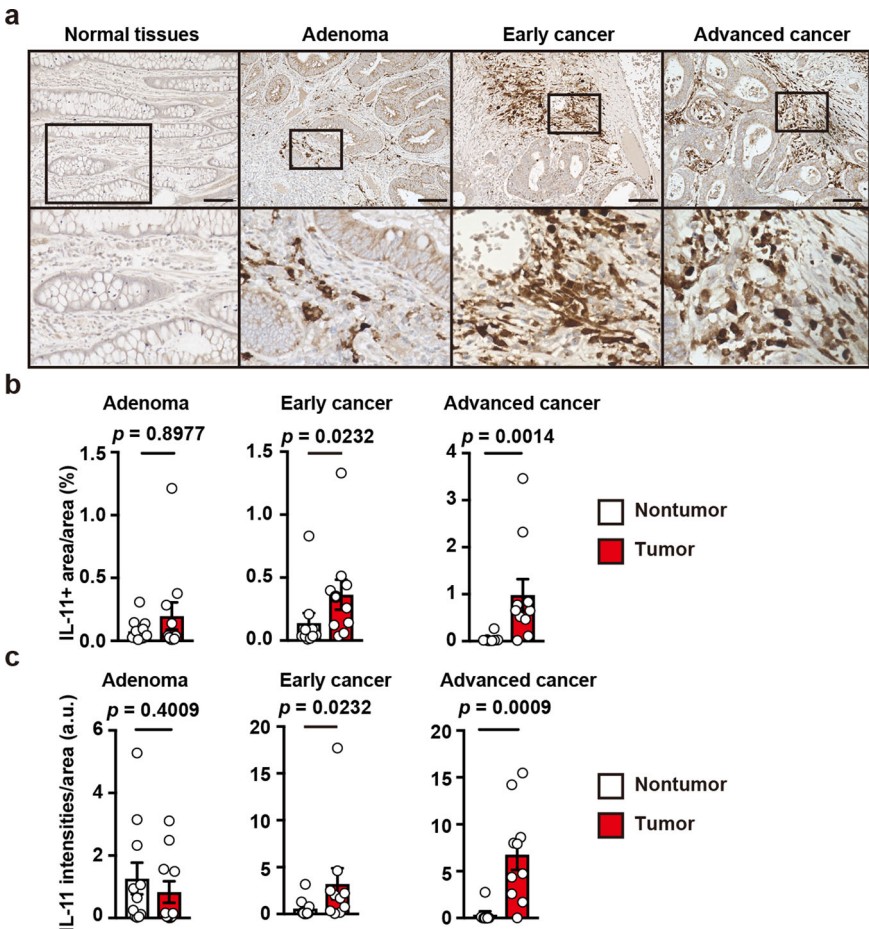

**Fig. 9 IL-11 expression is correlated with progression of human cancers. a–c** Adenomas ($n = 11$) and early ($n = 10$) and advanced ($n = 10$) colorectal cancers were stained with anti-IL-11 antibody (**a**). Representative staining of the respective tumors and adjacent normal tissues. Scale bars, 100 μm. IL-11+ area and total area were calculated and the percentages of IL-11+ area per area are plotted (**b**). The intensities of IL-11+ signals and total area were calculated and intensities of IL-11+ per area are expressed as arbitrary units (a.u.) (**c**). Results are mean ± SEM. $n = 11$ (adenoma), 10 (early cancer), 8 (nontumor, advanced cancer), or 10 (tumor, advanced cancer). Scale bars, 100 μm. Statistical significance was determined using the two-sided Mann–Whitney $U$ test (**b**, **c**). Source data are provided as a Source Data file.

expressed in IAFs, suggesting that IL-11+ fibroblasts and IAFs might be different populations.

In contrast to the results in DSS-induced colitis, we detected a few IL-11+ cells expressing epithelial cell markers, such as EpCAM and E-cadherin, in tumor tissues. Previous studies have reported that IL-11+ cells are derived from stromal or epithelial cells[17,24]. Our present results revealed that IL-11+, EpCAM+, and E-cadherin+ epithelial cells might be tumor cells themselves. A previous study has also reported that tumor cells, such as breast cancer or colon cancer cells, produce IL-11[38]. Future scRNA-seq analysis will further characterize heterogeneous populations of IL-11+ cells in the tumor microenvironment.

Elucidation of the mechanisms underlying IL-11 production by fibroblasts in the tumor microenvironment may be crucial for understanding how tumor cells instruct stromal cells. Previous studies have demonstrated that TGFβ induces IL-11 upregulation in various cell types[20,21]. Indeed, we found that TGFβ stimulation-induced IL-11 production by colonic fibroblasts. Moreover, in murine xenograft models, human tumor cell lines that ectopically express human TGFβ1 can elicit IL-11+ CAFs[24]. However, in our present study, anti-TGFβ antibody treatment did not downregulate *Il11* expression in colonic adenomas of *Apc^min/+* mice or in AOM/DSS-induced CAC. Overall, although TGFβ per se was able to induce IL-11 production, TGFβ may not be involved in the induction of IL-11+ fibroblasts in the present

tumor models, including AOM/DSS-induced CAC and colon adenoma in *Apc^Min/+* mice. On the other hand, blockade of the MEK/ERK pathway by trametinib attenuated *Il11* expression in mouse colon in DSS-induced colitis, AOM/DSS-induced CAC, and colon adenoma in *Apc^Min/+* mice. More importantly, blockade of the MEK/ERK pathway reduced proliferation and increased apoptosis of tumor cells in these tumor models. Given that IL-11 released from IL-11+ fibroblasts activated the MEK/ERK pathway in tumor cells, IL-11 and the MEK/ERK pathway may constitute the feed-forward loop via cancer-associated IL-11+ fibroblasts and tumor cells. It is reasonable to speculate that one of the mechanisms how trametinib inhibited proliferation and induced apoptosis of tumor cells would be caused by the shutoff of this feed-forward loop.

Previous studies have shown that IL-11 and IL-22 induce the proliferation and cell survival of colonic epithelial cells through STAT3 activation[15,35]. Indeed, under our experimental conditions in colonic epithelial organoids, IL-22 induced strong STAT3 phosphorylation, whereas IL-11 induced only weak STAT3 phosphorylation. In sharp contrast, IL-11 induced robust phosphorylation of STAT3 in colonic fibroblasts. Intriguingly, the expression of *Il11ra* in colon tumor organoids was higher than in colon epithelial organoids, suggesting that colon tumor cells can efficiently respond to IL-11 stimulation as the tumor develops. qPCR analysis revealed that the expression of

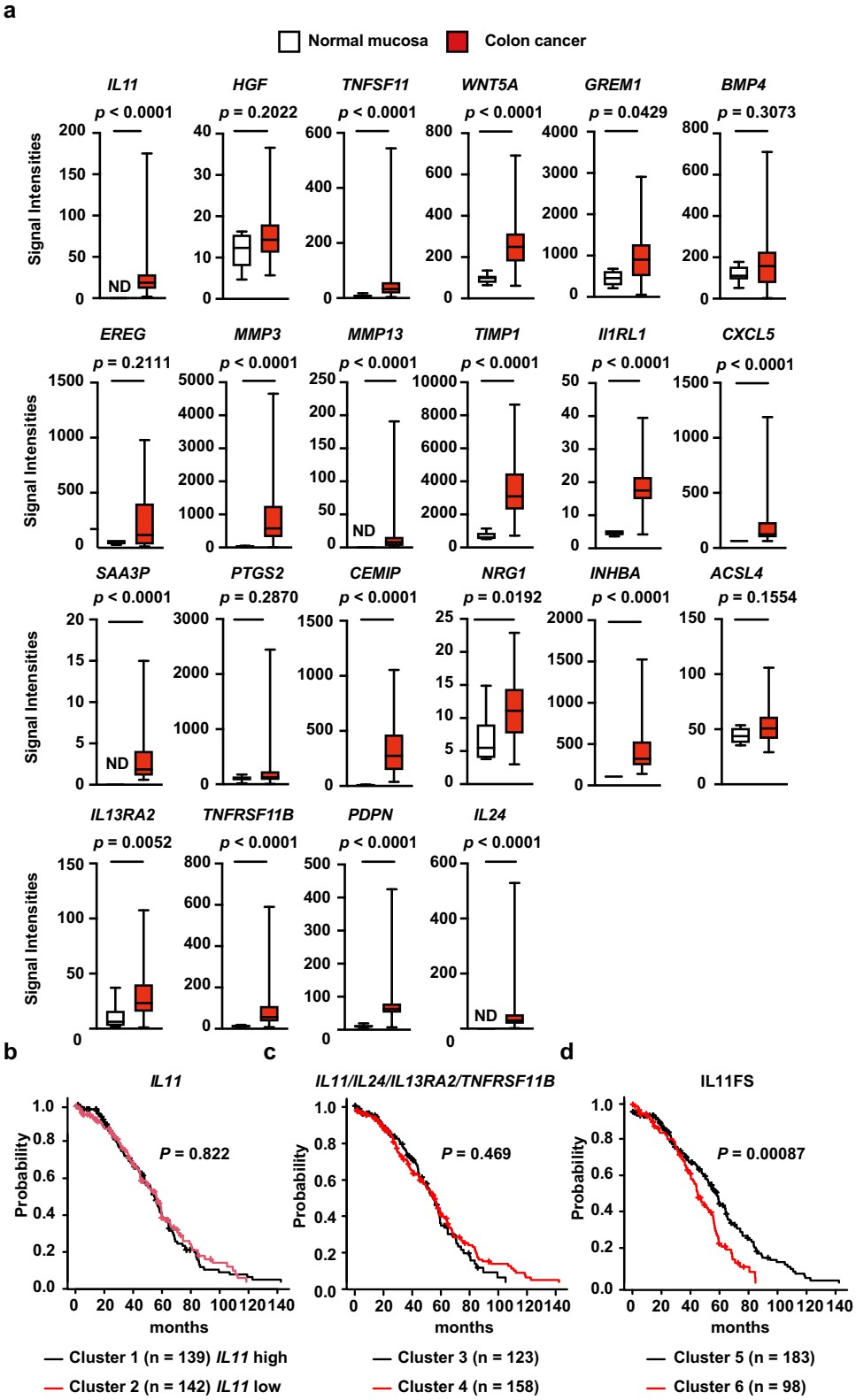

*Il11ra* did not differ between IL-11$^+$ and IL-11$^-$ fibroblasts. Thus, it appears that colonic fibroblasts produced IL-11 in response to factors from tumor cells, which, in turn, activated and induced STAT3 phosphorylation in both IL-11$^+$ and IL-11$^-$ colonic fibroblasts in an autocrine or paracrine manner. This feed-forward loop between fibroblasts and tumor cells via secreted IL-11 from IL-11$^+$ fibroblasts might contribute to tumor progression. Although ColVI$^+$ fibroblasts and IL-11$^+$ fibroblasts appear to promote tumor progression[10], global gene expression profiles in these two populations were different. It is currently unclear whether these two-cell populations are derived from different cell lineages or represent different activation stages. Further study will be required to address this issue.

**Fig. 10 Some Genes Enriched in IL-11$^+$ Fibroblasts is correlated with reduced recurrence disease-free durations in human CRC. a** Expression levels of genes enriched in IL-11$^+$ fibroblasts are elevated in human colon cancer tissues. From the data set (GSE33113), we retrieved the expression levels of the genes enriched in IL-11$^+$ fibroblasts described in Fig. 4. The signaling intensities of each gene in normal mucosa ($n = 6$) and colon cancer tissues ($n = 90$) are shown by box-and-whisker plots. Boxes and whiskers show the 25th–75th percentile with the median and the minimum–maximum, respectively. **b–d** Upregulation of a set of genes enriched in IL-11$^+$ fibroblasts (IL11FS) is correlated with reduced recurrence-free survival durations in colon cancer patients. GSE17536 and GSE17537 were obtained from the Gene Expression Omnibus. Based on the expression patterns of *IL11*, signature genes of IAFs (*IL11/IL24/IL13A2/TNFRSF11B*), or IL11FS, we classified patients into two respective clusters using the hierarchical clustering method. The recurrence-free durations of each group was calculated using the Kaplan–Meier method. Statistical significance was determined using two-sided Mann–Whitney $U$ test (**a**) or two-sided Mantel-Cox Log-rank test (**b–d**). Source data are provided as a Source Data file.

CAC development was attenuated in *Il11ra1*$^{-/-}$ and *Il11*$^{-/-}$ mice treated with AOM/DSS, but deletion of these genes did not dramatically affect tumor development compared with those tumors in a previous study[17]. The reason for these discrepancies is unknown. However, the genetic background of the mice (C57/BL6 vs. 129/C57/BL6 mixed background) might affect the tumor cells' dependence on IL-11 signaling. On the other hand, our transcriptome analysis revealed that IL-11$^+$ fibroblasts expressed various genes associated with tissue repair and cell proliferation. Indeed, we found that high expression of a set of genes in IL11FS, but not *IL11* or signature genes of IAFs (*IL11/IL24/IL13A2/TNFRSF11B*), was associated with reduced recurrence-free survival duration in CRC patients. A previous study reported that high expression of IL11-responsive signature (IL11RS) genes induced in a human colon cancer cell line stimulated with IL-11 is correlated with reduced recurrence-free survival duration in CRC patients[24]. Given that IL11FS and IL11RS might be different, IL-11 activates both fibroblasts and tumor cells, thereby contributing to CRC progression. Thus, IL-11$^+$ fibroblasts are potential targets in the treatment of CRC.

## Methods

**Reagents**. Azoxymethane (A5486, Sigma-Aldrich), murine IL-11 (418-ML-025, R&D), human IL-11 (418-ML-025, R&D), murine IL-22 (576202, BioLegend), NAC (11568-92, Nakalai), mouse TGFβ1 (763102, BioLegend), and trametinib (T-8123, LC Laboratories) were purchased from the indicated sources. The following antibodies used in this study were obtained from the indicated sources: anti-β-actin (622102, BioLegend, 1:1000), anti-BrdU (BU1/75, BIO-RAD, 1:100), anti-CD45 (13917, CST, 1:200), anti-cleaved caspase-3 (Asp175) (9664, CST, 1:2000), anti-collagen I (ab34710, Abcam, 1:500), anti-collagen IV (ab6586, Abcam, 1:500), anti-E-cadherin (560062, BD Biosciences, 1:200), anti-phospho-ERK (4370, CST, 1:800 for IHC, 1:1000 for Western blotting), anti-ERK (9102, CST, 1:1000), anti-GFP (GFP-Go-Af1480 or GFP-Rb-Af2020, Frontier Institute, 1:200), anti-human IL-11 (LS-C408373, LSBio, 1:100), anti-mouse IL-11 (in house, final 750 ng/ml)[32], anti-Ki67 (ab16667, Abcam, 1:100), anti-podoplanin (127403, BioLegend, 1:100), anti-α-SMA (ab5694, Abcam, 1:200), anti-phospho-STAT3 (9145, CST, 1:1000), anti-STAT3 (SC-482, Santa Cruz, 1:1000), anti-tubulin (T5168, Sigma-Aldrich, 1:50,000), anti-vimentin (9856, CST, 1:200), and anti-Wnt5a/b (C27E8, CST, 1:50). Anti-horseradish peroxidase (HRP)-conjugated anti-rabbit IgG (NA934, 1:5000) and anti-mouse IgG (NA931, 1:5000) antibodies were from GE Healthcare. Alexa Fluor 488-conjugated donkey anti-rabbit IgG (A21206, 1:500), Alexa Fluor 594-conjugated donkey anti-rabbit IgG (A21207, 1:500), Alexa Fluor Plus 594-conjugated donkey anti-rabbit IgG (A32754, 1:500), Alexa Fluor 647-conjugated donkey anti-rabbit IgG (A31573, 1:500), and Alexa Fluor 488-conjugated donkey anti-goat IgG (A11055, 1:500) antibodies, and Alexa Fluor 594-conjugated streptavidin (S11227, 1:500) were purchased from Invitrogen. ImmPRESS® VR anti-rabbit IgG HRP polymer detection kit (MP-6401, Vector Laboratories, 1:1) and biotinylated goat anti-Rabbit Immunoglobulins (E0432, Dako, 1:200) were purchased from the indicated sources.

Unless otherwise indicated, the following antibodies used for flow cytometry were obtained from TONBO Biosciences; anti-CD11b (20-0112, clone M1/70, 1:100), anti-CD16/CD32-mAb (2.4G2) (made in house, final 0.01 mg/ml), anti-CD24a (101813, BioLegend, clone M1/69, 1:100), anti-CD29 (102207, BioLegend, HMβ1-1, 1:100), anti-CD31 (17-0311-82, eBioscience, clone 390, 1:250), anti-CD34 (e13-0341-81, eBioscience, clone RAM34, 1:100), anti-CD45.1 (35-0453, clone A20, 1:100), anti-CD45.2 (20-0454, clone 104, 1:100), anti-EpCAM (118214, BioLegend, clone G8.8, 1:250), anti-ICAM-1 (561605, BD Biosciences, clone 3E2, 1:100), anti-Lyve1 (50-0443-80, eBioscience, clone ALY7, 1:100), anti-MHC Class II (130-102-139, Miltenyi Biotec, clone M5/114.15.2, 1:250), anti-PDGFRα (17-1401-81, eBioscience, clone APA5, 1:100), anti-podoplanin (127414, BioLegend, clone 8.1.1, 1:250), anti-Sca-1 (122512, BioLegend, clone E13161.7, 1:250), anti-TER-119

(116212, BioLegend, clone TER-119, 1:100), anti-Thy1.2 (20-0903, clone 30-H12, 1:250), anti-VCAM-1 (105718, BioLegend, clone 429, 1:100), and Streptavidin APC (17-4317-82, eBioscience, 1:500).

The hybridoma cell lines, 1D11[39] and 2.4G2 that produce neutralizing antibody against all TGFβ isoforms (β1, β2, and β3) and antibody against mouse CD16/32, respectively, were purchased from ATCC. These antibodies were produced in house. Control mouse IgG (I5381) was purchased from Sigma-Aldrich.

All antibodies that came from commercial vendors are validated by the manufactures for the species and assay in our study. Validation data are available on the manufacture's websites. All antibodies were initially tested against unstained controls and dilution series were performed to optimize.

**Cell lines**. Human colon cancer cell lines, HT29, and HCT116 cells were obtained from ATCC. A human breast cancer cell line, MDA-MB-231 cells were provided by T. Sakamoto. These cell lines were cultured in DMEM supplemented with 10% FBS. Cultrex® R-spondin1 (Rspo1) Cells were obtained from Trevigen.

**Mice**. *Il11*$^{-/-}$ mice (generated in our lab) and *Il11ra1*$^{-/-}$ mice (provided by L. Robb) were previously described[32,40]. *Apc*$^{min/+}$ (002020) were purchased from Jackson Lab. C57BL/6 (CD45.2$^+$) and C57BL/6-SJL (CD45.1$^+$) mice were purchased from Japan-SLC.

Mice were housed at 23 ± 2 °C, a humidity of 55 ± 5%, and a 12 hr dark/light cycle. Mice in different cages or derived from different sources were cohoused for 2 weeks to normalize the microbiota composition before experiments. All animals were housed and maintained under specific pathogen-free conditions in the animal facility at Juntendo University School of Medicine or Toho University School of Medicine. All experiments were performed according to the guidelines approved by the Institutional Animal Experiment Committee of Juntendo University School of Medicine or Toho University School of Medicine (19-51-414, 19-51-411).

Detailed information about mice, including genotypes, ages, and genders used in this study was described in Supplementary Table 2.

**Generation of *Il11-Egfp* reporter mice**. The *Egfp* reporter gene was introduced into the BAC clone (RP23-285B12) by two-step Red/ET recombineering technology according to the manufacturer's protocol (Gene Bridges). In the first step, a *rpsL-neo* cassette included in the kit was amplified by PCR using a primer set (*Il11*-ET1-F2: ACTCCCTCAGACCCAGAGTTTGGCCTGATTTCTCCCTTCTG TCCACAGGTGGCCTGGTGATGATGGCGGGATCG and *Il11*-ET1-R2: ACGA CTCTATCTGGCCAGAGGCTCAGCACCACCAGGACCAGGCGACAAACTCA GAAGAACTCGTCAAGAAGGCG) and inserted into the target region of the BAC clone. In a second step, the *rpsL-neo* cassette in the modified BAC clone was replaced with *Egfp-polyA* cassette, amplified from *Egfp*-expression vector (pAWZ) using a primer set (*Il11*-ET2-F2: ACTCCCTCAGACCCAGAGTTTGGCCTGATT TCTCCCTTCGTGTCCACAGGTATGGTGAGCAAGGGCGAG and *Il11*-ET2-R2: ACGACTCTATCTGGCCAGAGGCTCAGCACCACCAGGACCAGGCGACAAA CctCTAGTGGATCATTAACGCTTAC). A resultant clone designated as *IL11-Egfp* was verified by restriction digestion of BAC DNA and by sequencing.

Intracytoplasmic sperm injection (ICSI) was performed as previously described with slight modifications[41]. The mixture of sperm and *Il11-Egfp* DNA was diluted with Hepes-modified CZB containing 12% polyvinylpyrrolidone (Sigma-Aldrich) before being used for ICSI. Injections were performed by micromanipulators (Leica) with a PMM-150 FU piezo-impact drive unit (Prime Tech) using a blunt-ended mercury-containing injection pipette. After discarding the midpiece and tail, the head of spermatozoa was injected into an oocyte from C57/BL6 mice. Oocytes matured into two-cell stage embryos 24 h after injection, and then two-cell stage embryos were transferred to the oviducts of pseudopregnant females. Transgenic founders were backcrossed to C57/BL6J mice for several generations. One line exhibited an intimate correlation of *Il11*, and *Egfp* expressions were selected and used for further experiments.

**Induction of DSS-induced colitis and colitis-associated cancer (CAC) in mice**. Nine to fifteen-week-old male wild-type or *Il11-Egfp* reporter mice received 1.5% DSS (MW: 36,000–50,000 D; MP Biomedicals) ad libitum in drinking water for 5 days, which then was changed to regular water. To reduce gut commensal

microflora, mice received mixtures of antibiotics in drinking water containing ampicillin (1 g/L, Sigma-Aldrich), kanamycin (0.4 g/L, Sigma-Aldrich), gentamicin (0.035 g/L, Sigma-Aldrich), metronidazole (0.215 g/L, Sigma-Aldrich), vancomycin (0.18 g/L, Sigma-Aldrich), and colistin (0.042 g/L, Sigma-Aldrich). Administration of antibiotics into mice started at 4 weeks before DSS treatment and continued during DSS treatment. To attenuate oxidative stress in DSS-treated mice, mice received NAC (10 g/L) along with DSS in drinking water for 5 days. To neutralize TGFβ in DSS-treated mice, mice were intraperitoneally injected with anti-TGFβ antibody (1D11) or control mouse IgGs (5 mg/kg) day 2 and day 4 after DSS treatment. To inhibit ERK activation in DSS-treated mice, a MEK inhibitor, trametinib (2 mg/kg), was administered into mice by gavage at 30 and 6 h before DSS treatment at a fine suspension in 0.5% Hydroxypropyl Cellulose (Alfa Aesar) and 0.2% Tween-80.

To test whether EGFP$^+$ cells were proliferating in the colon of DSS-treated mice, we labeled proliferating cells in *Il11-Egfp* reporter mice by intraperitoneal BrdU injection (B5002, Sigma-Aldrich) (40 mg/kg) on day 6 after DSS treatment. Then, mice were killed at 24 hr after BrdU injection. Colonic sections were stained with anti-GFP and anti-BrdU antibodies.

To induce CAC, 9- to 15-week-old female mice were intraperitoneally injected with 10 mg/kg AOM (Sigma-Aldrich). One week later, mice received 1.5 % DSS *ad libitum* in drinking water for 5 days, followed by 2 weeks of regular water, and this was repeated for two additional cycles. To reduce numbers of commensal bacteria and oxidative stress in AOM/DSS-treated mice or *Apc^{min/+}* mice, we administered mixtures of antibiotics and NAC into the indicated mice for the last 8 weeks and 4 weeks just before the sacrifice, respectively. Trametinib (2 mg/kg) (−6 and −30 h) or anti-TGFβ antibody (5 mg/kg) (on day −1, −3, −5) were administered into AOM/DSS-treated mice or *Apc^{min/+}* mice at the indicated days just before sacrifice.

**Flow cytometry**. To isolate IL-11$^+$ cells, the colon was removed from DSS-treated *Il11-Egfp* reporter mice. Then, the colon was minced with scissors and digested in RPMI 1640 containing 100 U/mL Penicillin and 100 μg/mL Streptomycin, 1 mg/mL Collagenase (Wako), 0.5 mg/mL DNase (Roche), 0.5 mg/mL Dispase (Roche), and 2% (v/v) fetal bovine serum (FBS, Gibco) for 60 min. Single-cell suspensions were prepared, and cells were stained with the indicated antibodies and analyzed by LSRFortessa X-20 (BD Biosciences) or BD Verse (BD Biosciences). Data were processed by FlowJo software Version 10.7.1 (FlowJo), BD FACSDiva Software Version 8.0.1 (BD Bioscience), and BD FACSuite Software Version 1.0 (BD Bioscience).

To isolate IL-11$^+$ cells from tumors in the colon of *Il11-Egfp* reporter mice treated with AOM/DSS or *Apc^{min/+};Il11-Egfp* mice, tumor tissues were removed from non-tumor tissues. Then, single-cell suspension from tumor tissues was prepared as described above. Cells were stained with the indicated antibodies, and the expression of various cell surface markers on EGFP$^+$ cells was analyzed by flow cytometry.

**Generation of BM chimeras**. To test whether BM cells were differentiated into IL-11$^+$ stromal cells in the tumor microenvironment, we isolated BM cells from *Il11-Egfp* reporter mice (CD45.2) and then transferred BM cells into 4- to 5-week-old recipient mice [C57BL/6-SJL mice (CD45.1)] that had been exposed to lethal irradiation (9.0 Gray). One month following reconstitution, BM transferred mice were subjected to AOM/DSS treatment.

For reciprocal BM transfer experiments, BM cells were prepared from *Il11^{+/+}* or *Il11^{−/−}* mice (CD45.2). Then, 3–5 × 10$^6$ BM cells were transferred to 8-week-old recipient mice [C57BL/6-SJL mice (CD45.1)] that had been exposed to lethal irradiation (9.0 Gray). In reciprocal BM transfer experiments, BM cells from wild-type C57BL/6-SJL mice (CD45.1) were transferred to *Il11^{+/+}* or *Il11^{−/−}* mice (CD45.2) that had been exposed to lethal irradiation (9.0 Gray).

At 2–3 months after the transfer, peripheral blood mononuclear cells were collected and stained with FITC-conjugated anti-CD45.1 and allophycocyanin (APC)-conjugated anti-CD45.2 antibodies. The chimerism of BM cells was calculated by counting numbers of CD45.1$^+$ and CD45.2$^+$ by flow cytometry. Average chimerisms were more than 90%.

**Quantitative PCR (qPCR) assays**. Total RNAs were extracted from the indicated tissues of mice by using TRI Reagent (Molecular Research Center) or Sepasol II Super (Nacalai Tesque), and cDNAs were synthesized with the RevertraAce qPCR RT Kit (Toyobo). To remove residual DSS, mRNAs prepared from the colon of mice treated with DSS were further purified by LiCl precipitation as described previously[42]. Quantitative polymerase chain reaction (qPCR) analysis was performed with the 7500 Real-Time PCR detection system with CYBR green method of the target genes and murine *Hprt* an internal control with 7500 SDS software 2.3 (Thermo Fisher Scientific). The primers used in this study are shown in Supplementary Table 3.

**Transcriptome analysis**. We compared gene expression profiles of RNAs from EGFP-positive and negative fibroblasts from the colon of *Il11-Egfp* reporter mice (*n* = 3 mice) on day 7 after DSS administration. To isolate EGFP$^+$ and EGFP$^-$ cells by a cell sorter, colon tissues were minced with scissors and digested in RPMI1640 containing 0.25 mg/mL Liberase (Sigma), 50 μg/mL DNase (Roche), 100 U/mL

Penicillin, 100 μg/mL Streptomycin, and 5% (v/v) fetal bovine serum (Gibco) for 60 min. EGFP$^+$ and EGFP$^-$ cells among Ter119$^-$CD45$^-$ CD31$^-$ EpCAM$^-$ cell populations were enriched using MojoSort Mouse anti-APC Nanobeads (BioLegend) and MojoSort Magnet (BioLegend) and sorted by BD FACSAriaTM III Cell Sorter (BD Biosciences). Total RNAs were subjected to RNA-seq analysis. We collected roughly 10,000 fibroblasts per each group, and purity of EGFP$^+$ fibroblasts was ~90%. RNA-seq analyses were performed using the 5′Tag-seq method as described previously[43]. Briefly, polyA RNAs were trapped by oligo-dT-immobilized magnetic beads and reverse-transcribed by using Superscript IV (Thermo Fisher Scientific). Then, polyC tailing and 2nd strand synthesis were performed, and cDNA was amplified by PCR. Resulted cDNA products were converted to Ion Torrent sequencing libraries using NEBNext UltraII FS library prep kit (New England Biolabs). RNA sequencing was performed with Ion 540 Kit-Chef, Ion 540 Chip Kit, and Ion Genestudio S5 Sequencer (Thermo Fisher Scientific) according to the manufacturer's instructions. Adapter trimming and quality filtering of resulting fastq files were performed by using Cutadapt-v1.18[44]. Trimmed reads were mapped to the Refseq mm10 RNA using Bowtie2-2.2.5[45] with the following parameters: t -p 16 −N 1 -D 200 −R 20 -L 20 -i S,1,0.50 -nofw. Tag numbers of each gene represented the expression level of each gene and were used as count data. Genes with tag number ≥10 were deemed low expressed and were excluded from the analysis. The sample to sample normalization was performed with R (version 3.5.1) and TCC package[46]. Normalized data were then tested for differential gene expression using the TCC package, which integrates the edgeR package[47]. Genes with adjusted *P* < 0.05, fold-change >2 were identified as statistically significant differentially expressed genes. GO enrichment analysis of differentially expressed genes were performed using the DAVID Bioinformatics Resources (https://david.ncifcrf.gov/home.jsp). Data were deposited in NCBI as a GEO accession number GSE141644 and GSE164232.

To identify target genes by IL-11 in the colon, we intravenously injected 10 μg of IL-11R agonist (provided by K. Tsumoto) into wild-type mice (*n* = 3 mice). IL-11R agonist is a modified version of human IL-11, where the mutations were introduced into the linker region of human IL-11 to increase stability and biological activities to 10 fold higher than the original IL-11. The detailed characterization was described elsewhere[48]. Mice were sacrificed at 3 h after injection. Total RNAs were extracted from the colon of mice by using Sepasol II Super. Labeled cRNAs were prepared from total RNA using the Agilent's Quick Amp Labeling Kit (Agilent). Following fragmentation, cRNA were hybridized to SurePrint G3 Mouse Gene Expression 8x60K (Agilent) according to the manufacturer's instruction. Raw data were extracted using the software provided by Agilent Feature Extraction Software (v11.0.1.1). The raw data for the same gene were then summarized automatically in Agilent feature extraction protocol to generate raw data text file, providing expression data for each gene probed on the array. Array probes that have Flag A in samples were filtered out. The selected processed signal value was transformed by logarithm and normalized by quantile method. Statistical significance of the expression data was determined using fold change and local-pooled-error test[49] in which the null hypothesis was that no difference exists among two groups. Hierarchical cluster analysis was performed using complete linkage and Euclidean distance as a measure of similarity. Gene-Enrichment and Functional Annotation analysis for significant probe list was performed using Gene ontology (http://geneontology.org). All data analysis and visualization of differentially expressed genes was conducted using R 3.0.2 (www.r-project.org). Data were deposited in NCBI as a GEO accession number GSE141643 and GSE141644.

For comparing enriched genes in IL-11$^+$ fibroblasts and IAFs, we used Gene Set Enrichment Analysis (GSEA) software (version 4.1, Broad Institute) and *p*-value was calculated by two-tailed Kolmogorov-Smirnov test[50,51].

**Immunohistochemistry (IHC)**. Tissues were fixed in 10% formalin and embedded in paraffin blocks. Paraffin-embedded colonic sections were used for H&E staining, immunohistochemical, and immunofluorescence analyses. For IHC, paraffin-embedded sections were treated with Instant Citrate Buffer Solution (RM-102C, LSI Medicine) or Target Retrieval Solution (S1699, Dako) appropriate to retrieve antigen. Then tissue sections were stained with the indicated antibodies, followed by visualization of Alexa-conjugated secondary antibodies or biotin-conjugated secondary antibodies followed by Streptavidin-HRP (Vector Laboratories).

In tumor samples, tissue sections were preincubated with MaxBlock™ Autofluorescence Reducing Kit (MaxVision Biosciences) according to the manufacturer's instructions. After blocking, tissue sections were stained with the indicated antibodies as described above.

Pictures were obtained with an all-in-one microscope (BZ-X700, Keyence) or BX-63 (Olympus) and analyzed by BZ-X Analyzer (Keyence) or cellSens (Olympus) software. Positive areas and signaling intensities (for IL-11 staining) for the indicated antibodies and respective total areas were automatically calculated using Hybrid Cell Count (Keyence). Confocal microscopy was performed on an LSM 880 (Zeiss) or A1R (Nikon). Images were processed and analyzed using the ZEN software (Zeiss) or the NIS-Elements (Nikon) or an image-processing package, Fiji (https://fiji.sc/).

**Organoid culture and transplantation of tumor organoids**. Mouse colon epithelial and tumor organoids were established from isolated crypts of wild-type mice

and mutant mice carrying a truncated *Apc* gene (*Apc$^{delta716}$*) as previously described[52]. Obtained colonic organoids were cultured in Advanced DMEM/F12 (Thermo Fisher Scientific) containing 10 mM HEPES (Thermo Fisher Scientific), 1 x GlutaMAX (Thermo Fisher Scientific), 1 x B27 (Thermo Fisher Scientific), NAC (1 μM), murine epidermal growth factor (50 ng/mL, Thermo Fisher Scientific), murine Noggin (100 ng/mL, Peprotech), 0.5 mM A83-01 (Tocris), 3 μM SB202190 (Sigma), 1 μM nicotinamide (Sigma), Afamin and Wnt3A-condition medium (CM) (final concentration of 50%) which was provided by J. Takagi[53] and R-spondin 1-CM (final concentration of 10%) (Trevigen). Colon epithelial organoids from wild-type (WT) and *Apc$^{delta716}$* mice were stimulated with IL-11 (100 ng/mL) or IL-22 (100 ng/mL) for 30 min unless otherwise indicated. Total and phosphorylated STAT3 were analyzed by Western blotting.

The small intestinal tumor organoids from *villin-CreER-Apc$^{delta716}$-Kras$^{+/LSL-G12D}$-Tgfbr2$^{flox/flox}$-Tp53$^{+/LSL-R270H}$* (AKTP) mice were described previously[34]. AKTP organoids were cultured in Advanced DMEM/F12 containing 10 mM HEPES, 1 x GlutaMAX, 1 x B27, NAC (1 μM), murine epidermal growth factor (50 ng/mL), murine Noggin (100 ng/mL). For transplantation, *AKTP* organoids were mechanically dissociated, and $3 \times 10^5$ organoid cells mixed in Matrigel were injected into the subepithelial tissues of the rectum of *Il11-Egfp* reporter mice. At five weeks after transplantation, tumors were removed and subjected to histological analysis.

*AKTP* organoids, HT29, and HCT116 cells were stimulated with murine or human IL-11 (100 ng/mL) for 30 min. Total and phosphorylated STAT3 were analyzed by Western blotting.

**Isolation and stimulation of colonic fibroblasts**. To isolate colonic fibroblasts, single-cell suspensions were prepared as described above and cultured in DMEM containing 10% FBS, 1% GlutaMAX (Thermo Fisher Scientific), 10 mM HEPES, 1% MEM Non-Essential Amino Acids Solution (100×) (Nacalai Tesque), 100 U/mL Penicillin, 100 μg/mL Streptomycin, and 2.5 μg/mL of Amphotericin B (Sigma). The culture medium was changed every day until colonic fibroblasts started to grow spontaneously. Colonic fibroblasts ($5 \times 10^5$) were unstimulated or stimulated with mouse TGFβ1 (100 ng/mL) for 16 h. Concentrations of murine IL-11 in the culture supernatants were determined by ELISA according to the manufacturers' instruction (R&D Systems).

To detect phosphorylation of STAT3, colonic fibroblasts were stimulated with IL-11 (100 ng/mL) for 30 min. Total and phosphorylated STAT3 were analyzed by Western blotting.

**Western blotting**. Cells were lysed in a RIPA buffer (50 mM Tris-HCl, pH 8.0, 150 mM NaCl, 1% Nonidet P-40, 0.5% deoxycholate, 0.1% SDS, 25 mM β-gly-cerophosphate, 1 mM sodium orthovanadate, 1 mM sodium fluoride, 1 mM PMSF, 1 μg/ml aprotinin, and 1 μg/ml leupeptin). After centrifugation, cell lysates were subjected to SDS-PAGE and transferred onto polyvinylidene difluoride membranes (Millipore). To detect IL-11, cell lysates were subjected to SDS-PAGE under non-reducing conditions. The membranes were immunoblotted with the indicated antibodies. The membranes were developed with Super Signal West Dura extended duration substrate (Thermo Scientific) and analyzed by Amersham Biosciences imager 600 (GE Healthcare). In some experiments, blots were quantified using the freeware program Fiji.

**Comparison between the enriched genes in IL-11$^+$ fibroblasts and in the genes in IL-11-stimulated ColVI$^+$ colon fibroblasts**. The publicly available data set (E-MTAB-7764) was obtained from ArrayExpress database (www.ebi.ac.uk/arrayexpress). FASTQ files were pre-processed by the Fastp 0.20.1 (https://github.com/OpenGene/fastp), then they were quantified with Salmon algorithm 0.8.0 (https://github.com/COMBINE-lab/salmon). Analysis of differential gene expression was performed using DESeq2 1.14.1 with the following parameters: test: Wald; fitType: parametric. Genes with adjusted $p < 0.01$, fold change >2 were identified as statistically significant differentially expressed genes.

**Comparison of the expression of enriched genes in IL-11$^+$ fibroblasts between normal human mucosa and colon cancer tissues**. The publicly available data set (GSE33113) was obtained from the Gene Expression Omnibus (GEO). This data set contains gene expression data from 90 colon cancer patients and six healthy persons. The expression data of enriched genes in IL-11$^+$ fibroblasts was retrieved from the data set (GSE33113), and their signaling intensities of each gene in normal mucosa and colon cancer tissues were compared.

**Comparison of the expression of enriched genes in IL-11$^+$ fibroblasts between epithelium and stroma compartments in human colon cancer tissues**. The publicly available data set (GSE35602) was obtained from the GEO. This data set contains expression profiling data from 13 CRC tissues were micro-dissected using Laser Microdissection System (Leica Microsystems). The expression data of enriched genes in IL-11$^+$ fibroblasts was retrieved from the data set (GSE35602), and their signaling intensities of each gene in tumor stroma compartments and tumor epithelium compartments were compared.

**Enzyme-linked immunosorbent assay (ELISA)**. Concentrations of murine IL-11 in the culture supernatants were determined by ELISA according to the manu-facturers' instruction (R&D Systems).

**Knockdown of *Il11* by siRNAs**. A human breast cancer cell line, MDA-MB-231 cells were provided by T. Sakamoto. MDA-MB-231 cells were maintained in DMEM containing 10% FBS. MDA-MB-231 cells were transfected with control or *Il11* siRNAs using Lipofectamine 2000 (Invitrogen) at a final concentration of 50 nM. At 36 h after transfection, knockdown efficiency of IL-11 by siRNAs was analyzed by immunoblotting with anti-IL-11 antibody using cell lysates. Stealth RNAi™ siRNAs Negative Control, Med GC and human *Il11*, NM_000641.3 (HSS179893, HSS179894, HSS179895) were purchased from Thermo Fisher Scientific.

**Human CRC tissues**. Human colon tumors and adjacent normal tissues were obtained from Toho University Omori Medical Center. Colon tumors included 11 cases of adenomas, 10 cases of early cancers, and 10 cases of advanced cancers. Detailed clinical information is included in Supplementary Table 1. Histological assessment of adenomas and adenocarcinomas was according to the guidelines of the World Health Organization[54]. Early and advanced colon cancers were deter-mined according to the TNM criteria (pT1, early cancer; pT2-4, advanced cancers)[55]. The present study was designed as a retrospective study and was approved by the Ethics Committee of Toho University School of Medicine (A16111). The need for informed consent was waived by the Ethics Committee.

**Survival analysis**. Three publicly available data sets (GSE14333, GSE17536, and GSE17537) were obtained from the GEO. These data sets contain gene expression data and disease-free survival information from 232 primary CRC patients. Based on the expression patterns of *IL11*, a combination of *IL11/IL24/IL13A2/TNFRSF11B*, or IL11FS, we classified patients into two respective clusters using the hierarchical clustering method. The recurrence-free duration of each group was calculated using the Kaplan–Meier method. Statistical significance was analyzed by Mantel-Cox log-rank test. The hierarchical cluster analysis was performed using Ward's method and Euclidean distance as a measure of similarity. All data analysis and visualization of differentially expressed genes was conducted using R 4.0.2 (www.r-project.org). Statical analyses were conducted with the following packages: MASS 7.3.51.4, stats 3.6.1, ggplot2 3.2.1 and BiocManager 1.30.10.

**Statistical analysis**. Statistical significance was determined by the unpaired two-tailed Student's *t*-test, two-tailed Mann–Whitney *U* test, two-way ANOVA with Tukey's multiple comparison test, one-way ANOVA with Tukey's multiple com-parison test, two-tailed Kolmogorov-Smirnov test (for GSEA), two-tailed local-pooled-error test (for microarray analysis), or two-tailed Mantel-Cox log-rank test as indicated. *$P < 0.05$ was considered to be statistically significant. All statistical analysis was performed with Graph Pad Prism 7 software (GraphPad Software).

**Reporting summary**. Further information on research design is available in the Nature Research Reporting Summary linked to this article.

## Data availability

The RNA-seq and microarray data generated in this study have been deposited in the NCBI database under accession code GEO accession numbers GSE141643, GSE141644, and GSE164232. The microarray data used in this study are available in the NCBI database under accession code GEO accession numbers GSE14333, GSE17536, GSE17537, GSE35602, GSE33113. The RNA-seq data used in this study are available in the EMBL-EBI database under accession code number E-MTAB-7764. Source data are provided with this paper.

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

## Acknowledgements

We thank L. Robb for *Il11ra1*$^{-/-}$ mice, K. Asano and M. Tanaka for the generation of anti-mouse IL-11 antibody, M. Ohmuraya, K. Araki and T. Yamamoto for the generation of *Il11*$^{-/-}$ mice, K. Tsumoto for IL-11R agonist, T. Sakamoto for MDA-MB-231 cells, K. J. Takagi for Afamin- and Wnt3A-conditioned medium, and J. Yasuda for assistance with RNA-seq analysis. We also thank Y. Kurashima, H. Kiyono, T. Sato, M. Kikkawa, M. Aoki, A. Nakajima, E. Tosti, and L.S. Lopez for technical advice. This work was supported in part by Grants-in-Aid for Scientific Research (B) 20H03475 (to H.N.); Grants-in-Aid for Scientific Research (C) 19K07391 (to T.N.) from the Japan Society for the Promotion of Science (JSPS); the Japan Agency for Medical Research and Development (AMED) through AMED-CREST (JP20gm1210002, to H.N.); the Private University Research Branding Project (to T.M. and H.N.) from the MEXT (Ministry of Education, Culture, Sports, Science and Technology), Japan; Toho University Grant for

Research Initiative Program (TUGRIP) from Toho University (to H.N.); Extramural Collaborative Research Grant of Cancer Research Institute, Kanazawa University (to M.N., M.O., H.N., and T.N.); the Project Research Grant of Initiative for Realizing Diversity in the Research Environment, Toho University (to T.N.), and research grants from the Uehara Memorial Foundation (to T.N.) and the Takeda Science Foundation (to T.N.).

## Author contributions

T.N. and H.N. designed the research; T.N., Y.D., W.T., M. Ohtsuka., D.O., S.Y., M.K., E.N., Y.K., S.A-A., M.H., N.I., and N.T. performed the research; S.S., S.U., and K.M. performed the RNA-seq analysis; M.N., M. Oshima., H.Y., K.S., and T.M. contributed to new reagents/analytical tools; T.N., Y.D., W.T., D.O., N.I., and H.N. analyzed data; and T.N., N.I., and H.N. wrote the paper.

## Competing interests

The authors declare no competing interests.
