## [Peer Review File · Nature Communications]

Reviewers' comments:

Reviewer #1 (Remarks to the Author): expertise in IL11 and colorectal cancer

The authors explore the contribution of IL11 producing fibroblasts to colon cancer in murine models. To do this, they have generated a novel IL11 reporter mouse, I am not aware of any other reporter mouse that is available – so these observations add greatly to the literature.

They further explore the contribution of IL-11 signalling to CRC using an IL11KO mouse, and IL11RKO mouse in the CAC model – the latter of which has been published previously. They generated the IL11KO mouse, which was reported in a previous publication.

The authors argue that IL11+ cells in the colon are an IAF phenotype that may contribute to cancer progression. In some ways, many of their findings contradict previous publications, which should be discussed in more detail. Specific examples are outlined in the comments below.

While the specific contribution of IL11 expressing fibroblasts has not been fully explored, and is beyond the scope of this manuscript, I believe that major revisions are required for the data to convincingly support the conclusions, and be of a standard that would be expected for this journal.

I hope that the comments I have provided prove useful in this process.

SPECIFIC COMMENTS:

Introduction

(1) The introduction should expand on the observations by Calon et al (Cancer Cell 2012), where it was demonstrated that FACS isolated fibroblasts from CRC patients express the highest IL-11 compared to the epithelial and endothelial cells. It should also expand on the observations of Heichler et al (Gut 2019), where a fibroblast Cre with a reporter allele was used to demonstrate the presence of fibroblasts in tumours of the CAC model, and clearly demonstrated that IL-11 induces STAT3 activation in both colon fibroblasts and CAFs.

(2) Line 92 requires references.

(3) Heichler et al (Gut 2019) should also be referenced, for the experiments where recombinant IL11 was provided in the AOM/DSS model and tumour burden was larger.

(4) The introduction indicates that IL11 could attenuate colitis under certain conditions. This comment is not addressed in their results or discussion, nor is the observation that IL11 overexpression leads to colon fibrosis indicated (Lim et al PLoS One 2020 9:15).

Figure 1

The authors describe the generation and validation of a BAC transgenic reporter mouse, and characterise IL11 producing cells in a model of CAC. Additional data is required to validate the observations made with this mouse, as indicated below.

(5) Line 129, indicates that the "origin of IL11+ cells remains controversial". I don't find this to be an appropriate statement here, as they are not examining their origin, but rather visualising where they are localised. This sentence should be changed.

(6) Line 138 indicates "increased numbers" for Fig 1D, but the FACS is graphed as the % of cells. Were cell counts also performed? These should be shown, as % and # can be different.

(7) Panel e: quantification of the frequency of each IL11+ population is required, rather than just

representative plots. Are all of the "fibroblasts" IL11+? Or only a % of the fibroblasts?

(8) Panel g: In order to make the statement that 'a few E-Cadherin positive cells' requires quantification of the images, with appropriate N for statistical analysis.

(9) The authors indicate that IL-11+ cells are not present before AOM/DSS, I suggest this flow cytometry data is included in the supplemental material, or with the above experiment. Or, a reference made to the acute DSS experiments in subsequent figures.

(10) Panel h: The authors indicate that "the majority of IL11+ cells did not express Ki67", with only one image shown. To justify this statement, Flow cytometry quantification of eGFP/IL11+ and Ki67+ cells is required from non-tumour and tumour tissue in the CAC model. Additional fibroblast markers can be included in the panel to highlight that they are not proliferating.

(11) Panel h: The authors attempt to address the question as to whether IL-11 positive cells expand in situ, or whether an IL11- cell turns into an IL11 positive cell. To do address this question, they examine whether IL11+ cells express the proliferation marker ki67. Really, this experiment shows that IL11+ cells are likely not 'activated normal fibroblasts' or represent the expansion of 'quiescent fibroblasts', which would be expected to be proliferating. In my opinion it is a major limitation of this study that they did not address the possibility that instead they may represent a population of CAFs recruited from the bone marrow, for example, or they may be a population of cells undergoing EMT. The authors can easily address the recruitment scenario, by generating bone marrow chimeras with eGFP into WT mice, induce AOM/DSS and determine if IL11+ eGFP positive cells are present in the tumors of WT mice. Of note, they indicate in the Summary that the 'origin' of IL-11 producing cells is not fully understood, this would assist them with addressing this.

Minor comments:

(12) The AOM/DSS schematic goes to 72 days, the tumour shown is 77 days, and the figure legend indicates that the mice were collected 98-105 days after AOM. The schematic should better represent the experiment timeline used.

(13) The % IL11+ cells by FACS is quite low, compared to the number of positive cells observed by IHC. Is the GFP antibody specific, were staining controls performed? Has the IHC been quantified (ie green cells vs DAPI positive cells)? Does an IL11 antibody match the reporter expression in tissue? a number of commercial IL11 IHC antibodies are available.

Supplemental Figure 2.

The authors perform the CAC model in IL11RKO and IL11KO mice, and validate previous observations of a role for IL-11 in the CAC model.

(14) A, B) can the authors check their statistical analysis? It does not appear that many of the results should be significant as there is quite a lot of overlap in the dots. Are the animals gender matched? Could the male vs female differences be skewing the data? An indication of male and female mice on the graphs may help determine this. They acknowledge that these results are not as striking as previous publications (Putoczki et al Cancer Cell 2013 24:257) which may be due to differences in the background of the mice, it also appears that the experiments in that publication were performed on female mice.

(15) C, D) It is curious that WT bone marrow led to a difference in tumour number, but not overall area. While IL11KO bone marrow led to a difference in tumour area, but not tumour number. It appears that this is one large experiment. The experiment should be repeated at least once to determine if this observation holds true.

(16) Further to my comments above, IL11KO bone marrow appears to decrease the overall tumour area. It will be important to determine if eGFP bone marrow makes its way to the tumour of a WT mice.

(17) Does the loss of IL11 similarly affect tumor burden in the APCmin mice?

Figure 2.

(18) Line 176, it is an overstatement to suggest that the "tumours educated the IL11- cells to become IL11+ cells", this line should be removed. Moreover, the experiments performed do not demonstrate that "IL11- cells might cell-autonomously become IL11+ within the setting of the inflammatory milieu triggered by DSS". They only reveal their location.

(19) APCmin mice primarily get tumours in the SI and less frequently get tumours in the colon. Do the tumours in any region of the GI tract all have IL11+ cells within them?

Does the loss of IL-11 in APCmin mice impact tumours throughout the GI tract? Or is this specific to the colon?

(20) Panel D) in order to say almost all cells expressed Podoplanin, graphs of this quantification is required. Similar comments for all of the markers.

(21) Panel E-F) Quantification of the % EFGP+ and each marker is required to make conclusive statements. Ie, it is inappropriate to state that "most IL11+ CAFs were not Ki67+", without showing the quantification.

Supplemental Figure 3:

(22) do the tumour organoids express IL11R?

(23) It would be useful to have markers to confirm the organoid tumour growth, and not wounded epithelium following injection of the organoids. Are endoscopy images available? Is histology of the entire tumour with the underlying mucosa available?

(24) I disagree that the data indicates that the "tumour cells instructed IL11- cells to become IL11+ cells in the absence of inflammation". I feel quite strongly that the potential for IL11+ cells to be recruited to the tumour site needs to be excluded.

It is an important point that a reporter mouse with an IL11-IRES-GFP could address the transient nature of IL11 production, while the current reporter mouse fate maps the production of IL11.

Figure 3

(25) Line 205, I would suggest altering the first statement from "to test whether inflammation alone induced IL-11+ cell development" (as the experiments do not address the development of the cells), to "to determine whether inflammation along induced IL-11 expression". Really, the experiment the reporter mouse allows for is characterisation of source and expression. It doesn't address 'development' or 'regulation' of gene expression.

(26) Panel c) it is not possible to determine if there was a "rapid response" of IL-11 producing cells after 1 day of DSS, without quantification of the number of IL11+ cells at this stage compared to untreated to determine if this is significant. Indeed, gene expression is low on day 7 (panel a) – but there appears to be some variability in the DSS mice.

(27) Panel d) there are 2% IL11+ cells, compared to the lower % of the tumours in previous figures. Can the authors comment on why there may be so many more IL11+ cells following DSS treatment?

(28) Assuming that the number of IL11+ cells rises and then decreases after DSS, ie is low in the

non tumour tissue (which is what Fig 1d) suggests, when does IL-11 expression decrease in the AOM/DSS model (water cycle 1, water cycle 2)? What is the role of IL11 in the acute DSS phase compared to the AOM/DSS model?, if it is higher in the acute DSS model one would presume it has a major role. Is there a difference in disease pathology in IL11KO and IL11RKO mice in the acute DSS model?

(29) Quantification of the IL11+ BrdU+ cells across the sections is required (similar comments to other figures).

(30) I would adjust the sentence on line 222/223, as the experiment does not address 'development', but rather indicates presence vs absence.

Figure 4

(31) I find the set-up of this experiment perplexing. What was the profile of the EGFP- sorted cells? Were these sorted eGFP- fibroblasts vs eGFP+ fibroblasts? This needs to be clarified in the figure and text, as a direct comparison of one cell type (ie eGFP+, which up to this point the authors suggest are fibroblasts, but have shown include CD45 and EPCAM+ cells in Figure 2) to bulk tissue composed of a number of cell types makes it difficult to determine what the comparison would be expected to show – there would of course be major differences? A better comparison for microarray or bulk RNAseq would be sorted fibroblast eGFP+ vs fibroblast eGFP- to determine if there are differences in the fibroblasts (please see previous comments on what % of fibroblasts are eGFP+). Even better, but beyond the scope of this manuscript would be single cell analysis of the eGFP positive cells.

(32) Panel E) The authors suggest that IL11+ CAFs after acute DSS are an IAF subtype based on the gene expression profile. Earlier they suggested that IL11+ CAFs were present in tumour from Apc mice and CAC mice, so were not specifically inflammation related. If these IL11+ CAFs do not persist throughout the AOM/DSS model, it is not possible to conclude that they are the same IAFs in the cancer models based on the data presented. Is the gene signature of FACS isolated IL11+ cells in the CAC tumours and the Apc mice an IAF signature as well, or are they different?

Supplemental Figure 4:

(33) In following from comment #32 - The IAF signatures they examined in whole tumour tissue could be from other cell populations, as they are not genes restricted to fibroblasts. It would be more convincing to FACS isolate the IL11+ cells from the tumours, and perform qRT or perform Flow to show the IL11+ cells have expression of the other markers if antibodies are commercially available.

(34) I would suggest rewording of line 256, as the data presented do not compare the "phenotype" of the IAF and CAFs as suggested by the concluding statement – but rather, the data suggest the presence of IAFs based on the tumour gene expression. This can be confirmed by the above suggestion.

Figure 5.

(35) In panel C, the 16S expression level appears to average 0.1, while in panel D, it averages at 1. Can the authors explain the reason for such differences in the control cohorts for these experiments?

(36) In panel F, it appears that two outliers contribute to the significance, has this experiment been performed more than once? And do the same trends hold?

(37) In Panel E and G the addition of western blots (with 3 mice per group) would better demonstrate the consistency of the loss of pERK. Conclusions can not be supported by one representative image.

(38) The reasoning for panel h are not clearly articulated in the text. One assumes the intent was to demonstrate that TGFb signalling components are expressed, and thus one presumes are functional.

(39) In panel I: do the authors have a control to demonstrate that the concentration of anti-TGFb that was used successfully neutralised TGFb signalling? Conversely, does administration of recombinant TGFb increase IL-11 expression in the colon? Or does TgfB addition to eGFP+ cells isolated and grown in culture increase their IL-11 expression? Similar comments for panel J-O, as the authors see no differences, do they have controls to show that the treatments successfully reduced the targets.

(40) I would suggest further description of the experimental results within the results section. Similar comments re: the TGFb expression panels. If data is presented, the reason for its inclusion should be articulated.

(41) The authors show in 3 independent models that the inhibition of MEK/ERK leads to a reduction in IL11 expression, and suggest that MEK/ERK is involved in the upregulation of IL11 gene expression. Did any of these treatments alter tumour burden (this should be provided in the supplemental)? This information is important in order to determine if the changes in IL11 expression a reflection of changes in overall tumour burden? Others have shown that IL11 induced pERK (Schafer Nature 2017 7:552), the others should discuss how that relates to their observation.

(42) For the min model, what was the effect on the SI tumours, was it the same as the colon? (relates to earlier comments about IL11 expression in SI tumours).

(43) Overall - how do these observations relate specifically to the fibroblasts? Does this similarly change the number or % of infiltrating eGFP+ cells in the different models? (as other cell populations express IL11)? Does the number of fibroblasts remain the same, but the expression of IL11 within them differ (and does this support the idea of IL11 being turned on vs IL11+ cells being recruited in)? Or are these changes specific to the IL11 expressing epithelial cells and CD45 cells?

Figure 6

(44) A number of human CRC cell lines have been shown to be responsive to IL11 and have reduced xenograft growth when signalling is inhibited – suggesting that tumour cells respond (Putoczki Cancer Cell 2013 24:257), and tumour colon organoids have previously been shown to be responsive to IL11 (Pheesse 2014 Sci Signal) which contradict the arguments made here.

The authors need to dig deeper into these discrepancies. Is there a difference between IL11R expression in normal colon organoids and colon tumour organoids (ie do tumour organoids express IL11R, these should be available from data in previous figures)? How many biological replicates do the organoids represent? (ie were organoids generated from more than one mouse, and does each mouse have the same result?). Do normal organoids and fibroblasts similarly respond to the tumour organoids and matched fibroblasts (ie do both tumour organoids and tumour fibroblasts respond)? Do the fibroblasts and tumours FACS isolated from a tumour bearing mouse show the same thing (re: IL11R expression)? Are there differences between in vitro and in vivo observations.

(45) Panel B: would be helpful to have ng/mL on the top axis for the IL11 concentrations.

(46) Panel C: requires additional explanation as to what is being compared? How was the data analysed to determine what is upregulated, are these preparations of the entire colon or fibroblasts specifically? The IL11R agonist should be described more clearly, without the need to refer to previous publications, as it is important to the data conclusions. For example, is this hyper-IL11 (IL11 fused to IL11R) – and thus activating STAT3 in any cells that has gp130, and not specific to an IL11R expressing cell?

(47) Panel E: requires clarification, are the genes elevated in isolated IL-11+ IAFs specifically? Or is this the expression data from Fig 4 of all IL11+ isolated cells? I'm not sure what the intended comparison is... genes IL11+ cells are expressing vs cells where IL11 signalling has been induced by an agonist? Are the authors trying to draw a correlation between a subset of cells that may express IL11 and respond to IL11R in an autocrine manner? What were the 15 genes?

It would be better to show by FACS, and qRT that IL11+ cells also express IL11R, alternatively Flow to show IL11+ cells are also IL11R+, or a subset of them are. And then validate the expression of the 15 genes identified in these cells (+/- IL11 stimulation in culture).

(48) Supplemental Figure 6, I assume this is colon tissue collected? Is there a difference in gene expression across the colon? (as AOM/DSS tumours arise in the distal colon).

Figure 7

(49) The data presented thus far does not conclusively show that IL11+ IAFs exist, but rather suggests that they may be a sub-population. As such, the heading "genes enriched in IL11+ IAFs..." should be dampened down.

(50) Why is the data in Figure 7A from a breast cancer cell line? And Not a colon cancer cell line? And/or fibroblast cell line?? If this is just validation of the specificity of the antibody, this is better provided in the supplemental information as it is out of place here.

(51) In order to make the statement that the majority of the cells that expressed IL11 in the tumours were fibroblasts, the data in figure 7e must be quantified.

Supplemental Figure 7

(52) The authors did not present data for IL24, which is part of the IAF signature, this should be included.

(53) I am not sure I understand the argument that IL11 is not significant in terms of survival, but IL11+ ICAFs may relate to survival? The data for IL11/IL6R/GP130 does not make this conclusion, and would be better with a specific ICAF signature. I.e IL11/IL24/IL13RA3/TNFSFR – which was the signature that they used to suggest that there are IL11 IAFs would be more appropriate.

(54) Given the inclusion of IL-6/IL6R – are these expressed by the IL11+ IAFs?

(55) If IL11 is elevated in advanced disease, but does not correlate with survival – what is its role? And how does this compare with its role in colitis? This should be included in the discussion.

Discussion

(56) Line 364: Indicates that IL11 IAFs expressed genes associated with proliferation (yet they were Ki67-). The authors should discuss this discrepancy.

(57) Line 380. As the authors did not directly look at gene expression IL11+ FACS isolated cells from tumours, they cannot conclude that the IAFs are related/ not related to tumor CAFs or extend this to human comparisons. Moreover, as they have not investigated chemotherapy response, they cannot exclude that these genes are not induced in response to chemotherapy.

(57) There is no data or reference presented to support line 385. If this is a statement based on previous publications, this should be references. If this is data they have generated, it should be appropriately included in the results (only TGFb inhibition was presented).

(58) There is no data from line 390. Either a reference from previous studies should be provided,

or the data included in the manuscript.

(59) The others should discuss their results in light of Calon et al (Cancer Cell 2012) and Yuan et al (Cancer Cell 2014 25:666)

(60) Heichler et al (Gut 2019), Cleary demonstrated that IL-11 induces STAT3 activation in both colon fibroblasts and CAFs and performed gene expression analysis, so this idea is not novel. How their results compare with that of the authors must be included in their figure, and discussed in detail.

Minor comments:

The manuscript would benefit from some minor English language editing.

(61) The methods indicates that mice induced with DSS were 9-15 weeks old, which is a significant variation in age. Are the data presented all age and gender matched?

(62) The methods give DSS disease scores, but no DSS disease scores are presented in the manuscript.

(63) Do the culture fibroblasts secrete IL11 into the media? An IL11 Elisa is indicated in the methods, but no IL11 Elisa Data is shown.

(64) The clone/ supplier of the anti-IL11 antibody was not provided.

Reviewer #2 (Remarks to the Author): expertise in fibroblasts and colitis/colorectal cancer

The manuscript „Interleukin-11 is a Marker for Both Cancer- and Inflammation-Associated Fibroblasts that Contribute to Colorectal Cancer Progression“ by Nishina et al. deals with an interesting research topic. Colon cancer is still of high clinical relevance and studies of fibroblasts both cancer-associated and inflammation-associated are a frequent focus of current GI tract research. The paper is well written and adds information on IL-11 producing cells in the intestine. The authors suggest a concept in which tumor cells induce IL-11 in fibroblasts, and an autocrine feed-forward loop between IL-11 and IL-11+ fibroblasts contributes to tumor development. This concept is not entirely new, however.

The authors report a novel IL-11 -EGFP-reporter mouse that could evolve as a very helpful tool for the research community, in particular as antibody-based IL-11 staining can be challenging. The authors studied IL-11 production in three murine tumor models of CRC, and one murine acute colitis model using reporter mice. The authors report that the majority of IL-11+ cells are cancer- or inflammation-associated fibroblasts, respectively. The manuscript includes profiling of gene transcription patterns of the IL-11+ cells from the DSS model by microarray analysis. However, the comparative group (of total IL-11- cells?) is probably not optimum. They report that IL-11+ cells express genes associated with cell proliferation, tissue repair and suggest that the transcription profiles are also connected with tumor development. They provide interesting evidence with in vivo data that rather ERK/MEK signaling than TGFbeta signaling is the driver of IL-11 induction. The supplemental material contains a large amount of additional findings. The authors report that IL-11 deletion attenuated CAC development using IL-11 k.o. mice (and provide elegant evidence by studies with BM chimeric mice that the non-hematopoietic of the IL+11 cells

are critical) which fits to the known phenotype of IL-11r k.o. mice and the already established concept of IL-11 signaling during in CRC. The manuscript is mainly a murine study, but it also includes wet and dry bench analysis of human data.

The main strength and novelty of the paper is related to the novel reporter mouse. However, I am not enthusiastic because further functional insights into the role of IL11+ CAFs in CRC or IL11+IAFs in colitis are limited. Concerning this matter, the authors present mainly expression data, correlative findings and circumstantial evidence, but functional data on the role of IL-11+CAF and IL-11+IAF on tumor development or colitis severity are lacking. The human data presented are rather weak although claims are made in the abstract and the title. A few shortcomings are related to the methodology, e.g. selection of appropriate controls and how some of the data are presented (see below). Some of the claims are definitely too far-fetched and not sufficiently supported by experimental data. The authors are advised to tone down their claims (abstract, title).

Specific comments:

- The expression data/functionality of the reporter mouse is only partly convincing, FACS data on tumors as presented do not show a clearly distinctive population of IL-11+(EGFP+) cells. Most of the expression data are immunostaining data with a green signal. Without adequate controls, this signal could even result from autofluorescence
- recent papers in the field should be discussed, see Heckler et al. Gut. 2019 Nov 4. pii: gutjnl-2019-319200
- although there are similarities between them, cancer/CAF and inflammation/IAF are different diseases/cells -> suggestive transfers should be avoided, unless no direct comparison is provided
- Fig1: d) reporter mouse signal in FACS staining not very convincing (low numbers <1%, no clear signal of a distinctive population), e) legend/labeling/gating strategy/percentages unclear; g-h controls are missing; a reporter mouse enables detailed and precise quantifications in disease models -> the composition of the pool of IL-11+ cells should be better quantified (FACS data with mean +/-range/SD in mice), please add information on relative number of IL-11 producing cells among fibroblasts + other subsets f)control tissue missing
- Fig2: d) please improve quantitative information (as above), e-g) controls are missing / tissue from non-reporter mice
- Fig3: DSS FACS seems o.k., e) please improve quantitative information (as above); FACS data with the reporter mouse on the kinetics of IL-11+ cells, especially in the acute DSS model, would be helpful
- Fig4: what was the composition of the IL-11- and the IL11+ population? Some genes with highest upregulation in the IL11-(EGFP-) should be also presented. Please clarify, if epithelial cells and/or hematopoietic cells were excluded before analysis. If not: (and IL-11 is mainly found in IAFs) the comparison would not be that fair -> gene expression could be rather a representation of a general fibroblasts signature instead of an IL11+fibroblast signature -> validation experiments with more homogenous cell populations and IL11+fibroblasts would be helpful.
- Fig5: was any influence on tumor growth detected? Are more CAF/IAF specific data available in this context?
- Fig 6: c-e) fig legend/labeling of the figure not clear, please clarify IL-11R agonist (=IL-11?) and indicate what was compared in e)
- Fig7: limited novelty of information, technical: strong IL-11 staining by human tumor epithelial cells in e) seems contradictory to b)+ the murine data + the overall concept of the manuscript; missing isotype controls; the function of a) is unclear: antibody-testing in WB does not need to correlate with specificity in stainings on tissue sections (and scramble siRNA as control is missing) c) unit is missing (per cent?) d) unit is missing; what is meant by early and advanced cancers? Please specify with more accurate clinical terms and further information.
- Sppl. fig1: correlate the protein signal from the reporter mouse with the RNA expression data?
- Sppl fig 7: this is the only human data (along with Fig7): the cutoff levels for the definition of the subgroups in b) and the n per subgroup should be provided; the authors interpretation "IL-11+ fibroblasts correlated with reduced disease-free survival" seems highly speculative based on the data presented

Reviewers' comments:

Reviewer #1 (Remarks to the Author): expertise in IL11 and colorectal cancer

The authors explore the contribution of IL11 producing fibroblasts to colon cancer in murine models. To do this, they have generated a novel IL11 reporter mouse, I am not aware of any other reporter mouse that is available – so these observations add greatly to the literature.

They further explore the contribution of IL-11 signalling to CRC using an IL11KO mouse, and IL11RKO mouse in the CAC model – the latter of which has been published previously. They generated the IL11KO mouse, which was reported in a previous publication.

The authors argue that IL11+ cells in the colon are an IAF phenotype that may contribute to cancer progression. In some ways, many of their findings contradict previous publications, which should be discussed in more detail. Specific examples are outlined in the comments below.

While the specific contribution of IL11 expressing fibroblasts has not been fully explored, and is beyond the scope of this manuscript, I believe that major revisions are required for the data to convincingly support the conclusions, and be of a standard that would be expected for this journal. I hope that the comments I have provided prove useful in this process.

We really appreciate the positive comments on generation of *Il11-Egfp* reporter mice and characterization of IL-11⁺ fibroblasts. We also appreciate your critical comments on some discrepancies between our present results and previous studies. As suggested, we performed a large number of experiments to address the reviewers' concerns and have substantially improved the manuscript. Moreover, we have toned down the contribution of IL-11⁺ fibroblasts to the development of colorectal tumors. Accordingly, we have changed the title to "Interleukin-11-expressing fibroblasts have a unique gene signature correlated with poor prognosis of colorectal cancer".

We have not succeeded in isolation and characterization of IL-11⁺ fibroblasts from murine tumor tissues due to unknown reason, at least our experimental conditions. Thus, these experiments, including single RNAseq

analysis of IL-11⁺ fibroblasts, will be addressed in the future study.

SPECIFIC COMMENTS:

Introduction

(1) The introduction should expand on the observations by Calon et al (Cancer Cell 2012), where it was demonstrated that FACS isolated fibroblasts from CRC patients express the highest IL-11 compared to the epithelial and endothelial cells. It should also expand on the observations of Heichler et al (Gut 2019), where a fibroblast Cre with a reporter allele was used to demonstrate the presence of fibroblasts in tumours of the CAC model, and clearly demonstrated that IL-11 induces STAT3 activation in both colon fibroblasts and CAFs.

RESPONSE: Thank you for the thoughtful suggestions. As suggested, we have incorporated these observations by Calon et al. and Heichler et al. in the Introduction for a better understanding of the causal relationship between colonic fibroblasts and IL-11 production and contribution of STAT3 activation of fibroblasts to the development of CAC (lines 79-85 and 98-106).

(2) Line 92 requires references.

RESPONSE: As suggested, we have included the reference (line 104).

(3) Heichler et al (Gut 2019) should also be referenced, for the experiments where recombinant IL11 was provided in the AOM/DSS model and tumour burden was larger.

RESPONSE: As suggested, we have included their findings in the Introduction (lines 105-106).

(4) The introduction indicates that IL11 could attenuate colitis under certain conditions. This comment is not addressed in their results or discussion, nor is the observation that IL11 overexpression leads to colon fibrosis indicated (Lim et al. PLoS One 2020 9:15).

RESPONSE: Thank you for pointing out a critical issue. Indeed, we had included the results using *Il11ra1*^{-/-} mice treated with DSS in the early version of the manuscript but deleted the results in the submitted manuscript. Since the manuscript is too long at the current version, we hesitate to include the results of *Il11ra1*^{-/-} mice after DSS treatment. Thus, we would like to show the results in the rebuttal Fig. 1 and have deleted the sentences in the text and methods.

Rebuttal Fig. 1. DSS-induced colitis is exacerbated in *Il11ra1*^{-/-} and *Il11*^{-/-} mice. **(a, b)** *Il11ra1*^{+/+} and *Il11ra1*^{-/-} mice were treated with 1.5% DSS in drinking water for 5 days, followed by a change to regular water. Bodyweight (BW) was measured every day from day 5 to day 10 after DSS treatment **(a)**. Percentages of BW (%) was calculated by dividing the BW on the indicated day with that before DSS treatment, and BW kinetics is plotted. Colon length of mice of the indicated genotypes on day 10 after DSS treatment was determined **(b)**. Results are mean \pm SE (n = 7-10 mice). **(c, d)** *Il11*^{+/+} and *Il11*^{-/-} mice were treated and analyzed as in **(a, b)**. Results are mean \pm SE (n = 15-18 mice). Statistical significance was determined using the unpaired two-tailed Student's *t*-test. **P* < 0.05; ***P* < 0.01; ****P* < 0.001; ns, not significant.

Figure 1

The authors describe the generation and validation of a BAC transgenic reporter mouse, and characterize IL11 producing cells in a model of CAC. Additional data is required to validate the observations made with this mouse, as indicated below.

(5) Line 129, indicates that the “origin of IL11+ cells remains controversial”. I don't find this to be an appropriate statement here, as they are not examining their origin, but rather visualising where they are localised. This sentence should be changed.

RESPONSE: As suggested, we have changed the sentence to “cellular sources of IL-11⁺ cells remains controversial.” (line 140).

(6) Line 138 indicates “increased numbers” for Fig 1D, but the FACS is graphed as the % of cells. Were cell counts also performed? These should be shown, as % and # can be different.

RESPONSE: As suggested, we have changed the sentence to “percentages of IL-11⁺ cells” throughout the manuscript (line 148).

(7) Panel e: quantification of the frequency of each IL11+ population is required, rather than just representative plots. Are all of the “fibroblasts” IL11+? Or only a % of the fibroblasts?

RESPONSE: Thank you for indicating a critical point. As suggested, we performed additional experiments and re-analyzed the results of flow cytometry. We calculated the percentages of EGFP⁺ cells expressing each cell surface marker. The majority of IL-11⁺ cells expressed mesenchymal stromal cell markers, such as Thy1.2, podoplanin, CD29, and Sca-1, but not CD31 or Lyve-1, whereas only very small percentages of IL-11⁺ cells expressed EpCAM. Conversely, 4% of podoplanin⁺ cells (mostly fibroblasts) expressed EGFP, indicating that only minor populations of fibroblasts did express IL-11. We have replaced an old Fig. 1e with a new one and made a new Fig. 1f to include these results and mentioned them in the text (lines 150-154).

(8) Panel g: In order to make the statement that ‘a few E-Cadherin positive cells’ requires quantification of the images, with appropriate N for statistical analysis.

RESPONSE: As shown in Fig. 1e, numbers of IL-11⁺ cells expressing epithelial marker (EpCAM) were below 10% of total GFP⁺ cells. We calculated percentages of EpCAM⁺ (another epithelial cell marker) cells by flow cytometry (Fig. 1e). We have changed the sentence to “Consistent with the results by flow cytometry (Fig. 1e), a few E-cadherin-positive tumor cells were positive for EGFP expression (Fig. 1j) “(lines 162-163).

(9) The authors indicate that IL-11⁺ cells are not present before AOM/DSS, I suggest this flow cytometry data is included in the supplemental material, or with the above experiment. Or, a reference made to the acute DSS experiments in subsequent figures.

RESPONSE: As suggested, we calculated percentages of IL-11⁺ cells in the colon by flow cytometry before and after DSS treatment. IL-11⁺ cells were hardly detected in the colon of mice before DSS treatment. We have included these results in Fig. 3d, e and mentioned them in the text (lines 166-168, 259-261).

(10) Panel h: The authors indicate that “the majority of IL11⁺ cells did not express Ki67”, with only one image shown. To justify this statement, Flow cytometry quantification of eGFP/IL11⁺ and Ki67⁺ cells is required from non-tumour and tumour tissue in the CAC model. Additional fibroblast markers can be included in the panel to highlight that they are not proliferating.

RESPONSE: Thank you for your thoughtful suggestion. We tried to perform flow cytometry to examine whether EGFP⁺ cells express Ki67 several times. However, signal intensities of EGFP became low when we stained cells with anti-Ki67 antibody. We surmised that staining conditions to detect Ki67 might reduce the signals of EGFP (data not shown). Thus, we changed the strategy to quantify numbers of EGFP⁺ Ki67⁺ podoplanin⁺ cells by immunohistochemistry. Consistent with Fig. 1e, most IL-11⁺ cells expressed podoplanin in tumor tissues. We found that only 4% of EGFP⁺ (IL-11⁺) cells expressed Ki67, suggesting that the majority of IL-11⁺ cells did not express Ki67. We have made a new Fig. 1k and l to include new results and mentioned them in the text (lines 173-177).

(11) Panel h: The authors attempt to address the question as to whether IL-11 positive cells expand in situ, or whether an IL11⁻ cell turns into an IL11 positive cell. To do address this question, they examine whether IL11⁺ cells express the proliferation marker ki67. Really, this experiment shows that IL11⁺ cells are likely not ‘activated normal fibroblasts’ or represent the expansion of ‘quiescent fibroblasts’, which would be expected to be proliferating. In my opinion it is a major limitation of this study that they

did not address the possibility that instead they may represent a population of CAFs recruited from the bone marrow, for example, or they may be a population of cells undergoing EMT. The authors can easily address the recruitment scenario, by generating bone marrow chimeras with eGFP into WT mice, induce AOM/DSS and determine if IL11+ eGFP positive cells are present in the tumors of WT mice. Of note, they indicate in the Summary that the 'origin' of IL-11 producing cells is not fully understood, this would assist them with addressing this.

RESPONSE: Thank you for your thoughtful suggestions. As suggested, we have transferred BM cells from *Il11-Egfp* mice into WT mice. One month following reconstitution, *Il11-Egfp* BM chimeric mice were treated with AOM/DSS to induce colorectal tumors. As expected, expression of *Il11* was elevated in tumor tissues compared to non-tumor tissues in the colon of *Il11-Egfp* BM chimeric mice treated with AOM/DSS. However, expression of *Egfp* was not elevated in either tumor tissues or non-tumor tissues. Given that *Egfp* expression was a hallmark of *Il11* expression by cells derived from BM cells of *Il11-Egfp* mice, these results suggest that IL-11⁺ cells appeared in tumor tissues may not be derived from BM cells.

Consistent with these qPCR results, IHC revealed that IL-11⁺, but not EGFP⁺ cells, were detected in the tumor tissues in the colon of *Il11-Egfp* BM chimeric mice. In contrast, EGFP/IL-11-double positive cells were easily detected in tumor tissues in the colon of *Il11-Egfp* reporter mice after AOM/DSS treatment (Fig. 1h). Together, these results suggest that the majority of IL-11⁺ cells were not likely derived from BM cells. We have made a new Supplementary Fig. 1g-i to include the results and mentioned them in the text (lines 178-191).

Minor comments:

(12) The AOM/DSS schematic goes to 72 days, the tumour shown is 77 days, and the figure legend indicates that the mice were collected 98-105 days after AOM. The schematic should better represent the experiment timeline used.

RESPONSE: As suggested, we have changed the experimental timeline to a more representative one.

(13) The % IL11+ cells by FACS is quite low, compared to the number of positive cells observed by IHC. Is the GFP antibody specific, were staining controls performed? Has the IHC been quantified (ie green cells vs DAPI positive cells)? Does an IL11 antibody match the reporter expression in tissue? a number of commercial IL11 IHC antibodies are available.

RESPONSE: You have raised an important point. Regarding the specificity of anti-GFP antibody, we could not detect EGFP⁺ cells in the colon of *Il11-Egfp* reporter mice before AOM/DSS or DSS treatment or wild-type mice even after AOM/DSS or DSS treatment (Rebuttal Fig. 2a). These results indicate that anti-GFP antibody did not react with an antigen other than EGFP. Moreover, it is unlikely that GFP⁺ signals solely reflected the autofluorescence of the tissues.

Regarding the discrepancy between percentages of IL-11⁺ cells by flow cytometry and IHC, IL-11⁺ cells preferentially accumulated in the luminal sides of tumor tissues (Rebuttal Fig. 2b). We usually included the images of these areas as representative ones. Thus, percentages of IL-11⁺ cells in the tumor tissues by flow cytometry appeared to be low compared to IHC with anti-GFP antibody.

Moreover, we calculated numbers of IL-11⁺ cells stained with both anti-GFP and anti-IL-11 antibodies by IHC. Approximately 70% of EGFP⁺ cells were positive for IL-11 (Rebuttal Fig. 2c, d). Together, these results indicate that EGFP⁺ cells in the colon of *Il11-Egfp* mice can represent IL-11⁺ cells. We have made a new Fig. 1h, and i and Supplementary Fig. 1e to include these results and mentioned them in the text (lines 155-160).

Rebuttal Fig. 2. *Il11-Egfp* mice monitor appearance of IL-11⁺ cells in vivo. Wild-type (a) and *Il11-Egfp* reporter (a-d) mice were untreated or treated with AOM/DSS as in Fig. 1a. Colonic tissue sections or tumor tissue sections (from AOM/DSS-treated mice) were stained with anti-GFP (a, b) or both anti-GFP and anti-IL-11 antibodies (c). Right panels are enlarged images of the left boxes (b). Numbers of IL-11⁺ cells among EGFP⁺ cells were counted, and their percentages are plotted. Results are mean ± SE (n=4 mice). White arrows indicate cells expressing both IL-11 and EGFP in tumor tissues. Scale bar, 100 μm.

Supplemental Figure 2.

The authors perform the CAC model in IL11RKO and IL11KO mice, and validate previous observations of a role for IL-11 in the CAC model.

(14) A, B) can the authors check their statistical analysis? It does not appear that many of the results should be significant as there is quite a lot of overlap in the dots. Are the animals gender matched? Could the male vs female differences be skewing the data? An indication of male and female mice on the graphs may help determine this. They acknowledge that these results are not as striking as previous publications (Putoczki et al *Cancer Cell* 2013 24:257) which may be due to differences in the background of the mice, it also appears that the experiments in that publication were performed on female mice.

RESPONSE: Yes, we checked the statistical analysis. Indeed, the difference in numbers and sizes of tumors developed in *Il11ra1^{+/+}* and *Il11ra1^{-/-}* mice did not appear to be dramatic compared to those in a previous study (Putoczki et al *Cancer Cell* 2013 24:257). One of the reasons would be that very few *Il11ra1^{-/-}* mice developed large numbers of colorectal tumors comparable to those in *Il11ra1^{+/+}* mice as in Supplementary Fig. 2a. It is unclear whether we had better exclude these *Il11ra1^{-/-}* mice showing large numbers of colorectal tumors. After confirming that each dot showed the normal distribution by the Shapiro-Wilk test, we have performed statistical analysis using the two-tailed unpaired Student's *t*-test. The unpaired Student's *t*-test showed statistical significance. We have only mentioned them in the rebuttal letter.

Regarding contribution of sex differences to the development of CAC, we only used female mice for these experiments, as reported in a previous study.

(15) C, D) It is curious that WT bone marrow led to a difference in tumour number, but not overall area. While IL11KO bone marrow led to a difference in tumour area, but not tumour number. It appears that this is one large experiment. The experiment should be repeated at least once to determine if this observation holds true.

RESPONSE: Thank you for pointing out critical issues. The results included in Supplementary Fig. 2 are pooled results of three independent experiments. Since the effects of deletion of *Il11* in BM cells or non-hematopoietic stromal cells on tumor development was not dramatic, individual differences in response to AOM/DSS treatment might cancel the statistically significant differences of tumor area or tumor numbers between BM chimeric mice. We have only mentioned them in the rebuttal letter.

(16) Further to my comments above, IL11KO bone marrow appears to decrease the overall tumour area. It will be important to determine if eGFP bone marrow makes its way to the tumour of a WT mice.

RESPONSE: As suggested, we performed BM transfer experiments to test whether BM cells were recruited to tumor tissues and then became IL-11⁺ cells. Please see the response to the comment (11).

(17) Does the loss of IL11 similarly affect tumor burden in the APC^{min} mice?

RESPONSE: Thank you for the thoughtful suggestion. As suggested, we crossed *Apc*^{Min/+} mice with *Il11*^{-/-} mice. Consistent with the results using *Il11ra1*^{-/-} mice (Putoczki et al., 2013), deletion of *Il11* gene attenuated the development of tumors in the colon of *Apc*^{Min/+} mice.

In addition, we found expression of *Il11* and *Egfp* was elevated in the tumor tissues compared to non-tumor tissues in the small intestine of *Apc*^{Min/+}; *Il11-Egfp* reporter mice, and IHC revealed that IL-11⁺ fibroblasts appeared in the

tumor tissues in the small intestine (Supplementary Fig. 3b, c). Deletion of *Il11* gene also attenuated the development of tumors in the small intestine of *Apc^{Min/+}* mice. We have made a new Fig. 2i and j to include these results and mentioned them in the text (lines 227-232).

Figure 2.

(18) Line 176, it is an overstatement to suggest that the “tumours educated the IL11⁻ cells to become IL11⁺ cells”, this line should be removed. Moreover, the experiments performed do not demonstrate that “IL11⁻ cells might cell-autonomously become IL11⁺ within the setting of the inflammatory milieu triggered by DSS”. They only reveal their location.

RESPONSE: We agree with your criticism, since we cannot completely exclude the possibility that IL-11⁺ cells are recruited into the colon along with tumor development or inflammation. Thus, we have changed the sentence to “IL-11⁻ cells might cell-autonomously become IL-11⁺ cells in the colonic tissues within the inflammatory milieu or tumors triggered by AOM/DSS.” (lines 209-211).

(19) APC^{min} mice primarily get tumours in the SI and less frequently get tumours in the colon. Do the tumours in any region of the GI tract all have IL11⁺ cells within them? Does the loss of IL-11 in APC^{min} mice impact tumours throughout the GI tract? Or is this specific to the colon?

RESPONSE: Please see the response to the comment (17).

(20) Panel D) in order to say almost all cells expressed Podoplanin, graphs of this quantification is required. Similar comments for all of the markers.

RESPONSE: We agree with your comment. We re-analyzed the results of flow cytometry and calculated the percentages of EGFP⁺ cells expressing each cell surface marker. While approximately 70 to 80% of EGFP⁺ cells expressed Thy1.2 and podoplanin, only 10% of EGFP⁺ cells expressed EpCAM. We have replaced old Fig. 2d with a new one.

(21) Panel E-F) Quantification of the % EGFP⁺ and each marker is required to make conclusive statements. Ie, it is inappropriate to state that “most

IL11+ CAFs were not Ki67+”, without showing the quantification.

RESPONSE: We agree with your comment. As suggested, percentages of IL-11⁺ cells expressing the indicated cell surface markers, including Thy1, podoplanin, and EpCAM (another marker for epithelial cells), were calculated by flow cytometry.

Regarding the populations of IL-11⁺ cells expressing Ki67, we counted Ki67⁺ cells among EGFP⁺ cells in tumor tissues. Only 4 % of EGFP⁺ cells expressed Ki67, suggesting that most IL-11⁺ fibroblasts were not positive for Ki67. We have made a new Fig. 2d and 2g to include these results and mentioned them in the text (lines 225-226).

Supplemental Figure 3:

(22) do the tumour organoids express IL11R?

RESPONSE: Thank you for pointing out a critical issue. To test whether the tumor organoids used here expressed functional IL-11 receptor, we treated the tumor organoids with IL-11. We also treated the organoids with IL-22 as a positive control. IL-11 induced STAT3 phosphorylation in the tumor organoids and human colon cancer cell lines, although the intensity of pSTAT3 was relatively low compared to that induced by IL-22. We have made a new Supplementary Fig. 4e and mentioned them in the text (lines 244-249).

(23) It would be useful to have markers to confirm the organoid tumour growth, and not wounded epithelium following injection of the organoids. Are endoscopy images available? Is histology of the entire tumour with the underlying mucosa available?

RESPONSE: Unfortunately, an endoscopy is not available in our lab. Thus, we included the representative image of transplanted tumors in the colon at lower magnification. Transplanted tumor cells became a large tumor and destroyed normal epithelium in the colon. Thus, these areas were indeed tumor tissues, but not granulation tissues in response to epithelial injury. We have made a new Supplementary Fig. 4b to include the representative image of transplanted tumors in the colon and mentioned them in the text (lines 237-238).

(24) I disagree that the data indicates that the “tumour cells instructed IL11- cells to become IL11+ cells in the absence of inflammation”. I feel quite strongly that the potential for IL11+ cells to be recruited to the tumour site needs to be excluded. It is an important point that a reporter mouse with an IL11-IRES-GFP could address the transient nature of IL11 production, while the current reporter mouse fate maps the production of IL11.

RESPONSE: Thank you for pointing out a critical issue. As suggested, we have changed the sentence to “IL-11⁻ cells became IL-11⁺ cells; alternatively, IL-11⁺ cells were recruited to tumor tissues in the absence of colitis.” (lines 241-243).

To test whether IL-11⁺ cells are recruited from the bone marrow to tumor tissues, we performed BM transfer experiments. Please see the response to the comment (11).

Il11-Egfp mice can only monitor cells that currently express IL-11 but not expressed it in the past. To perform the fate-mapping study of IL-11⁺ cells, we need to generate *Il11-Cre^{ERT2}* mice. Generation of *Il11-Cre^{ERT2}* mice is useful to characterize IL-11⁺ cells in more detail but beyond the scope of the present study.

Figure 3

(25) Line 205, I would suggest altering the first statement from “to test whether inflammation alone induced IL-11+ cell development” (as the experiments do not address the development of the cells), to “to determine whether inflammation alone induced IL-11 expression”. Really, the experiment the reporter mouse allows for is characterization of source and expression. It doesn’t address ‘development’ or ‘regulation’ of gene expression.

RESPONSE: As suggested, we have changed the sentence to “To determine whether colitis alone induced IL-11 expression” (line 253). In addition, we have incorporated this suggestion throughout our paper and changed “development of IL-11⁺ cells” to “expression of IL-11 in cells”.

(26) Panel c) it is not possible to determine if there was a “rapid response” of IL-11 producing cells after 1 day of DSS, without quantification of the

number of IL11+ cells at this stage compared to untreated to determine if this is significant. Indeed, gene expression is low on day 7 (panel a) – but there appears to be some variability in the DSS mice.

RESPONSE: Thank you for picking up a point. Percentages of IL-11⁺ cells were significantly increased in the colon on day 1 and further increased on day 7 after DSS treatment by flow cytometry. In addition, we also showed the kinetics of *Il11* expression in the colon after DSS treatment. We have made new Fig. 3a and e to include these results and mentioned them in the text (lines 254-255 and 259-261). Please also see the response to the comment (9).

(27) Panel d) there are 2% IL11+ cells, compared to the lower % of the tumours in previous figures. Can the authors comment on why there may be so many more IL11+ cells following DSS treatment?

RESPONSE: You have raised an important question. The differences in percentages of IL-11⁺ cells in the colon of DSS-induced colitis and AOM/DSS-induced-CAC may depend on the size of the affected areas in the colon and the strength of the stimuli to induce IL-11. Indeed, the affected areas of the colon of DSS-induced colitis might be larger than those in AOM/DSS-induced CAC (Rebuttal Fig. 3). Moreover, acute and strong oxidative stress induced IL-11 expression in fibroblasts in DSS-induced colitis, whereas chronic signal(s) other than oxidative stress induced IL-11 expression in AOM/DSS-induced CRC. Thus, it is reasonable to surmise that percentages of IL-11⁺ cells in DSS-induced colitis might be higher than those in AOM/DSS-induced CAC. We have only included the results and discussed them

in the rebuttal letter.

Rebuttal Fig. 3. Distribution of IL-11⁺ cells in the colon of *Il11-Egfp* mice treated with DSS or AOM/DSS. **a, b** *Il11-Egfp* mice were treated as in Fig. 3a and colonic tissue sections on day 7 after DSS were stained with anti-GFP antibody. **c, d** *Il11-Egfp* mice were treated as in Fig. 1a and colonic tissue sections on day 98-105 after AOM/DSS were stained with anti-GFP antibody (n = 4 mice). White lines indicate areas containing IL-11⁺ cells. Scale bar, 300 μ m.

(28) Assuming that the number of IL11+ cells rises and then decreases after DSS, ie is low in the non tumour tissue (which is what Fig 1d) suggests, when does IL-11 expression decrease in the AOM/DSS model (water cycle 1, water cycle 2)? What is the role of IL11 in the acute DSS phase compared to the AOM/DSS model?, if it is higher in the acute DSS model one would presume it has a major role. Is there a difference in disease pathology in IL11KO and IL11RKO mice in the acute DSS model?

RESPONSE: Thank you for pointing out critical issues. Our analysis revealed that the expression of *Il11* in the colon peaked on day 7, then declined thereafter, and reached the basal levels on day 14 after DSS treatment (Fig. 3a). We did not check the kinetics of *Il11* expression during the repeated cycles of DSS treatment until CAC development. Thus, it is currently unclear whether 2nd or third round administration of DSS induced *Il11* expression in the same populations of fibroblasts that had responded to 1st DSS administration. To answer this question, we need to perform fate-mapping study of IL-11⁺ fibroblasts by generating *Il11-Cre^{ERT2}* mice. However, such a study is beyond the scope of the present study.

Regarding a functional role for IL-11 in DSS-induced colitis, we included the results of *Il11^{-/-}* and *Il11ra1^{-/-}* mice treated with DSS (Rebuttal Fig. 1). DSS-induced colitis was moderately exacerbated in *Il11^{-/-}* and *Il11ra1^{-/-}* mice compared to respective control mice. Please also see the response to the comment (1). Since the manuscript's length is beyond the upper limit of the standard manuscript in *Nature Communications*, we have only included the results in the rebuttal letter.

(29) Quantification of the IL11+ BrdU+ cells across the sections is required (similar comments to other figures).

RESPONSE: As suggested, we quantified the numbers of BrdU⁺ cells among EGFP⁺ cells. Only 2% of EGFP⁺ cells expressed BrdU. We have included the result in Fig. 3i and mentioned it in the text (lines 267-270).

(30) I would adjust the sentence on line 222/223, as the experiment does not address 'development', but rather indicates presence vs absence.

RESPONSE: As suggested, we changed the sentence to "IL-11 expression in fibroblasts" (lines 270-271).

Figure 4

(31) I find the set-up of this experiment perplexing. What was the profile of the EGFP- sorted cells? Ie were these sorted eGFP- fibroblasts vs eGFP+ fibroblasts? This needs to be clarified in the figure and text, as a direct comparison of one cell type (ie eGFP+, which up to this point the authors suggest are fibroblasts, but have shown include CD45 and EPCAM+ cells in Figure 2) to bulk tissue composed of a number of cell types makes it difficult to determine what the comparison would be expected to show – there would of course be major differences? A better comparison for microarray of bulk RNAseq would be sorted fibroblast eGFP+ vs fibroblast eGFP- to determine if there are differences in the fibroblasts (please see previous comments on what % of fibroblasts are eGFP+). Even better, but beyond the scope of this manuscript would be a single-cell analysis of the eGFP positive cells.

RESPONSE: Sorry for the ambiguous expression in the methods. Indeed, in old Fig. 4, we compared EGFP⁺ cells (mostly IL-11⁺ fibroblasts) and EGFP⁻ cells that included not only IL-11⁻ fibroblasts but also hematopoietic and epithelial cells. We totally agreed with the criticism that comparing gene expression between different cell lineages might not be appropriate. Thus, we sorted EGFP⁺ and EGFP⁻ cells from EpCAM⁻ CD45⁻ Ter119⁻ CD31⁻ cell populations. Sorted cells were then subjected to RNA-seq analysis. Then, we compared gene expression profiles between two populations (IL-11⁺ fibroblasts vs. IL-11⁻ fibroblasts). We

have replaced old Fig. 4 with a new Fig. 4. Accordingly, we have changed the genes for subsequent analysis in Fig. 10 and Supplementary Figs 5, 8, 10 and mentioned them in the text (lines 275-279, 282-288).

(32) Panel E) The authors suggest that IL11+ CAFs after acute DSS are an IAF subtype based on the gene expression profile. Earlier they suggested that IL11+ CAFs were present in tumour from *Apc* mice and CAC mice, so were not specifically inflammation related. If these IL11+ CAFs do not persist throughout the AOM/DSS model, it is not possible to conclude that they are the same IAFs in the cancer models based on the data presented. Is the gene signature of FACS isolated IL11+ cells in the CAC tumours and the *Apc* mice an IAF signature as well, or are they different?

RESPONSE: Thank you for the thoughtful suggestion to compare gene expression profiles of IL-11⁺ fibroblasts from DSS-induced colitis and those from murine tumor models, including CAC in mice and *Apc*^{Min/+} mice. So far, we have not obtained sufficient IL-11⁺ fibroblasts from CAC in mice. Of note, numbers of mononuclear cells prepared from the colon of DSS-treated and AOM/DSS-treated mice before cell sorting did not show a big difference (Rebuttal Table 1). However, numbers of isolated IL-11⁺ fibroblasts after cell sorting from AOM/DSS-treated mice were five to ten times fewer than those from DSS-treated mice (Rebuttal Table 1). Accordingly, qPCR using sorted cells from AOM/DSS-treated mice did not work well (data not shown). Currently, we do not have any idea to explain these differences, but we are currently trying to improve the experimental conditions to sort IL-11⁺ cells from the colon of AOM/DSS-treated mice.

Moreover, numbers of colon tumors in *Apc*^{Min/+} mice were fewer than those in CAC in mice; therefore, it was technically difficult to isolate sufficient IL-11⁺ cells for gene expression analysis. Thus, at this moment, we could not compare gene expression profiles in between DSS-induced IL-11⁺ fibroblasts and tumor-induced IL-11⁺ fibroblasts.

Rebuttal Table 1. Numbers of IL-11+ cells from the colon by cell sorting.				
1) AOM/DSS model				
		Cell numbers		
	Numbers of mice	Before sorting	After sorting	% recovery
Experiment 1	1	2 x 10 ⁷	2543	0.012
Experiment 2	1	1.1 x 10 ⁷	1680	0.015
Experiment 3	3	2.0 x 10 ⁷	566	0.0028
Experiment 4	1	2.6 x 10 ⁷	2184	0.0084
2) DSS model				
		Cell numbers		
	Numbers of mice	Before sorting	After sorting	% recovery
Experiment 1	2	1.8 x 10 ⁷	24195	0.134
Experiment 2	3	2.2 x 10 ⁷	13535	0.062
Experiment 3	1	0.6 x 10 ⁷	12253	0.20

Mononuclear cells were prepared from the colon of the indicated numbers of mice treated as described.

To support our findings using an alternative way, we used the public database (GSE35602) that reported gene expression profiles of tumor cell compartments vs. stromal cell compartments in human colorectal tumor tissues. Expression of a set of genes elevated in IL-11⁺ fibroblasts was similarly elevated in tumor-associated stromal tissues compared to tumor tissues. Although tumor-associated stromal tissues include both IL-11⁻ and IL-11⁺ fibroblasts, these results suggest that human stromal cells have similar gene expression patterns to murine IL-11⁺ fibroblasts. We have made a new Supplementary Fig. 10 to include these results and a slightly modified version of Supplementary Fig. 5a to compare gene expression of tumor tissues and non-tumor tissues in the colon of AOM/DSS-treated mice. We have mentioned them in the text (lines 299-308 and 420-423).

To compare gene expression patterns of IL-11⁺ fibroblasts and IAFs, we performed gene set enrichment analysis. Gene set enrichment analysis revealed that global gene expression profiles between IL-11⁺ fibroblasts and IAFs appeared to be different, although the expression of *IL11*, *IL13RA2*, and *TNFSF13B* was high in both cell populations. Thus, these results suggest that IL-11⁺ fibroblasts had partially overlapping gene signature of IAFs but were not

identical to IAFs. Thus, we have not referred to IL-11⁺ fibroblasts as IL-11⁺ IAFs in the revised manuscript. We have included a new Fig. 4f and mentioned them in the text (lines 295-298).

Supplemental Figure 4:

(33) In following from comment #32 - The IAF signatures they examined in whole tumour tissue could be from other cell populations, as they are not genes restricted to fibroblasts. It would be more convincing to FACS isolate the IL11+ cells from the tumours, and perform qRT or perform Flow to show the IL11+ cells have the expression of the other markers if antibodies are commercially available.

RESPONSE: Please see the response to the comment (32).

(34) I would suggest rewording of line 256, as the data presented do not compare the “phenotype” of the IAF and CAFs as suggested by the concluding statement – but rather, the data suggest the presence of IAFs based on the tumour gene expression. This can be confirmed by the above suggestion.

RESPONSE: We agree with your criticism. As we mentioned in the response to the comment (32), we have not succeeded in performing gene expression analysis of isolated IL-11⁺ cells from tumor tissues. As suggested, we have changed the sentence to “Collectively, IL-11⁺ fibroblasts appeared in DSS-induced colitis may have overlapping, at least in part, gene expression profiles of those in AOM/DSS-induced colon tumors and stromal cells in human CRC (lines 306-308)”.

Figure 5.

(35) In panel C, the 16S expression level appears to average 0.1, while in panel D, it averages at 1. Can the authors explain the reason for such differences in the control cohorts for these experiments?

RESPONSE: Actually, we performed these experiments at different animal facilities in different Universities during our transfer from Juntendo University to Toho University; we performed experiments (c) and (d) in Juntendo University

and Toho University, respectively. Thus, we surmised that the different animal facilities might result in commensal bacteria in the colon of mice and cause such differences.

(36) In panel F, it appears that two outliers contribute to the significance, has this experiment been performed more than once? And do the same trends hold?

RESPONSE: Thank you for pointing out a critical point. If we deleted two outliers from the results, we reached the same conclusion. Moreover, we repeated a similar experiment and obtained the same conclusion.

(37) In Panel E and G the addition of western blots (with 3 mice per group) would better demonstrate the consistency of the loss of pERK. Conclusions can not be supported by one representative image.

RESPONSE: As suggested, we performed Western blotting analysis with anti-phospho-specific ERK antibody using tissue extracts of the colon of mice with or without trametinib treatment. Trametinib completely abolished pERK signals in tissue extracts of the colon of mice treated with trametinib (Rebuttal Fig. 4a). Consistently, areas of pERK-positive cells in the colon of mice were decreased by the treatment of trametinib (Rebuttal Fig. 4b).

The effect of NAC treatment on numbers of pERK⁺ cells is a bit complicated. Under normal conditions, many epithelial cells constitutively express pERK, suggesting that phosphorylation of ERK in normal epithelial cells is induced by homeostatic stimuli but not oxidative stress. Accordingly, the effect of NAC treatment on phosphorylation of ERK was marginal based on the results of Western blotting (Rebuttal Fig. 4c). We assumed that DSS-induced phosphorylation of ERK was largely localized in stromal fibroblasts or infiltrated immune cells, and these signals might be diminished by NAC treatment. Along with this line, we found that areas of pERK-positive cells in the stromal tissues, but not in the epithelial cells of the colon of mice, were decreased by the treatment of NAC (Rebuttal Fig. 4d). To avoid confusion, we have only included the results of counting pERK areas in a new Fig. 5f and i. and mentioned them in the text (lines 328-330).

Rebuttal Fig. 4. Trametinib and NAC inhibit ERK phosphorylation in the colon of DSS-treated mice. **a, b** Wild-type mice were treated with DSS in the absence or presence of trametinib injection (-6 and -30 hours), colonic tissue extracts were analyzed by immunoblotting with anti-pERK and anti-ERK antibodies (n = 4 mice) (**a**). Colonic tissue sections were stained with anti-pERK antibody. pERK⁺ and DAPI⁺ areas were calculated, and the ratios of pERK⁺/DAPI⁺ areas (%) are plotted (**b**). Results are mean ± SE (n = 7 mice). **c, d** NAC inhibits ERK phosphorylation in the colon of DSS-treated mice. Wild-type mice were treated with DSS in the absence or presence of NAC in the drinking water for 5 days. Colonic tissue extracts were analyzed as in (**c**) (n = 4 mice). Colonic sections were stained with anti-pERK antibody. The ratios of pERK⁺/DAPI⁺ areas (%) are plotted as in (**d**). Results are mean ± SE (n = 7 mice). Statistical significance was determined using the unpaired two-tailed Student's *t*-test (**b, d**). ***P* < 0.01; ****P* < 0.001.

(38) The reasoning for panel h are not clearly articulated in the text. One assumes the intent was to demonstrate that TGFβ signalling components are expressed, and thus one presumes are functional.

RESPONSE: Thank you for pointing out the lack of a detailed explanation. Since the TGFβ story is not a major story in our paper, we have moved the results of TGFβ to new Supplementary Figs. 6 and 7. As suggested, we have included the

sentence “Indeed, TGFβ strongly induced IL-11 production by colonic fibroblasts (Supplementary Fig. 6a). While expression of *Tgfb3* was slightly elevated in the colon of DSS-treated mice (Supplementary Fig. 6b), neutralizing antibody against TGFβ did not block *Il11* expression in the colon (Supplementary Fig. 6c).” in the text (line 315-318).

(39) In panel I: do the authors have a control to demonstrate that the concentration of anti-TGFβ that was used successfully neutralised TGFβ signalling? Conversely, does administration of recombinant TGFβ increase IL-11 expression in the colon? Or does Tgfβ addition to eGFP+ cells isolated and grown in culture increase their IL-11 expression? Similar comments for panel J-O, as the authors see no differences, do they have controls to show that the treatments successfully reduced the targets.

RESPONSE: Thank you for pointing out critical comments on our experimental conditions. Anti-TGFβ antibody (clone name 1D11) used in these experiments is widely used to block biological activities of TGFβ in vitro and in vivo (Dasch et al., 1989). As expected, anti-TGFβ antibody blocked TGFβ-induced *Il11* expression by colonic fibroblasts in vitro (Rebuttal Fig. 5a). Although we did not test whether administration of recombinant TGFβ induced elevation of *Il11* in the colon, we confirmed that administration of anti-TGFβ antibody decreased expression of target genes of TGFβ, such as *Foxp3* and *Angptl2* in the colon of DSS-treated mice (Rebuttal Fig. 5b). Since the TGFβ story is not a major point in our paper, we have only included these results in Rebuttal Figure and mentioned them in the rebuttal letter.

Rebuttal Fig. 5. a Anti-TGFβ antibody blocks TGFβ-induced *Il11* expression in colonic fibroblasts. Primary colonic fibroblasts were stimulated with TGFβ (100

ng/mL) in the absence or presence of anti-TGF β antibody (5 μ g/mL) for 4 hours. *Ili1* expression was determined by qPCR. Results are mean \pm SD of triplicate samples. Results are representative of two independent experiments. **b** Treatment with anti-TGF β antibody downregulates target genes of TGF β in the colon of DSS-treated mice. Mice were intraperitoneally administered 100 μ g anti-TGF β antibody on days 2 and 4 after DSS administration and sacrificed on day 5. Expression of the indicated genes was determined by qPCR. Results are mean \pm SE (n = 11–16 mice). Statistical significance was determined using the one-way ANOVA with Tukey's post-hoc test (**a**) or the unpaired two-tailed Student's *t*-test (**b**). **P* < 0.05; *****P* < 0.0001.

(40) I would suggest further description of the experimental results within the results section. Similar comments re: the TGFb expression panels. If data is presented, the reason for its inclusion should be articulated.

RESPONSE: As suggested, we have mentioned the results of expression of *Tgfb1-3* (lines 316-318 and 339-340). Please see the responses to the comments (38) and (39).

(41) The authors show in 3 independent models that the inhibition of MEK/ERK leads to a reduction in IL11 expression and suggest that MEK/ERK is involved in the upregulation of IL11 gene expression. Did any of these treatments alter tumour burden (this should be provided in the supplemental)? This information is important in order to determine if the changes in IL11 expression a reflection of changes in overall tumour burden? Others have shown that IL11 induced pERK (Schafer Nature 2017 7:552), the authors should discuss how that relates to their observation.

RESPONSE: Thank you for pointing out critical issues. Since we treated AOM/DSS-treated mice and *Apc*^{Min/+} mice with trametinib only twice before the sacrifice of mice, we did not see any tumor regression. Thus, we assume that reduction of *Ili1* expression in tumor tissues might not reflect a decrease in the sizes of tumors, but downregulation of *Ili1* expression in fibroblasts. Notably, numbers of Ki67⁺ cells were dramatically decreased, and conversely, numbers of CC3⁺ cells (a hallmark of apoptosis) were increased in colonic tumors of AOM/DSS-treated mice and *Apc*^{Min/+} mice after trametinib treatment compared

to untreated mice. Therefore, repeated administration of trametinib may reduce the sizes of tumors in mice. These experiments are of extreme importance but will be addressed in future studies. We also found similar effects of trametinib on tumors of the small intestine of *Apc^{Min/+}* mice, although a decrease in expression of *Il11* by trametinib was not statistically significant. We made a new Fig. 6e, f, and Fig. 7e-i and mentioned them in the text (lines 342-344 and 348-353).

Regarding contribution of IL-11-induced ERK activation to the expression of *Il11*, we also found that IL-11 stimulation induced phosphorylation of ERK of colonic fibroblasts and colon normal and tumor organoids (Fig. 8d, Supplementary Fig. 4e). Given that IL-11 released from IL-11⁺ fibroblasts activates the MEK/ERK pathway in tumor cells, IL-11 and the MEK/ERK pathway may constitute the feed-forward loop via cancer-associated IL-11⁺ fibroblasts and tumor cells. It is reasonable to speculate that one of the mechanisms of trametinib-induced apoptosis and inhibited proliferation of tumor cells would be caused by the shutoff of this feed-forward loop. We mentioned them in the Discussion (lines 481-490).

(42) For the min model, what was the effect on the SI tumours, was it the same as the colon? (relates to earlier comments about IL11 expression in SI tumours).

RESPONSE: Please see the response to the comment (41).

(43) Overall - how do these observations relate specifically to the fibroblasts? Does this similarly change the number or % of infiltrating eGFP+ cells in the different models? (as other cell populations express IL11)? Does the number of fibroblasts remain the same, but the expression of IL11 within them differ (and does this support the idea of IL11 being turned on vs IL11+ cells being recruited in)? Or are these changes specific to the IL11 expressing epithelial cells and CD45 cells?

RESPONSE: Sorry for confusing percentages of IL-11⁺ epithelial cells and IL-11⁺ hematopoietic cells. As shown in a new Fig. 1e, 2d, and 3f, the majority of IL-11⁺ cells express stromal cell surface markers. Only very few IL-11⁺ cells express EpCAM or CD45. Thus, we focused on IL-11⁺ fibroblasts. Moreover, BM transfer experiments did not appear to support that BM-derived cells were recruited to

tumor tissues and expressed IL-11.

Figure 6

(44) A number of human CRC cell lines have been shown to be responsive to IL11 and have reduced xenograft growth when signalling is inhibited – suggesting that tumour cells respond (Putoczki Cancer Cell 2013 24:257), and tumour colon organoids have previously been shown to be responsive to IL11 (Pheesse 2014 Sci Signal) which contradict the arguments made here. The authors need to dig deeper into these discrepancies. Is there a difference between IL11R expression in normal colon organoids and colon tumour organoids (ie do tumour organoids express IL11R, these should be available from data in previous figures)? How many biological replicates do the organoids represent? (ie were organoids generated from more than one mouse, and does each mouse have the same result?). Do normal organoids and fibroblasts similarly respond to the tumour organoids and matched fibroblasts (ie do both tumour organoids and tumour fibroblasts respond)?

Are there differences between in vitro and in vivo observations.

RESPONSE: Thank you for your thoughtful suggestion. As suggested, we prepared colon organoids from wild-type and *Apc*^{delta716} mice (n = 3 organoids per each group, which were independently established from the colon of different mice) and determined the expression of *Il11ra*. Expression of *Il11ra* and *Il6st* (encoding gp130), a signal transducer for both IL-6 and IL-11, were elevated in *Apc*^{delta716} colon organoids compared to normal colon organoids. Consistently, IL-11 stimulation more efficiently induced phosphorylation of STAT3 in *Apc*^{delta716} colon organoids than in normal colon organoids. We have made a new Fig. 8c and d and mentioned them in the text (lines 367-377).

Regarding the responses of normal colon organoids and colon fibroblasts to IL-11 stimulation, expression of *Il11ra* in colon fibroblasts was higher than that in normal colon organoids. Accordingly, IL-11-induced STAT3 activation in colon fibroblasts was stronger than normal colon organoids (Fig. 8a, b). We mentioned them in the text (lines 362-367).

Regarding isolation of IL-11⁺ CAFs from AOM/DSS-treated mice and comparing the response of normal fibroblasts and IL-11⁺ CAFs, it is technically difficult due to poor recovery of IL-11⁺ CAFs by cell sorting. Please see the

response to the comment (32). Thus, such an experiment will be performed in a future study.

(45) Panel B: would be helpful to have ng/mL on the top axis for the IL11 concentrations.

RESPONSE: As suggested, we have included the units in Fig. 8b.

(46) Panel C: requires additional explanation as to what is being compared? How was the data analysed to determine what is upregulated, are these preparations of the entire colon or fibroblasts specifically? The IL11R agonist should be described more clearly, without the need to refer to previous publications, as it is important to the data conclusions. For example, is this hyper-IL11 (IL11 fused to IL11R) – and thus activating STAT3 in any cells that has gp130 and not specific to an IL11R expressing cell?

RESPONSE: Sorry for the lack of a detailed description of cells or tissues subjected to analysis. We compared gene expression profiles in the whole colon but not isolated fibroblasts from the colon of wild-type mice before and after an IL-11 receptor agonist injection. The IL-11 receptor agonist is a modified version of human IL-11, in which several amino acids in the linker region of human IL-11 were mutated to increase the half-life of degradation (Nishina et al., 2012). We have briefly mentioned it in the text (lines 379-381 and 716-719). Since the heatmap alone in an old Fig. 7c was not informative, we have deleted the heatmap.

(47) Panel E: requires clarification, are the genes elevated in isolated IL-11+ IAFs specifically? Or is this the expression data from Fig 4 of all IL11+ isolated cells? I'm not sure what the intended comparison is... genes IL11+ cells are expressing vs cells where IL11 signalling has been induced by an agonist? Are the authors trying to draw a correlation between a subset of cells that may express IL11 and respond to IL11R in an autocrine manner? What were the 15 genes?

It would be better to show by FACS, and qRT that IL11+ cells also express

IL11R, alternatively Flow to show IL11+ cells are also IL11R+, or a subset of them are. And then validate the expression of the 15 genes identified in these cells (+/- IL11 stimulation in culture).

RESPONSE: Thank you for the thoughtful suggestion. We hypothesize that IL-11 released from IL-11⁺ fibroblasts may act on IL-11⁺ fibroblasts in an autocrine or paracrine manner. To address this issue, we prepared colonic fibroblasts from wild-type mice and then stimulated them with IL-11. We assumed that a set of upregulated genes after IL-11 stimulation might be included in the enriched genes in IL-11⁺ fibroblasts characterized in Fig. 4. However, we stimulated colonic fibroblasts with IL-11, we found that very few genes such as *Metallothioneine 2 (Mt2)* were elevated, suggesting that colonic fibroblasts may be activated during procedures of preparation and subsequent expansion during culture at least our experimental conditions. To circumvent this problem, we injected the IL-11R agonist into wild-type mice. We then compared a set of genes enriched in IL-11⁺ fibroblasts and elevated genes in the whole colon of wild-type mice after IL-11 stimulation. We found approximately 19 genes were overlapped between two sets of genes and confirmed upregulation of some of these genes in the whole colon after the IL-11R agonist stimulation (Supplementary Fig. 8). These genes may be a hallmark of IL-11⁺ fibroblasts activated by IL-11 in an autocrine or paracrine manner. We have included the results in new Fig. 8f and Supplementary Fig. 8 and mentioned them in the text (lines 383-387).

Regarding the expression of IL-11R, commercially available antibodies against IL-11R did not work well. Expression of *Il11ra* in colonic fibroblasts was higher than colon organoids (Fig. 8a) but not different between IL-11⁺ and IL-11⁻ fibroblasts based on RNA-seq analysis (Fig. 4e). These results suggest that IL-11 released from IL-11⁺ fibroblasts can activate both IL-11⁺ and IL-11⁻ fibroblasts.

(48) Supplemental Figure 6, I assume this is colon tissue collected? Is there a difference in gene expression across the colon? (as AOM/DSS tumours arise in the distal colon).

RESPONSE: Yes, we collected the colon tissues and subjected them to qPCR analysis. Since we prepared mRNAs from the whole colon from individual mice,

we could not compare expression of the indicated genes in the proximal and distal colons.

Figure 7

(49) The data presented thus far does not conclusively show that IL11+ IAFs exist, but rather suggests that they may be a sub-population. As such, the heading “genes enriched in IL11+ IAFs...” should be dampened down.

RESPONSE: I assume that the criticism might be the title of old Supplementary Fig. 7, but not Fig. 7. We have made a new Fig. 10a and changed the title to “Some Genes Enriched in IL-11+ Fibroblasts is correlated with reduced recurrence-free survival durations in human CRC” (lines 411-412).

(50) Why is the data in Figure 7A from a breast cancer cell line? And Not a colon cancer cell line? And/or fibroblast cell line?? If this is just validation of the specificity of the antibody, this is better provided in the supplemental information as it is out of place here.

RESPONSE: We included the results to show the validation of anti-IL-11 antibody used in this study. As suggested, we have moved it to Supplementary Fig. 9a.

(51) In order to make the statement that the majority of the cells that expressed IL11 in the tumours were fibroblasts, the data in figure 7e must be quantified.

RESPONSE: As suggested, we quantified percentages of IL-11⁺ cells expressing EpCAM, CD45, or podoplanin. To our surprise, we found that expression of these markers varied in different areas, even in the same tumor tissues. Thus, we assumed that variability of tumor cells and tumor-associated stromal cells in human tumor samples were bigger than we had expected. To avoid confusion, we have deleted the results showing percentages of IL-11⁺ cells expressing EpCAM, CD45, or podoplanin (old Figure 7e).

Supplemental Figure 7

(52) The authors did not present data for IL24, which is part of the IAF

signature, this should be included.

RESPONSE: As suggested, we have included the result of expression of *IL24* in Fig. 10a.

(53) I am not sure I understand the argument that IL11 is not significant in terms of survival, but IL11+ IAFs may relate to survival? The data for IL11/IL6/R/GP130 does not make this conclusion, and would be better with a specific IAF signature. I.e IL11/IL24/IL13RA3/TNFSRFR – which was the signature that they used to suggest that there are IL11 IAFs would be more appropriate.

RESPONSE: Using human colorectal cancer datasets (GSE17537 and GSE14333), we first divided patients into two clusters based on the expression of *IL11* or signature genes of IAFs (*IL11/IL24/IL13RA2/TNFRSR11B*) by the hierarchical clustering method. However, neither clusters did not result in any differences of recurrence-free survival durations (Fig. 10b, c). We next divided patients based on the expression patterns of genes preferentially expressed in IL-11⁺ fibroblasts (2-fold higher than IL-11⁻ fibroblasts), which we referred to as the IL-11⁺ fibroblast signature (IL11FS). Intriguingly, recurrence-free survival durations were significantly decreased in cluster 6 compared with cluster 5 (Fig. 10d). Given that the expression of a set of genes including *HGF*, *IL13RA2*, *PTGS2*, and *TNFSF11* in cluster 6 was higher than in cluster 5 (Supplementary Table 2), these upregulated genes may be associated with poor CRC prognosis. We have made a new Fig. 10b-d, and Supplementary Table 2 to include these results and mentioned them in the text (lines 424-437).

(54) Given the inclusion of IL-6/IL6R – are these expressed by the IL11+ IAFs?

RESPONSE: As suggested, we isolated EGFP⁺ fibroblasts and EGFP⁻ fibroblasts from the colon of DSS-treated mice. RNA-seq analysis revealed that expression of *Il6* and *Il6ra* was not elevated in EGFP⁺ fibroblasts compared to EGFP⁻ fibroblasts (Rebuttal Figure 6). Thus, we deleted the data of expression of *IL6* in a new Fig. 9a.

Rebuttal Fig. 6. RNA-seq analysis of expression of *Il6* and *Il6ra* in EGFP⁻ and EGFP⁺ fibroblasts. RNA-seq analysis was performed by using RNAs from sorted EGFP⁻ and EGFP⁺ fibroblasts. Normalized read counts are shown. Results are mean ± SE (n = 3 mice). Statistical significance was determined using the unpaired two-tailed Student's *t*-test. **P* < 0.05; ns, not significant.

(55) If IL11 is elevated in advanced disease, but does not correlate with survival – what is its role? And how does this compare with its role in colitis? This should be included in the discussion.

RESPONSE: Thank you for pointing out a critical issue. We do not think that IL-11 itself is associated with poor prognosis of CRC. We hypothesized that some enriched genes in IL-11⁺ fibroblasts are involved in the progression of CRC in human patients. Indeed, hierarchical clustering of human CRC patients based on the expression of enriched genes in IL-11⁺ fibroblasts (IL-11FS) divided two clusters. Intriguingly, recurrence-free survival durations were significantly decreased in cluster 6 compared with cluster 5. Given that the expression of a set of genes including *HGF*, *IL13RA2*, *PTGS2*, and *TNFSF11* in cluster 6 was higher than in cluster 5, these upregulated genes may be associated with poor CRC prognosis. We have mentioned them in the text (lines 424-437 and 515-522).

Due to the upper limit of the manuscript, we did not include the results of *Il11ra1*^{-/-} and *Il11*^{-/-} mice after DSS treatment. We found that IL-11 plays a protective role in DSS-induced tissue injury (please see Rebuttal Fig. 1). Assuming that aberrant tissue repair or excessive tissue repair responses along with chronic inflammation result in the development of tumors (Westbrook et al., 2016), IL-11 along with other factors that are overexpressed in IL-11⁺ fibroblasts may participate in the progression of CRC. Since we did not include the results

in Rebuttal Fig. 1 in the current version of the manuscript, we have only discussed this point in the Rebuttal letter.

Discussion

(56) Line 364: Indicates that IL11 IAFs expressed genes associated with proliferation (yet they were Ki67-). The authors should discuss this discrepancy.

RESPONSE: IL-11⁺ fibroblasts may produce several growth factors that induce the proliferation of nearby tumor cells. We have mentioned it in the text (lines 447-448).

(57) Line 380. As the authors did not directly look at gene expression IL11+ FACS isolated cells from tumours, they cannot conclude that the IAFs are related/ not related to tumor CAFs or extend this to human comparisons. Moreover, as they have not investigated chemotherapy response, they cannot exclude that these genes are not induced in response to chemotherapy.

RESPONSE: We agree with your criticism. As suggested, we could not compare the gene expression of IL-11⁺ fibroblasts prepared from the colon of DSS-treated mice and tumor tissues of AOM/DSS-treated mice. Thus, we have changed the description of “IL-11⁺ IAFs” and “IL-11⁺ CAFs” to “IL-11⁺ fibroblasts” throughout the manuscript.

(57) There is no data or reference presented to support line 385. If this is a statement based on previous publications, this should be references. If this is data they have generated, it should be appropriately included in the results (only TGFb inhibition was presented).

RESPONSE: As suggested, we have included the results that TGFβ induced IL-11 production by colonic fibroblasts in Supplementary Fig. 6a.

(58) There is no data from line 390. Either a reference from previous studies should be provided, or the data included in the manuscript.

RESPONSE: Since the culture supernatant of *AKTP* tumor organoids did not induce IL-11 production by ELISA and TGF β story is not relevant to our study, we have only included the results in Rebuttal Fig. 7 and deleted the sentence in the manuscript.

Rebuttal Fig. 7. Culture supernatant of *AKTP* organoids does not induce IL-11 production by colonic fibroblasts. Colonic fibroblasts were untreated or stimulated with culture supernatants of *AKTP* organoids or TGF β for 16 hours. Concentrations of IL-11 in culture supernatants were determined by ELISA. Results are mean \pm SD of triplicate samples. Results are representative of two independent experiments. Statistical significance was determined using the one-way ANOVA with Tukey's post-hoc test. **** $P < 0.0001$; ns, not significant.

(59) The authors should discuss their results in light of Calon et al (Cancer Cell 2012) and Yuan et al (Cancer Cell 2014 25:666)

RESPONSE: Thank you for the thoughtful suggestion. Their results showed that ectopic overexpression of TGF β in tumor cells induces IL-11 expression in stromal fibroblasts. We assume that expression levels of TGF β in tumor tissues in CAC in mice or *Apc^{Min/+}* mice might not be elevated compared to tumors that exogenously expressed TGF β . Accordingly, blockade of TGF β signal did not inhibit upregulation of *Il11* in stromal fibroblasts in these tumor models. Combined these data, TGF β can induce *Il11* expression in vivo; however, under our experimental conditions, tumor cells do not produce TGF β sufficient to induce *Il11* expression in non-tumor cells. We have them in the text (lines 473-481).

(60) Heichler et al (Gut 2019), Cleary demonstrated that IL-11 induces STAT3 activation in both colon fibroblasts and CAFs and performed gene expression analysis, so this idea is not novel. How their results compare with that of the authors must be included in their figure, and discussed in detail.

RESPONSE: As requested, we compared global gene expression profiles of enriched genes in IL-11⁺ fibroblasts and upregulated genes in IL-11-stimulated ColVI⁺ fibroblasts. We found that very few genes were overlapped in two gene populations. Thus, at this moment, IL-11⁺ fibroblasts do not appear to be identical to IL-11-stimulated ColVI⁺ fibroblasts. We have included the results in a new Fig. 8e and mentioned them in the text (lines 388-394 and 504-508).

Minor comments:

The manuscript would benefit from some minor English language editing.

RESPONSE: Although the San Francisco Edit had edited the manuscript, we have asked them to edit the revised version of the manuscript again.

(61) The methods indicates that mice induced with DSS were 9-15 weeks old, which is a significant variation in age. Are the data presented all age and gender matched?

RESPONSE: Yes, we used age- and sex-matched mice for the experiments.

(62) The methods give DSS disease scores, but no DSS disease scores are presented in the manuscript.

RESPONSE: Please see the response to the comment (4). Since we have not included the results of DSS-induced colitis in *Il11ra1*^{-/-} and *Il11*^{-/-} mice, we have deleted the methods to evaluate DSS disease scores.

(63) Do the culture fibroblasts secrete IL11 into the media? An IL11 Elisa is indicated in the methods, but no IL11 Elisa Data is shown.

RESPONSE: Sorry for the lack of the results of IL-11 production by fibroblasts.

Please see the response to the comment (57).

(64) The clone/ supplier of the anti-IL11 antibody was not provided.

RESPONSE: We have included the detailed information about human anti-IL-11 antibody (LS-C408373, LSBio) in the Methods (line 536).

Reviewer #2 (Remarks to the Author): expertise in fibroblasts and colitis/colorectal cancer

The manuscript „Interleukin-11 is a Marker for Both Cancer- and Inflammation-Associated Fibroblasts that Contribute to Colorectal Cancer Progression“ by Nishina et al. deals with an interesting research topic. Colon cancer is still of high clinical relevance and studies of fibroblasts both cancer-associated and inflammation-associated are a frequent focus of current GI tract research. The paper is well written and adds information on IL-11 producing cells in the intestine. The authors suggest a concept in which tumor cells induce IL-11 in fibroblasts, and an autocrine feed-forward loop between IL-11 and IL-11+ fibroblasts contributes to tumor development. This concept is not entirely new, however.

The authors report a novel IL-11 –EGFP-reporter mouse that could evolve as a very helpful tool for the research community, in particular as antibody-based IL-11 staining can be challenging. The authors studied IL-11 production in three murine tumor models of CRC, and one murine acute colitis model using reporter mice. The authors report that the majority of IL-11+ cells are cancer- or inflammation-associated fibroblasts, respectively. The manuscript includes profiling of gene transcription patterns of the IL-11+ cells from the DSS model by microarray analysis. However, the comparative group (of total IL-11- cells?) is probably not optimum. They report that IL-11+ cells express genes associated with cell proliferation, tissue repair and suggest that the transcription profiles are also connected with tumor development. They provide interesting evidence with in vivo data that rather ERK/MEK signaling than TGFbeta signaling is the driver of IL-11 induction. The supplemental material contains a large amount of additional findings. The authors report that IL-11 deletion attenuated CAC development using IL-11 k.o. mice (and provide elegant evidence by studies with BM chimeric mice that the non-hematopoietic of the IL+11 cells are critical) which fits to the known phenotype of IL-11r k.o. mice and the already established concept of IL-11 signaling during in CRC. The manuscript is mainly a murine study, but it also includes wet and dry bench analysis of human data.

The main strength and novelty of the paper is related to the novel reporter mouse. However, I am not enthusiastic because further functional insights

into the role of IL11+ CAFs in CRC or IL11+IAFs in colitis are limited. Concerning this matter, the authors present mainly expression data, correlative findings and circumstantial evidence, but functional data on the role of IL-11+CAFs and IL-11+IAFs on tumor development or colitis severity are lacking. The human data presented are rather weak although claims are made in the abstract and the title. A few shortcomings are related to the methodology, e.g. selection of appropriate controls and how some of the data are presented (see below). Some of the claims are definitely too far-fetched and not sufficiently supported by experimental data. The authors are advised to tone down their claims (abstract, title).

We really appreciate the positive comments on the generation of *Il11-Egfp* reporter mice and characterization of IL-11⁺ fibroblasts. As suggested, we performed a large number of experiments to address the reviewers' concerns and have substantially improved the manuscript. We have tone downed the contribution of IL-11⁺ fibroblasts to the development of colorectal tumors in the text. Accordingly, we have changed the title to "Interleukin-11-expressing fibroblasts have a unique gene signature correlated with poor prognosis of colorectal cancer".

Indeed, we have only shown the correlation of appearance of IL-11⁺ fibroblasts and the development of tumors in several murine models in the present study. To directly demonstrate the causal relationship between IL-11⁺ fibroblasts and the progression of colorectal cancer will be addressed in the future study.

Specific comments:

- The expression data/functionality of the reporter mouse is only partly convincing, FACS data on tumors as presented do not show a clearly distinctive population of IL-11+(EGFP+) cells. Most of the expression data are immunostaining data with a green signal. Without adequate controls, this signal could even result from autofluorescence

RESPONSE: Thank you for pointing out a critical issue. Indeed, since numbers of IL-11⁺ fibroblasts were very few in the colon of mice treated with AOM/DSS, IL-11⁺ fibroblasts did not appear to constitute a distinct cell population. Regarding the specificity of EGFP signals, indeed, we found autofluorescence of

unstained tissue sections (data not shown). However, we used blocking solutions as described in Methods and finally succeeded in abolishing autofluorescence of tissue sections (lines 749-751). As shown in Rebuttal Fig. 8a, we could not detect EGFP⁺ cells in untreated *Il11-Egfp* reporter mice but detected them in AOM/DSS- and DSS-treated *Il11-Egfp* reporter mice by IHC. In sharp contrast, EGFP⁺ cells were not detected in the colon of untreated, even AOM/DSS- or DSS-treated WT mice. Importantly, most EGFP⁺ cells were also stained with anti-IL-11 antibody (Rebuttal Fig. 8b, c), further substantiating that these EGFP signals represent the expression of IL-11.

Moreover, IL-11⁺ cells were not detected in the colon of untreated wild-type or *Il11-Egfp* reporter mice but detected DSS-treated *Il11-Egfp* reporter mice by flow cytometry (Rebuttal Fig. 8d). Together, these results indicate that *Il11-Egfp* reporter mice specifically monitor IL-11⁺ cells. We have made new Figs. 1h, i, 3e, and Supplementary Fig. 1e to include these results and mentioned them in the text (lines 155-160 and 259-261).

Rebuttal Fig. 8. *Il11-Egfp* mice monitor IL-11⁺ cells in vivo. Wild-type (**a, d**) and *Il11-Egfp* reporter (**a-d**) mice were untreated or treated with AOM/DSS or DSS as in Fig. 1a and 3a. Colon tissue sections (untreated or DSS-treated mice) or

colon tumor sections (AOM/DSS-treated mice) were stained with anti-GFP (**a**) or both anti-GFP and anti-IL-11 antibodies (**b**, **c**). Right panels are enlarged images of the left boxes (**b**). Numbers of IL-11⁺ cells among EGFP⁺ cells were calculated, and their percentages are plotted (**c**). Results are mean \pm SE (n = 4 mice). White arrows indicate accumulation of cells expressing both IL-11 and EGFP in the tumor tissues. Scale bar, 100 μ m. **d** WT, and *Il11-Egfp* reporter mice were untreated or treated with DSS as in Fig. 3a. EGFP⁺ (IL-11⁺) cells were analyzed by flow cytometry. Percentages of IL-11⁺ cells were calculated and plotted. Results are mean \pm SE (n = 4 mice). Statistical significance was determined using the one-way ANOVA with Tukey's post-hoc test (left panel) or the unpaired two-tailed Student's *t*-test (right panel). **P* < 0.05; ***P* < 0.01.

OK - recent papers in the field should be discussed, see Heickler et al. Gut. 2019 Nov 4. pii: gutjnl-2019-319200

RESPONSE: Thank you for the thoughtful suggestion. As suggested, we have discussed our results, citing the paper by Heicker et al (lines 388-394 and 504-508).

OK - although there are similarities between them, cancer/CAFs and inflammation/IAFs are different diseases/cells -> suggestive transfers should be avoided, unless no direct comparison is provided.

RESPONSE: We totally agreed with your suggestion. Thus, we have changed the sentence "IL-11⁺ IAFs" or "IL-11⁺ CAFs" to "IL-11⁺ fibroblasts" throughout the manuscript.

- Fig1: d) reporter mouse signal in FACS staining not very convincing (low numbers <1%, no clear signal of a distinctive population), e) legend/labeling/gating strategy/percentages unclear; g-h controls are missing; a reporter mouse enables detailed and precise quantifications in disease models -> the composition of the pool of IL-11+ cells should be better quantified (FACS data with mean +/-range/SD in mice), please add information on relative number of IL-11 producing cells among fibroblasts + other subsets f)control tissue missing

RESPONSE: Thank you for indicating a critical point. As suggested, we have made a new Supplementary Fig. 1d to include a gating strategy for analyzing IL-11⁺ cells. In addition, we performed additional experiments and re-analyzed the results of flow cytometry. We calculated the percentages of EGFP⁺ cells expressing each cell surface marker. The majority of IL-11⁺ cells expressed mesenchymal stromal cell markers, such as Thy1.2, podoplanin, CD29, and Sca-1, but not CD31 or Lyve-1, whereas only very small percentages of IL-11⁺ cells expressed EpCAM.

On the other hand, 4% podoplanin⁺ cells (mostly fibroblasts) expressed EGFP, suggesting that only minor populations of fibroblasts did express IL-11. We have changed old Fig. 1d to a more representative one and replaced an old Fig. 1e with a new one. We have made new Fig. 1f to include these results and mentioned them in the text (lines 150-160).

Regarding the control tissue staining with anti-GFP antibody, please see Rebuttal Fig. 9a. These results indicate that *Il11-Egfp* reporter mice did not reflect autofluorescence but did specifically monitor IL-11⁺ cells.

Fig2: d) please improve quantitative information (as above), e-g) controls are missing / tissue from non-reporter mice

RESPONSE: As suggested, we re-analyzed the results of flow cytometry and calculated the percentages of EGFP⁺ cells expressing each cell surface marker. While approximately 70 to 80% of GFP⁺ cells expressed Thy1.2 and podoplanin, only 10% of GFP⁺ cells expressed EpCAM.

To further verify the specificity of EGFP signals in the colon of *Apc^{Min/+};Il11-Egfp* mice, we immunostained nontumor and tumor tissues of the colon of *Apc^{Min/+}* and *Apc^{Min/+};Il11-Egfp* mice with anti-GFP antibody. We could not detect any EGFP⁺ cells in nontumor tissues of the colon of *Apc^{Min/+}* mice or *Apc^{Min/+};Il11-Egfp* mice. Of note, we did detect EGFP⁺ cells in tumor tissues of the colon of *Apc^{Min/+};Il11-Egfp* mice, but not *Apc^{Min/+}* mice. These results further substantiate that anti-GFP antibody specifically detected EGFP⁺ (IL-11⁺) cells that appeared in tumor tissues of the colon of *Apc^{Min/+};Il11-Egfp* mice. We have replaced old Fig. 2d with a new one and included the results in Supplementary Fig 3a, and mentioned them (lines 217-223).

Fig3: DSS FACS seems o.k., e) please improve quantitative information (as above); FACS data with the reporter mouse on the kinetics of IL-11+ cells, especially in the acute DSS model, would be helpful.

RESPONSE: As suggested, we have changed the FACS profiles of IL-11⁺ (EGFP⁺) cells to more representative ones. Moreover, we analyzed the kinetics of percentages of IL-11⁺ cells and expression of *Il11* in the colon of *Il11-Egfp* reporter mice on days 1 and 7 after DSS treatment. Small numbers of IL-11⁺ cells rapidly appeared in the colon of DSS-treated mice on day 1 and further increased on day 7. Consistently, expression of *Il11* was very low before or on day 4 but peaked on day 7 and then declined on day 9 after DSS treatment. We have made new Fig. 3a, e, and f to include these results and mentioned them in the text (lines 253-255 and 259-261).

Fig4: what was the composition of the IL-11- and the IL11+ population? Some genes with highest upregulation in the IL11-(EGFP-) should be also presented. Please clarify, if epithelial cells and/or hematopoietic cells were excluded before analysis. If not: (and IL-11 is mainly found in IAFs) the comparison would not be that fair -> gene expression could be rather a representation of a general fibroblast signature instead of an IL11+fibroblast signature -> validation experiments with more homogenous cell populations and IL11+fibroblasts would be helpful.

RESPONSE: Sorry for the ambiguous expression in the methods. Indeed, in old Fig. 4, we compared EGFP⁺ cells (mostly fibroblasts) and EGFP⁻ cells that included not only fibroblasts but also hematopoietic and epithelial cells. We totally agree with the reviewer's criticism that comparing gene expression between different cell lineages might not be appropriate. Thus, we sorted EGFP⁺ and EGFP⁻ cells among EpCAM⁻ CD45⁻ Ter119⁻ CD31⁻ cell populations and subjected them to RNA-seq analysis. Then, we compared gene expression profiles between two populations. We have replaced an old Fig. 4 with a new Fig. 4. Accordingly, we have changed the genes for subsequent analysis in Fig. 10, Supplementary Figs 5, 8, 10 and mentioned them in the text (lines 276-279, 283-288, and 295-308).

Fig5: was any influence on tumor growth detected? Are more CAF/IAF

specific data available in this context?

RESPONSE: RESPONSE: Thank you for pointing out critical issues. Since we treated AOM/DSS-treated mice and *Apc*^{Min/+} mice with trametinib only twice before sacrificing mice, we did not see any tumor regression. Thus, we assume that reduction of *Il11* expression in tumor tissues might not reflect a decrease in tumor size but downregulation of *Il11* expression in fibroblasts. Notably, numbers of Ki67⁺ cells were dramatically decreased, and conversely, numbers of CC3⁺ cells (a hallmark of apoptosis) were increased in colonic tumors of AOM/DSS-treated mice and *Apc*^{Min/+} mice after trametinib treatment compared to untreated mice. Therefore, repeated administration of trametinib may reduce the sizes of tumors in mice. These experiments are of extreme importance but will be addressed in future studies.

We also found similar effects of trametinib on tumors of the small intestine of *Apc*^{Min/+} mice, although a decrease in expression of *Il11* by trametinib was not statistically significant. We made a new Fig. 6e, f, and Fig. 7e-i to include these results and mentioned them in the text (lines 348-353 and 481-490).

Fig 6: c-e) fig legend/labeling of the figure not clear, please clarify IL-11R agonist (=IL-11?) and indicate what was compared in e)

RESPONSE: Sorry for the lack of a detailed description of the experimental procedures and an IL-11R agonist. The IL-11R agonist is a modified version of IL-11 in which several amino acids were mutated to increase the stability and biological activity of human IL-11 (Yanaka et al., 2011). We have included the text and Methods (lines 379-381 and 716-719).

Regarding the comparison IL-11⁺ fibroblasts and IL-11-stimulated whole colon, we compared genes enriched in IL-11⁺ fibroblasts (more than 2-fold higher than those in IL-11⁻ fibroblasts) and upregulated genes in the whole colon (more than 2-fold higher than those in the unstimulated whole colon after the IL11R agonist injection). This analysis may identify genes enriched in IL-11⁺ fibroblasts that were upregulated with IL-11 stimulation. These results suggest that IL-11 activates, at least in part, IL-11⁺ fibroblasts in an autocrine or paracrine manner, but other signals may also induce other enriched genes in IL-11⁺ fibroblasts. We have mentioned them in the text (lines 384-387).

- Fig7: limited novelty of information, technical: strong IL-11 staining by human tumor epithelial cells in e) seems contradictory to b)+ the murine data + the overall concept of the manuscript; missing isotype controls; the function of a) is unclear: antibody-testing in WB does not need to correlate with specificity in stainings on tissue sections (and scramble siRNA as control is missing c) unit is missing (per cent?) d) unit is missing; what is meant by early and advanced cancers? Please specify with more accurate clinical terms and further information.

RESPONSE: Thank you for the thoughtful suggestion. Although we counted numbers of IL-11⁺ cells expressing CD45, E-cadherin, or vimentin, numbers of these double-positive cells appeared to vary depending on the areas of tumors, even in the same sample. Numbers of tumor samples analyzed in this study were relatively few; thus, we could not perform statistical analysis. Therefore, we have deleted old Fig. 7e to avoid confusion.

Regarding the specificity of anti-IL-11 antibody used in this study, we first tested whether IL-11 signal was abolished in cells treated with *IL11* siRNAs, but not control siRNA ([-] indicated control siRNA in old Fig. 7a) by Western blotting. We assumed that this strategy might be an alternative method to verify the specificity of anti-IL-11 antibody, since it was technically impossible to use human tumor samples lacking *IL11* gene. Of course, we cannot rule out the possibility that anti-IL-11 antibody working on Western blotting does not work well in IHC. We have moved the result to Supplementary Fig. 9a. To show the specificity of IL-11 staining, we have also included the results of tissue staining of advanced colorectal tumors using isotype control antibody (Supplementary Fig. 9b).

As suggested, we have included the clinical information of patients used in the present study in Supplementary Table 1.

Sorry for not including the unit in Fig. 9. IL-11⁺ area and total area were calculated and the percentages of IL-11⁺ area per area are plotted (b). The intensities of IL-11⁺ signals and total area were calculated and intensities of IL-11⁺ per area are expressed as arbitrary units (a.u.) (c). We mentioned them in the text (lines 755-757 and 1301-1304).

Suppl. fig1: correlate the protein signal from the reporter mouse with the

RNA expression data?

RESPONSE: Of note, expression of *Il11* and *Egfp* was correlated with EGFP⁺ cells by IHC (Fig. 1c, 1g, 2a, 2b, 3a, and 3b).

- Suppl fig 7: this is the only human data (along with Fig7): the cutoff levels for the definition of the subgroups in b) and the n per subgroup should be provided; the authors interpretation “IL-11+ fibroblasts correlated with reduced disease-free survival” seems highly speculative based on the data presented.

RESPONSE: Thank you for pointing out a critical issue. As suggested, we have tone downed and changed the sentence to “Some Genes Enriched in IL-11+ Fibroblasts is correlated with reduced recurrence disease-free durations in human CRC”.

Regarding the cutoff levels between each cluster, we have applied the hierarchical clustering methods to divide each cluster. Thus, we have not set the cutoff levels. As suggested, we have included numbers of patients in each cluster.

Using human colorectal cancer datasets (GSE17537 and GSE14333), we first divided patients into two clusters based on the expression of *IL11* or combination of *IL11/IL24/IL13RA2/TNFRSF11B* expression by the hierarchical clustering method. However, neither clusters did not result in any differences of recurrence-free survival durations (Fig. 10b, c). We next divided patients based on the expression patterns of genes preferentially expressed in IL-11⁺ fibroblasts (2-fold higher than IL-11⁻ fibroblasts), which we referred to as the IL-11⁺ fibroblast signature (IL11FS). Intriguingly, recurrence-free survival durations were significantly decreased in cluster 6 compared with cluster 5 (Fig. 10d). Given that the expression of a set of genes including *HGF*, *IL13RA2*, *PTGS2*, and *TNFSF11* in cluster 6 was higher than in cluster 5 (Supplementary Table 2), these upregulated genes may be associated with poor CRC prognosis. We have made a new Fig. 10b-d, and Supplementary Table 2 to include these results and mentioned them in the text (lines 424-437).

Supplementary References

Dasch, J.R., Pace, D.R., Waegell, W., Inenaga, D., and Ellingsworth, L. (1989). Monoclonal antibodies recognizing transforming growth factor-beta. Bioactivity neutralization and transforming growth factor beta 2 affinity purification. *J Immunol* 142, 1536-1541.

Nishina, T., Komazawa-Sakon, S., Yanaka, S., Piao, X., Zheng, D.M., Piao, J.H., Kojima, Y., Yamashina, S., Sano, E., Putoczki, T., *et al.* (2012). Interleukin-11 links oxidative stress and compensatory proliferation. *Sci Signal* 5, ra5.

Putoczki, T.L., Thiem, S., Loving, A., Busuttil, R.A., Wilson, N.J., Ziegler, P.K., Nguyen, P.M., Preaudet, A., Farid, R., Edwards, K.M., *et al.* (2013). Interleukin-11 is the dominant IL-6 family cytokine during gastrointestinal tumorigenesis and can be targeted therapeutically. *Cancer Cell* 24, 257-271.

Westbrook, A.M., Szakmary, A., and Schiestl, R.H. (2016). Mouse models of intestinal inflammation and cancer. *Arch Toxicol* 90, 2109-2130.

Yanaka, S., Sano, E., Naruse, N., Miura, K., Futatsumori-Sugai, M., Caaveiro, J.M., and Tsumoto, K. (2011). Non-core region modulates interleukin-11 signaling activity: generation of agonist and antagonist variants. *J Biol Chem* 286, 8085-8093.

REVIEWERS' COMMENTS

Reviewer #1 (Remarks to the Author):

Minor comment: It would have been useful to have the text changes in a different colour in the manuscript for easier review of the text changes that have been made.

Major comment: figure legends need to include the number of representative experimental repeats performed, animal genders and animal ages consistently.

Major comment: methods need to outline all controls consistently.

Major comment: Unfortunately, since the last review of this manuscript it has been brought to my attention that there is already an IL-11 GFP reporter mouse that has been published and is available in the public domain: <https://www.jax.org/strain/034466>, this unfortunately diminishes my enthusiasm for the mouse described within the manuscript. I do note however, that the published mouse does not appear to have been characterised in the context of CRC at this point.

In general: The overstated text has been appropriately dampened down.

The authors have addressed the majority of the comments / experiments suggested; however, technical limitations (ie FACS profiling) appear to remain.

Additional comments / concerns are outlined below.

(1) Title change: seems appropriate, but becomes more of an observation style paper, rather than mechanistic.

(2) Supplemental Figure 2 C/D) The authors should indicate in the legend that the results are pooled from 3 independent experiments. It is the opinion of this reviewer that this is not the appropriate way to present AOM/DSS data – as there can be considerable variation between experiments (as DSS water intake can not be regulated), and box to box variation. I would be more comfortable with one individual experiment presented, and indicated that it is representative of 3 independent experiments. This comment holds for any experiment presented with pooled data.

Reviewer #2 (Remarks to the Author):

Thank you for addressing my concerns. I appreciate it. The revisions made by the authors have substantially improved the manuscript. I recommend that the figure presented in the point-by-point Reply in response to my first comments (Rebuttal Fig 8) should be included in the manuscript, e.g. in the supplement.

REVIEWERS' COMMENTS

Reviewer #1 (Remarks to the Author):

Minor comment: It would have been useful to have the text changes in a different colour in the manuscript for easier review of the text changes that have been made.

RESPONSE: As suggested, we have indicated the text changes shown in yellow.

Major comment: figure legends need to include the number of representative experimental repeats performed, animal genders and animal ages consistently.

RESPONSE: Thank you for pointing out critical important issues. As also pointed out by the editorial team, we have mentioned how many experiments were replicated or pooled, and numbers of mice used in the experiments in the legends. Regarding the genders, we usually used male mice for DSS-induced colitis, female mice for AOM/DSS-induced colorectal cancer (because male mice were more susceptible to repeated administration of DSS and easily succumbed compared to female mice), and both female and male mice for analysis of the colon and small intestinal tumors in *Apc^{Min/+}* mice. To reduce the length of the legends and increase readability, we have made a new Supplementary Table 2 to include the detailed information about the genotypes, genders, and ages of mice used in the experiments.

Major comment: methods need to outline all controls consistently.

RESPONSE: As suggested, we have described the methods in more detail as far as we can.

Major comment: Unfortunately, since the last review of this manuscript it has been brought to my attention that there is already an IL-11 GFP reporter mouse that has been published and is available in the public

domain: <https://www.jax.org/strain/034466>, this unfortunately diminishes my enthusiasm for the mouse described within the manuscript. I do note however, that the published mouse does not appear to have been characterised in the context of CRC at this point.

RESPONSE: Thank you for bringing our attention to the deposition of IL-11 GFP reporter mice in the Jackson Laboratory. So far, the investigators only reported that GFP-positive cells are detected in hepatocytes following APAP-induced liver injury using their IL-11 GFP mice (<https://doi.org/10.1101/830018>). Since they deposited their paper in the BioRxiv, their manuscript has not been formally published following critical reviewed yet. Moreover, it is unclear whether their IL-11 GFP reporter mice are working well in other murine models, including DSS-induced colitis or AOM/DSS-induced colorectal cancer. Taken together, our *Il11-Egfp* mice may be suitable for analyzing in IL-11⁺ cells in murine colitis and colorectal cancer models at this moment.

In general: The overstated text has been appropriately dampened down. The authors have addressed the majority of the comments / experiments suggested; however, technical limitations (ie FACS profiling) appear to remain.

RESPONSE: As suggested by the reviewer and the editorial team, we have made a new Supplementary Figure 5a to include the sorting strategy to isolate EGFP⁺ cells from DSS-treated *Il11-Egfp* reporter mice (Supplementary Materials, lines 98-102). Accordingly, an old Supplementary Figure 5a and 5b are changed to a new Supplementary Figure 5b and 5c, respectively.

Additional comments / concerns are outlined below.

(1) Title change: seems appropriate, but becomes more of an observation style paper, rather than mechanistic.

RESPONSE: According to your advice, we have toned down the claim that IL-11⁺ fibroblasts contribute to the development of colorectal tumor in mice and human and poor prognosis of cancer.

(2) Supplemental Figure 2 C/D) The authors should indicate in the legend that the results are pooled from 3 independent experiments. It is the opinion of this reviewer that this is not the appropriate way to present AOM/DSS data – as there can be considerable variation between experiments (as DSS water intake cannot be regulated), and box to box variation. I would be more comfortable with one individual experiment presented, and indicated that it is representative of 3 independent experiments. This comment holds for any experiment presented with pooled data.

RESPONSE: I have understood what you have meant. However, it is technically difficult to prepare sufficient numbers of knockout mice and their littermate control mice with the same age and gender at the same time. Moreover, the fact that some mice unexpectedly died during the repeated administration of DSS further might hamper the opportunity to obtain sufficient numbers of mice in one experiment. Therefore, we had to repeat the same experiments several times and pooled two or three independent experiments and performed statistical analysis. Regarding *Apc^{Min/+};I111-/-* double mutant mice, we needed to pool seven experiments to obtain sufficient numbers of mice for statistical analysis (Fig. 2i, j). In contrast, if we obtain sufficient numbers of mice from one experiment, according to the reviewer's suggestion, we have not pooled the results, but analyzed the data from one experiment (new Supplementary Figures 5b and 8). We have clearly mentioned these points in the indicated Figure legends (lines 1193-1194; Supplementary Materials, lines 105-106; 160-161).

In some experiments, we pooled five to seven experiments and performed statistical analysis (Figure 5g, Supplementary Figure 2a-d, and 6c), even we had obtained sufficient numbers of mice from two or three independent experiments. As suggested by the reviewer, we think that these experiments may not be suitable. Thus, we have divided these pooled experiments to two or three pooled experiments and analyzed the data (lines 1273-1274; Supplementary Materials, lines 50-60; 126-127). Accordingly, the numbers of mice presented in the indicated Figures were different from the previous ones, resulting in changes of the exact *p*-values. However, these changes do not affect the main conclusion of our study.

Reviewer #2 (Remarks to the Author):

Thank you for addressing my concerns. I appreciate it. The revisions made by the authors have substantially improved the manuscript. I recommend that the figure presented in the point-by-point Reply in response to my first comments (Rebuttal Fig 8) should be included in the manuscript, e.g. in the supplement.

RESPONSE: Thank you for the appreciation of the revision that we have made. Regarding the results presented in Rebuttal Figure 8, we already included these or similar results in Figure 1h and i (corresponding to Rebuttal Figure 8b and c), Figure 3e (corresponding to Rebuttal Fig 8d), and Supplementary Fig. 1e (corresponding to Rebuttal Figure 8a) in the revised version of the Figures.